**Large Contributions from Biogenic Monoterpenes and Sesquiterpenes to Organic Aerosol in the Southeastern United States**

Lu Xu[1,a], Havala O.T. Pye[2], Jia He[3], Yunle Chen[4], Benjamin N. Murphy[2], Nga Lee Ng[1,3*]

[1]School of Chemical and Biomolecular Engineering, Georgia Institute of Technology, Atlanta, GA 30332, USA

[2]National Exposure Research Laboratory, US Environmental Protection Agency, Research Triangle Park, NC 27711, USA

[3]School of Earth and Atmospheric Sciences, Georgia Institute of Technology, Atlanta, GA 30332, USA

[4]School of Materials Science and Engineering, Georgia Institute of Technology, Atlanta, GA 30332, USA

[*]To whom correspondence should be addressed. E-mail: ng@chbe.gatech.edu

[a]Present address: Division of Geological and Planetary Sciences, California Institute of Technology, Pasadena, CA 91125, USA

## Abstract

Atmospheric organic aerosol (OA) has important impacts on climate and human health but its sources remain poorly understood. Biogenic monoterpenes and sesquiterpenes are critical precursors of OA. The OA generation from these precursors predicted by models has considerable uncertainty owing to a lack of appropriate observations as constraints. In this study, we propose that the less-oxidized oxygenated organic aerosol (LO-OOA) factor resolved from positive matrix factorization (PMF) analysis on aerosol mass spectrometry (AMS) data can be used as a surrogate for fresh SOA from monoterpenes and sesquiterpenes in the southeastern U.S. We support this hypothesis based on a weight of evidence, including lab-in-the-field perturbation experiments, extensive ambient ground-level measurements, and state-of-the-art modeling. We performed lab-in-the-field experiments, in which the ambient air is perturbed by the injection of selected monoterpenes and sesquiterpenes and subsequent SOA formation. PMF analysis on the perturbation experiments provides an objective link between LO-OOA and fresh SOA from monoterpenes and sesquiterpenes as well as insights into the sources of other OA factors. Further, we use an upgraded atmospheric model and show that modeled SOA concentrations from monoterpenes and sesquiterpenes could reproduce both the magnitude and diurnal variation of LO-OOA at multiple sites in the southeastern U.S., building confidence in our hypothesis. We predict the annual average concentration of SOA from monoterpenes and sesquiterpenes in the southeastern U.S. is $\sim$2.1 $\mu$g m$^{-3}$. This amount is substantially higher than represented in current regional models and accounts for 21% of World Health Organization PM$_{2.5}$ standard, indicating a significant contributor of environmental risk to the 77 million habitants in the southeastern U.S.

## 1 Introduction

Organic aerosol (OA) constitutes a substantial fraction of ambient fine particulate matter (PM) and has large impacts on air quality, climate change, and human health (Carslaw et al., 2013; Lelieveld et al., 2015). OA can be directly emitted from sources (primary OA, POA) or formed by the oxidation of volatile organic compounds (VOCs) (secondary OA, SOA). Global measurements revealed the dominance of SOA over POA in various atmospheric environments (Jimenez et al., 2009; Ng et al., 2010). The VOCs can be emitted from natural sources (i.e., biogenic) or human activities (i.e., anthropogenic). However, the relative contribution of biogenic and anthropogenic sources to SOA formation in the atmosphere is poorly constrained. This knowledge is critical for formulating effective pollution control strategies that aim at reducing ambient PM concentrations and accurately assessing the climate effects of OA (Hallquist et al., 2009). Biogenic VOCs such as monoterpenes (MT, $C_{10}H_{16}$) and sesquiterpenes (SQT, $C_{15}H_{24}$) are recognized as critical precursors of SOA (Tsigaridis et al., 2014; Hodzic et al., 2016; Pye et al., 2010). The predicted global SOA production from MT and SQT varies from 14 to 246 Tg yr$^{-1}$ (Spracklen et al., 2011; Pye et al., 2010). This large variation in model estimates arises from a number of factors (including uncertainty in SOA yield) and introduces significant uncertainties in estimating OA concentrations and its subsequent influences on climate and human exposure.

The large model uncertainties call for ambient observations to constrain model results. Isolating and measuring SOA production from specific sources are challenging because SOA is a complex mixture consisting of thousands of compounds and SOA evolves dynamically in the atmosphere. A widely used method to apportion OA into different characteristic sources is positive matrix factorization (PMF) analysis on the organic mass spectra measured by aerosol mass spectrometer (AMS) (Ulbrich et al., 2009; Jimenez et al., 2009; Ng et al., 2010). PMF-AMS analysis groups OA constituents with similar mass spectra and temporal variations into characteristic OA subtypes (i.e., factors). This analysis has revealed that concentration of oxygenated OA (OOA), which is a surrogate of SOA, is much greater than that of hydrocarbon-like OA (HOA), which is a surrogate of POA (Zhang et al., 2007). In many circumstances especially in warmer months, more than one SOA factor is resolved from PMF analysis, often including less-oxidized oxygenated OA (LO-OOA, also denoted as semi-volatile oxygenated organic aerosol in older studies) and more-oxidized oxygenated OA (MO-OOA, also denoted as low-volatility oxygenated organic aerosol in older studies). LO-OOA and MO-OOA are

differentiated by their degree of carbon oxidation. These two factors together account for more
than half of total submicron OA (Crippa et al., 2014; Xu et al., 2015a; Jimenez et al., 2009). Despite
of their large abundance, the sources of LO-OOA and MO-OOA are unclear and likely vary with
location and season. Early studies, primarily based on comparison of the mass spectra of OA
factors with those of laboratory-generated SOA, proposed that LO-OOA is freshly formed SOA
from various sources and evolves into MO-OOA with photochemical aging in the atmosphere
(Jimenez et al., 2009; Ng et al., 2010). Later, a number of possible sources have been proposed for
MO-OOA, including SOA from long-range transport (Hayes et al., 2013; Robinson et al., 2011b),
aged biomass burning OA (Bougiatioti et al., 2014; Grieshop et al., 2009), humic-like substances
(El Haddad et al., 2013), highly oxygenated molecules (HOMs) formed in the oxidation of
monoterpenes (Mutzel et al., 2015; Ehn et al., 2014), and aqueous phase processing (Xu et al.,
2016c). Regarding the sources of LO-OOA, Zotter et al. (2014) applied radiocarbon analysis and
showed that 68-75% of carbon in LO-OOA in California stems from fossil sources. In the
southeastern U.S., Xu et al. (2015a) suggested that the oxidation of biogenic β-pinene by nitrate
radicals ($NO_3$) contributes to LO-OOA, though this reaction alone cannot replicate the magnitude
of LO-OOA (Pye et al., 2015). These studies significantly advanced our knowledge of the sources
and evolution of ambient OA; however, uncertainties associated with the sources of these OA
factors still exist. As a result, atmospheric models typically use the lumped LO-OOA and MO-
OOA concentration to constrain simulated total SOA concentration (Spracklen et al., 2011;
Tsigaridis et al., 2014), which hinders our ability to diagnose the cause of discrepancies between
modeled and observed aerosol concentrations (Spracklen et al., 2011). Many sources of LO-OOA
and MO-OOA are proposed based on comparing the mass spectra between OA factors and
laboratory-generated SOA (Jimenez et al., 2009; Palm et al., 2018; Kiendler-Scharr et al., 2009).
However, the similarity between two mass spectra is a subjective determination. Further, the
subjectively-defined similarity cannot tell what is the fraction of SOA from a certain source
contributes to one OA factor. Overall, considering the large abundance of OOA subtypes and their
use as surrogates for ambient SOA, understanding the sources of compounds composing these two
OA subtypes is critical to constrain atmospheric models and the SOA budget.
In this study, we integrate lab-in-the-field experiments, extensive ambient ground
measurements, and state-of-the-art modeling to improve the understanding of the sources of OA
factors and better constrain the OA budget from MT and SQT. Based on lab-in-the-field
experiments, we provide objective evidence that newly formed SOA from α-pinene (an important
monoterpene) and β-caryophyllene (an important sesquiterpene) is dominantly apportioned to LO-
OOA in the southeastern U.S. In addition, we model the SOA concentration from the oxidation of
MT and SQT (denoted as $SOA_{MT+SQT}$) and show that $SOA_{MT+SQT}$ reasonably reproduces the
magnitude and diurnal variability of LO-OOA measured at multiple sites in the southeastern U.S.
Together with other evidence in the literature, we propose that LO-OOA can be used as a measure
of $SOA_{MT+SQT}$ in the southeastern U.S. Finally, we discuss how the lab-in-the-field approach
allows for the study of SOA formation under realistic atmospheric conditions, which bridges
laboratory studies and field measurements and provides a direct way to evaluate the atmospheric
relevancy of laboratory studies.

## 2 Method

### 2.1 Lab-in-the-field perturbation experiments

The perturbation experiments were performed in July-August 2016 on the rooftop of the
Environmental Science and Technology building on the Georgia Institute of Technology campus.
This measurement site is an urban site in Atlanta, Georgia. Multiple ambient field studies have
been performed at this site previously (Xu et al., 2015b; Hennigan et al., 2009; Verma et al., 2014).
A $2m^3$ Teflon chamber (cubic shape) (Fig. 1) was placed outdoor on the rooftop of the building.
The eight corners of the chamber were open (~2"×2") to the atmosphere to allow for continuous
exchange of air with the atmosphere. The perturbation procedure is briefly described below and
illustrated in Fig. A1. Firstly, we continuously flushed the chamber with ambient air using two
fans, which were placed at two corners of the chamber. During this flushing period, all instruments
sampled ambient air and were not connected to the chamber. The flushing period lasted at least 3
hours to ensure that the air composition in the chamber is the same as ambient composition.
Secondly, we stopped both fans and connected all instruments to chamber. Because of the
continued sampling by the instruments (~20 liter per minute) and the open corners of the chamber,
ambient air continuously entered the chamber, even though the two fans were turned off. Thirdly,
after sampling the chamber for about 30min, we injected a known amount of VOC (liquid) into
the chamber with a needle, where the liquid vaporized upon injection. We continuously monitored
the chamber composition for ~40 min after VOC injection. Lastly, we disconnected all instruments
from the chamber, sampled ambient air, and turned on two fans to flush the chamber to prepare
for the next perturbation experiment.
Each perturbation experiment can be divided into the following four periods: Amb_Bf
(30min ambient measurement period before sampling chamber), Chamber_Bf (from sampling
chamber to VOC injection, a period ~30min), Chamber_Af (from VOC injection to stop sampling
chamber, a period ~40min), and Amb_Af (30min ambient measurement period after sampling
chamber). We calculate the changes in the mass concentration of OA factors after perturbation
based on the difference between Chamber_Bf and Chamber_Af, after taking ambient variation into
account. The detailed procedure is presented in Appendix A. We develop a comprehensive set of
criteria to determine if the changes are statistically significant and if the changes are simply due to
ambient variations. The details of these criteria are also discussed in Appendix A.
We perturbed the chamber content by injecting one of the following VOCs: isoprene, α-
pinene, β-caryophyllene, $m$-xylene, or naphthalene, which are major biogenic or anthropogenic
emissions. We focused on α-pinene and β-caryophyllene, because they are widely studied in the
literature (Eddingsaas et al., 2012a; Kurtén et al., 2015; Tasoglou and Pandis, 2015; Ehn et al.,
2014; Pathak et al., 2007) and they have large abundances in their classes. For example, α-pinene
accounts for about half of monoterpenes emissions (Guenther et al., 2012) and β-caryophyllene is
one of the most abundant sesquiterpenes (Helmig et al., 2007). The injected VOC amounts were
carefully selected. If the injection amount is too large, it is not atmospherically relevant, produces
too much SOA, and will bias subsequent analysis. If the injection amount is too small, the produced
SOA would fall below the detection limit of the experimental approach. The VOC oxidation
occurred in ambient air (inside the chamber) and lasted ~40 min. The OA concentration in the
chamber after perturbation ranges from 4 to 16 $\mu g\ m^{-3}$, which is within the range of typical ambient
OA concentrations.
We note that several previous studies have used ambient air (Palm et al., 2017; Leungsakul
et al., 2005; Peng et al., 2016), but experimental approaches and purposes of previous studies are
different from this study. For example, In Leungsakul et al. (2005), the rural ambient air was used
to flush and clean the $270m^3$ outdoor chamber reactor. After the flushing, both VOCs and oxidants
were injected to produce SOA, the concentration of which were orders of magnitude higher than
atmospheric levels. In this study, we use ambient air with pre-existing OA in order to examine
which factor(s) the fresh α-pinene and β-caryophyllene SOA are apportioned into by PMF analysis.
We aim to produce SOA only from injected α-pinene or β-caryophyllene, so that an important
distinction between our study and pervious work is that we perturbed the ambient air by only VOCs,
not extra oxidant.

**2.2 Analytical instruments**

A suite of analytical instruments was deployed to characterize both the gas-phase and particle-
phase compositions. The particle-phase composition was monitored by a scanning mobility
particle sizer (SMPS, TSI) and a high resolution time-of-flight aerosol mass spectrometer (HR-
ToF-AMS, Aerodyne), which shared the same stainless steel sampling line. A diaphragm pump
(flow rate ~8 liter per minute) was connected to this sampling line, which increased the sampling
flow rate and reduced particle loss in the sampling line by reducing the residence time in the tubing.
The HR-ToF-AMS measures the chemical composition and size distribution of submicron non-
refractory species (NR-PM$_1$) with high temporal resolution. The instrument details about HR-ToF-
AMS have been extensively discussed in the literature (Canagaratna et al., 2007; DeCarlo et al.,
2006) and the operation of HR-ToF-AMS in this study is described in the section S2 of Supplement.
The gas-phase composition and oxidation products was monitored by an O$_3$ analyzer
(Teledyne T400, lower detectable limit 0.6ppb), an ultrasensitive chemiluminescence NO$_x$ monitor
(Teledyne 200EU, lower detectable limit 50ppt), and a high-resolution time-of-flight chemical
ionization mass spectrometer (HR-ToF-CIMS). The HR-ToF-CIMS with I$^-$ as regent ion can
measure a suite of oxygenated volatile organic compounds (oVOCs) at high frequency (1Hz).
Detailed working principles and sampling protocol can be found in Lee et al. (2014). The
concentrations of VOCs were not measured in this study. All gas-phase measurement instruments
shared the same Teflon sampling line. Similar to the particle sampling line, a diaphragm pump
(flow rate ~8 liter per minute) was connected to the gas sampling line to reduce the residence time
in the tubing.

**2.3 Positive Matrix Factorization (PMF) analysis**

PMF analysis has been widely used for aerosol source apportionment in the atmospheric chemistry
community (Jimenez et al., 2009; Crippa et al., 2014; Xu et al., 2015a; Ng et al., 2010; Ulbrich et
al., 2009; Beddows et al., 2015; Visser et al., 2015). PMF solves bilinear unmixing factor model
by minimizing the summed least squares errors of the fit weighted with the error estimates of each
measurement (Paatero and Tapper, 1994; Ulbrich et al., 2009). We utilized the PMF2 solver, which
does not require a priori information and reduces subjectivity. In this study, we performed PMF
analysis on the high-resolution mass spectra of organic aerosol (inorganic species are excluded) of
combined ambient and perturbation data in the one-month measurements. Considering that (1) the
perturbation data only account for ~10% of total data and (2) the OA concentration is similar
between the perturbation experiments and typical ambient measurements, the perturbation
experiments do not create a new factor that does not already exist in the ambient data. This is
desirable because it allows PMF analysis to apportion the newly formed OA in the perturbation
experiments into pre-existing OA factors in the atmosphere.

We resolved five OA factors, including hydrocarbon-like OA (HOA), cooking OA (COA),

isoprene-derived OA (isoprene-OA), less-oxidized oxygenated OA (LO-OOA), and more-
oxidized oxygenated OA (MO-OOA). The time series and mass spectra of OA factors are shown
in Fig. 2. The same 5 factors have been identified at the same measurement site and extensively
discussed in the literature (Xu et al., 2015a; Xu et al., 2015b; Xu et al., 2017). Below, we only
provide a brief description on these OA factors and more details are discussed in section S3 of
Supplement. The mass spectrum of HOA is dominated by hydrocarbon-like ions ($C_xH_y^+$ ions) and
HOA is a surrogate of primary OA from vehicle emissions (Zhang et al., 2011). For COA, its
concentration is higher at meal times and its mass spectrum is characterized by prominent signal
at ions $C_3H_5^+$ (*m/z* 41) and $C_4H_7^+$ (*m/z* 55), which likely arise from fatty acids (Huang et al., 2010;
Mohr et al., 2009; Allan et al., 2010). The mass spectrum of isoprene-OA is characterized by
prominent signal at ions $C_4H_5^+$ (*m/z* 53) and $C_5H_6O^+$ (*m/z* 82) and it is related to the reactive uptake
of isoprene oxidation products, isoprene epoxydiols (IEPOX) (Budisulistiorini et al., 2013; Hu et
al., 2015; Robinson et al., 2011a; Xu et al., 2015a). LO-OOA and MO-OOA are named based on
their differing carbon oxidation state, that is, from -0.70 to -0.34 for LO-OOA and from -0.18 to
0.71 for MO-OOA in the southeastern U.S. (Xu et al., 2015b). We performed 100 bootstrapping
runs to quantify the uncertainty of PMF results. As shown in Fig. S1, the statistical uncertainties
in the time series and mass spectra of 5 factors are small and the PMF results reported in this study
are robust.
**2.4 Details of multiple ambient sampling sites**

Measurements at multiple sites in the southeastern U.S. were performed as part of Southeastern Center for Air Pollution and Epidemiology study (SCAPE) and Southern Oxidant and Aerosol Study (SOAS) in 2012 and 2013. Detailed descriptions about these field studies have been discussed in the literature (Xu et al., 2015a; Xu et al., 2015b) and section S4 of Supplement. The sampling periods are shown in Table S1 and the sampling sites are briefly discussed below.

• Georgia Tech site (GT): This site is located on the rooftop of the Environmental Science and Technology building on the Georgia Institute of Technology (GT) campus, which is about 30-40m above the ground and 840m away from interstate I75/85. This is an urban site in Atlanta. This is also where the perturbation experiments in this study were conducted.

• Jefferson Street site (JST): This is a central SEARCH (SouthEastern Aerosol Research and Characterization) site, which is in Atlanta's urban area with a mixed commercial and residential neighborhood. It is about 2 km west of the GT site. The JST and GT sites are in the same grid cell in CMAQ.

• Yorkville site (YRK): This is a central SEARCH site located in a rural area in Georgia. This site is surrounded by agricultural land and forests and is at about 80 km northwest of JST site.

• Centreville site (CTR): This is a central SEARCH site in rural Alabama. The sampling site is surrounded by forests and away from large urban areas (55km SE and 84 km SW of Tuscaloosa and Birmingham, AL, respectively). The is the main ground site for the SOAS campaign.

**2.5 Laboratory chamber study on SOA formation from α-pinene**

To compare with results from the lab-in-the-field perturbation experiments, we performed laboratory experiments to study the SOA formation from α-pinene photooxidation under different $NO_x$ conditions in the Georgia Tech Environmental Chamber (GTEC) facility. The facility consists of two 12 $m^3$ indoor Teflon chambers, which are suspended inside a temperature-controlled enclosure and surrounded by black lights. The detailed description about chamber facility can be found in Boyd et al. (2015). The experimental procedures have been discussed in Tuet et al. (2017). In brief, the chambers were flushed with clean air prior to each experiment. Then, α-pinene and oxidant sources (i.e., $H_2O_2$, $NO_2$, or HONO) were injected into chamber. Once the concentrations of species stabilize, the black lights were turned on to initiate photooxidation. The experimental conditions are summarized in Table S2. Considering that the OA mass concentration affects the

partitioning of semi-volatile organic compounds (Odum et al., 1996) and hence affects the organic
mass spectra measured by AMS, we calculated the average mass spectra in these laboratory studies
by only using the data when the OA mass concentration is below 10 $\mu$g m$^{-3}$, which is similar to
that in our ambient perturbation experiments.
**2.6 Community Multiscale Air Quality (CMAQ) Model**
We used the Community Multiscale Air Quality (CMAQ) atmospheric chemical transport model
to simulate the pollutant concentrations across the southeastern U.S. CMAQ v5.2gamma was run
over the continental U.S. for time periods between May 2012 to July 2013 with 12km × 12km
horizontal resolution. We focus our analysis on the southeastern U.S., which comprises 11 states
(Arkansas, Alabama, Florida, Georgia, Kentucky, Louisiana, Mississippi, North Carolina, South
Carolina, Tennessee, and Virginia). The meteorological inputs were generated with version 3.8 of
the Weather Research and Forecasting model (WRF), Advanced Research WRF (ARW) core. We
also applied lightning assimilation to improve convective rainfall (Heath et al., 2016).
Anthropogenic emissions were based on the EPA (Environmental Protection Agency) NEI
(National Emission Inventory) 2011 v2. Biogenic emissions were predicted by the BEIS (Biogenic
Emission Inventory System) v3.6.1. The gas-phase chemistry was based on CB6r3 (Carbon Bond
v6.3).

We performed two simulations with different organic aerosol treatment. The "default

simulation" generally follows the scheme of Carlton et al. (2010), with the addition of IEPOX
SOA following Pye et al. (2013) and documented in Appel et al. (2017) (Fig. S2a). The traditional
two-product absorptive partitioning scheme (Odum et al., 1996) is used in "default simulation" to
describe SOA formation from monoterpenes using data from laboratory experiments by Griffin et
al. (1999). In the "updated simulation", we incorporate two recent findings. Firstly, we
implemented MT+NO$_3$ chemistry to explicitly account for the organic nitrate compounds that have
recently been shown to be a ubiquitous and important component of OA (Pye et al., 2015;
Kiendler-Scharr et al., 2016; Lee et al., 2016; Ng et al., 2017). We follow the scheme described in
Pye et al. (2015) to represent the formation and partition of organic nitrates from monoterpenes
via multiple reaction pathways (i.e., oxidation by NO$_3$ and oxidation by OH/O$_3$ followed by
RO$_2$+NO). Secondly, we improved the parameterization of SOA formation from MT+O$_3$/OH
based on a recent study by Saha and Grieshop (2016), who applied a dual-thermodenuder system
to study the α-pinene ozonolysis SOA. The authors extracted parameters (i.e., SOA yields and
enthalpies of evaporation) by using an evaporation-kinetics model and volatility basis set (VBS).
The SOA yields in Saha and Grieshop (2016) are consistent with recent findings on the formation
of HOMs (Ehn et al., 2014; Zhang et al., 2015) and help to explain the observed slow evaporation
of α-pinene SOA (Vaden et al., 2011). In the updated simulation, we use the VBS framework with
parameters derived from Saha and Grieshop (2016). The new parameterization allows for
enthalpies of vaporization that are more consistent with species of the specified volatility. The
properties of the volatility bins in the VBS framework are listed in Table S3. A schematic of SOA
treatment in "updated simulation" is shown in Fig. S2b. In the following discussions, we focus on
the results from "updated simulation". The comparison between "default simulation" and "updated
simulation" can be found in the section S5 of Supplement.
**3 Results and Discussions**
**3.1 α-pinene perturbation experiments**
A total of 19 α-pinene perturbation experiments were performed at different times of the day (i.e.,
from 9am to 9pm) to probe a wide range of reaction conditions. The injection time and
concentrations of $O_3$ and $NO_x$ during α-pinene perturbation experiments are summarized in Table
S4. Based on the chamber volume and injected liquid α-pinene volume, initially ~14 ppb α-pinene
is injected into chamber. Due to lack of VOC measurements, we build a box model to simulate the
fate of α-pinene in the chamber (section S6 of Supplement). We estimate that only a small fraction
(2-5ppb) of α-pinene is reacted in the chamber and most of α-pinene is carried out of the chamber
due to dilution with ambient air.

Fig. 3 shows the time series of OA factors in a typical α-pinene perturbation experiment.
An evident burst and increase of LO-OOA after α-pinene injection occurs. This provides direct
evidence that freshly formed α-pinene SOA contributes to LO-OOA. About 15 min after α-pinene
injection, LO-OOA concentration starts to decrease, as ambient air continuously flows into the
chamber and dilutes the concentration of LO-OOA (section S6 of Supplement). As shown in Fig.
S3, the major known gas-phase oxidation products of α-pinene measured by HR-ToF-CIMS
(Eddingsaas et al., 2012b; Lee et al., 2016; Yu et al., 1999) show an immediate increase after α-
pinene injection. This verifies the rapid oxidation of α-pinene in the chamber.

Fig. 4a shows the perturbation-induced changes in the concentrations of OA factors for all

α-pinene experiments. Out of 19 experiments, the LO-OOA concentration is enhanced in 14
experiments. Also, among all OA factors, LO-OOA shows the largest enhancement. This directly
supports that freshly formed α-pinene SOA contributes to LO-OOA. The enhancement in LO-
OOA concentration differs between experiments, mainly because the perturbations were
performed at different times of day (i.e., from 9am to 9pm) and with different reaction variables
(i.e., temperature, relative humidity, oxidants concentrations, $NO_x$, etc). Despite the large
difference in reaction conditions, we note that both LO-OOA enhancement amount and LO-OOA
formation rate (i.e., slope of LO-OOA increase) correlate positively with ozone concentration (Fig.
5). This correlation suggests that the concentration of oxidants, both ozone and hydroxy radical
(OH, which is not measured in this study but is known to positively correlate with ozone in the
atmosphere), plays a more controlling role in the amount of OA formed in α-pinene experiment
than other reaction variables do. This is likely because higher oxidant concentrations lead to more
α-pinene consumption and hence more OA production with the same reaction time.
MO-OOA only increases in 1 out of 19 α-pinene experiments. The highly oxygenated
molecules (HOMs), which are rapidly produced from the oxidation of α-pinene, are a hypothesized
source of MO-OOA, because of the high O:C ratio of HOMs (Ehn et al., 2014; Mutzel et al., 2015).
However, HOMs are first generation monoterpene products co-formed with semivolatile SOA
species, and the lack of enhancement in MO-OOA suggests that the HOMs are unlikely
contributors to MO-OOA. We cannot rule out the possibilities that HOMs are not formed under
our experimental conditions, and future studies on the simultaneous verification of HOMs
formation and apportion of HOMs by PMF analysis are warranted.
Isoprene-derived OA (isoprene-OA) increases in 7 out of 19 α-pinene experiments. This
increase is surprising because the isoprene-OA factor (also referred to as "IEPOX-OA" in some
studies) is typically interpreted as SOA from the reactive uptake of IEPOX, but our results suggest
that the isoprene-OA factor could have interferences from α-pinene SOA. The isoprene-OA
enhancement is due to interference from newly formed α-pinene SOA, rather than that the injected
α-pinene affecting the oxidation of pre-existing isoprene or affecting the gas/particle partitioning
of pre-existing semi-volatile species in the chamber, because of the following reasons. Firstly,
based on $I^-$ HR-ToF-CIMS measurement, the concentration of isoprene oxidation products, such
as IEPOX+ISOPOOH ($C_5H_{10}O_3 \cdot I^-$) and isoprene hydroxyl nitrates ($C_5H_9NO_4 \cdot I^-$), did not change

after α-pinene injection (Fig. S3b). In addition, after injecting α-pinene, the SOA concentration increases less than 4 μg m$^{-3}$, which does not substantially perturb the gas/particle partition of pre-existing semi-volatile species. Finally, the time series of isoprene-OA and LO-OOA in the same α-pinene perturbation experiment is strongly correlated (Fig. S4a). It is well studied that isoprene produces SOA slower than α-pinene, as isoprene SOA involves higher-generation products. If the enhancement in isoprene-OA factor is due to isoprene oxidation, the enhancement of isoprene-OA is expected to occur later than the enhancement of LO-OOA, but it is not observed in the experiments. Thus, the strong correlation between isoprene-OA and LO-OOA in the same α-pinene perturbation experiment serves as another evidence that the enhancement in isoprene-OA factor is due to interference from newly formed α-pinene SOA, rather than oxidation of isoprene after injecting α-pinene.

The interference of α-pinene SOA on isoprene-OA factor helps to address some uncertainties regarding the isoprene-OA factor in the literature. For example, Liu et al. (2015) compared the mass spectrum of laboratory-derived IEPOX SOA with isoprene-OA factors at some sites. The authors observed stronger correlation for isoprene-OA factors resolved at Borneo (Robinson et al., 2011a) and Amazon (Chen et al., 2015), and weaker correlation at Atlanta, U.S. (Budisulistiorini et al., 2013) and Ontario, Canada (Slowik et al., 2011). As another example, the fraction of measured total IEPOX SOA molecular tracers in isoprene-OA factor highly varies with location, ranging from 26% at Look Rock, TN (Budisulistiorini et al., 2015) to 78% at Centreville, AL (Hu et al., 2015). To address the uncertainties in above two examples, one possible reason is that the isoprene-OA factors resolved at different sites are not purely from IEPOX uptake. Isoprene-OA factors likely have interference from monoterpenes SOA or other sources, but the interference magnitude varies with locations.

While the perturbation experiments clearly point out the possibility that isoprene-OA factor could have interference from α-pinene SOA, two caveats should be kept in mind. First, in this study, the enhancement magnitude of isoprene-OA is ~20% of LO-OOA enhancement (Fig. S5a), but the interference magnitude would vary with locations and seasons. Second, the perturbation experiments simulate a period with increasing α-pinene SOA concentration. The applicability of the conclusions drawn from this specific scenario to general atmosphere with more dynamic variations of OA sources warrants further exploration.

Primary OA factors, i.e., HOA and COA, only show slight increases in 1 or 2 α-pinene
experiments, indicating a lack of interference from α-pinene SOA in these factors.
**3.2 β-caryophyllene perturbation experiments**
A total of 6 β-caryophyllene perturbation experiments were performed. Initially ~10 ppb β-
caryophyllene is injected into the chamber. The concentrations of $O_3$ and $NO_x$ during β-
caryophyllene perturbation experiments are summarized in Table S4. In all β-caryophyllene
perturbation experiments, LO-OOA also shows a significant enhancement (Fig. 4b). This clearly
shows that the freshly formed SOA from β-caryophyllene oxidation can be another source of LO-
OOA. In addition to LO-OOA, COA shows an unexpected increase in 5 out of 6 β-caryophyllene
experiments. We have ample evidence that the COA factor at the measurement site has
contributions from cooking activities. Firstly, the diurnal variation of COA peaks during meal
times (Fig. S6a). Additionally, the COA concentration shows clear increase on football days,
consistent with barbecue activities on campus and close to the measurement site. Finally, the COA
concentration is enhanced on the days right before the start of a new semester when there are many
fraternity/sorority rush events (i.e., barbecue activities) on campus (Fig. S6b and S6c). However,
the COA enhancement in β-caryophyllene experiments underscores the fact that COA may not be
purely from cooking activities in areas with large biogenic emissions.
**3.3 Perturbation experiments with other VOCs**
In addition to α-pinene and β-caryophyllene, we also performed a few perturbation experiments
by injecting isoprene, *m*-xylene, or naphthalene. However, the SOA formation from these VOCs
is not detectable. This is mainly due to either lower SOA yields (of isoprene) or slower oxidation
rates (of *m*-xylene and naphthalene) compared to α-pinene and β-caryophyllene, which are
discussed in section S6 of Supplement.
We have also performed four perturbation experiments by injecting acidic sulfate particles
to probe reactive uptake of IEPOX. We observed enhancement in isoprene-OA concentration after
the injection of sulfate particles. The detailed results are included in Appendix B.
**3.4 Compare conclusions from lab-in-the-field perturbation experimental approach vs. mass**
**spectra comparison approach**
Based on the lab-in-the-field perturbation experiments, we show that fresh SOA from α-pinene
and β-caryophyllene oxidations are mainly apportioned into LO-OOA. This finding is consistent
with previous studies which concluded that LO-OOA (also denoted as semi-volatile oxygenated
organic aerosol, SV-OOA, in older studies) represents freshly formed SOA. The conclusion from
previous studies is mainly based on mass spectra comparison approach, that is, the mass spectra
of laboratory-generated fresh SOA from various sources are similar to that of LO-OOA (Jimenez
et al., 2009; Ng et al., 2010; Marcolli et al., 2006; Kiendler-Scharr et al., 2009). While we
acknowledge that the mass spectra comparison approach largely improves our understanding of
OA factors, we believe that the perturbation experimental approach provides more objective and
quantitative conclusions by addressing some limitations of the mass spectra comparison approach.
The mass spectra comparison approach has the following limitations. Firstly, the similarity
between two mass spectra is a subjective determination. In other words, what correlation
coefficient (R) value implies SOA from a certain source contributes to a specific OA factor? For
example, the R values between laboratory generated α-pinene SOA (using HONO as OH source)
with LO-OOA, isoprene-OA, and MO-OOA in this study are 0.96, 0.88, and 0.81, respectively.
Using these R values to imply whether α-pinene SOA contributes to a certain OA factor or not is
subjective. As another example, Jimenez et al. (2009) showed that the mass spectrum of α-pinene
SOA becomes more similar to that of MO-OOA than that of LO-OOA with photochemical aging.
The ability to determine when and how much α-pinene SOA is apportioned to MO-OOA based on
an R value is subjective. Secondly, the conclusions from mass spectra comparison approach are
qualitative. Even if the mass spectrum of α-pinene SOA is the most similar to LO-OOA, this
similarity does not guarantee that all α-pinene SOA is apportioned into LO-OOA and this
similarity does not provide information regarding what fraction of α-pinene SOA is apportioned
into LO-OOA.

The perturbation experiments could address the limitations of mass spectra comparison
approach and provide more objective and quantitative conclusions. Firstly, the perturbation
experiments simulate a short period of time with increasing α-pinene SOA concentration. We
perform PMF analysis on the combined ambient data and perturbation data. PMF analysis does
not distinguish SOA from natural α-pinene vs. from injected α-pinene, so that PMF analysis can
objectively apportion α-pinene SOA into factors. Thus, the conclusions from the perturbation
experiments are directly drawn without any subjective judgement on the similarity in mass spectra.
Secondly, using the perturbation data, we attempt to quantify the fraction of fresh α-pinene SOA
that is apportioned into different factors (i.e., ~80% into LO-OOA, ~20% into isoprene-OA, 0%
into MO-OOA, COA, and HOA). Although further studies are required to extrapolate the
conclusions from perturbation experiments to real atmosphere, a similar quantitative
understanding cannot be obtained from simple mass spectra comparison approach. Thirdly, the
perturbation experiments have the potential to utilize subtle differences across the entire the mass
spectrum to evaluate the sources of OA factors. Based on previous laboratory study, the mass
spectrum of α-pinene SOA is highly correlated (R = 0.97) with that of β-caryophyllene SOA
(Bahreini et al., 2005). Using a mass spectra comparison approach would suggest that these mass
spectra are too similar to be differentiated by PMF analysis. However, perturbation experiments
show different behaviors of α-pinene SOA and β-caryophyllene SOA. That is, a fraction of the
fresh β-caryophyllene SOA is apportioned into COA factor, but similar behavior is not observed
for α-pinene SOA. The different behaviors are likely due to the subtle differences in their mass
spectra. For example, $f_{55}$ (i.e., the ratio of $m/z$ 55 to total signal in the mass spectrum) is typically
higher in β-caryophyllene SOA than α-pinene SOA (Bahreini et al., 2005; Tasoglou and Pandis,
2015), and the mass spectrum of COA is characterized by prominent signal at $m/z$ 55 (Fig. 2).
Overall, the perturbation experiments provide more objective and quantitative insights into the
sources of OA factors than traditional mass spectra comparison approach.
**3.5 LO-OOA as a surrogate of SOA$_{MT+SQT}$ in the Southeastern U.S.**
We propose that the major source of LO-OOA in the southeastern U.S. is the fresh SOA from
oxidation of MT and SQT by various oxidants (O$_3$, OH, and NO$_3$), based on the following piece
of evidence. First, the southeastern U.S. is characterized by large biogenic emissions, including
monoterpenes and sesquiterpenes (Guenther et al., 2012). Second, the majority of carbon in SOA
is modern in the southeastern U.S. Weber et al. (2007) measured that the biogenic fraction of SOA
is roughly 70-80% at two urban sites in Georgia that were also used in our study. We note that
measurements in Weber et al. (2007) were performed in 2004 and the biogenic fraction of SOA is
expected to be higher in 2016 than 2004, as a result of reductions in anthropogenic emissions
(Blanchard et al., 2010). Third, previous studies suggest that the oxidation of β-pinene (another
important monoterpene) by nitrate radicals (NO$_3$) contributes to LO-OOA in the southeastern U.S.
(Boyd et al., 2015; Xu et al., 2015a), though this reaction alone cannot replicate the magnitude of
LO-OOA (Pye et al., 2015). Fourth, the mass spectra of LO-OOA are almost identical (i.e., R
ranges from 0.95 to 0.99 in Fig. S7) across all the seven datasets in our study. In addition, LO-
OOA across all datasets also shares the same diurnal trends (Xu et al., 2015a). The similarity in
LO-OOA features suggests that LO-OOA generally share similar sources across multiple sites and
in different seasons in the southeastern U.S. Fifth, the lab-in-the-field perturbation experiments
provide objective evidence that the majority of freshly formed SOA from the oxidation of MT and
SQT contributes to LO-OOA. Sixthly, using the updated CMAQ model (i.e., explicit organic
nitrates and Saha and Grieshop (2016) VBS for MT+$O_3$/OH SOA), we found that the simulated
$SOA_{MT+SQT}$ reasonably reproduces both the magnitude and diurnal variability of LO-OOA for all
sites (Fig. 6a). The model bias is within ~20% for most sites, except for Centreville, Alabama (i.e.,
43% for CTR_June dataset). Fig. 6b present maps of ground-level $SOA_{MT+SQT}$ concentration
corresponding to the time periods of observational data, and the $SOA_{MT+SQT}$ concentration is
substantially higher in the southeast than other U.S. regions. While, the $SOA_{MT+SQT}$ is present
throughout the year, it reaches the largest concentration in summer. The spatial and seasonal
variation of $SOA_{MT+SQT}$ concentration is consistent with MT and SQT emissions (Guenther et al.,
2012). The consistency between modeled $SOA_{MT+SQT}$ and measured LO-OOA at multiple sites
and in different seasons builds confidence in our hypothesis that LO-OOA largely arises from the
oxidation of MT and SQT in the southeastern U.S.

We note that we do not conclude that LO-OOA arises exclusively from MT and SQT. SOA

from other precursors or other pathways may contribute to LO-OOA, but the related contributions
are expected to be much smaller than MT and SQT in the southeastern U.S. Firstly, the
contributions of anthropogenic SOA to LO-OOA are likely small. The emissions of anthropogenic
VOCs are much weaker than that of biogenic VOCs in the southeastern U.S. (Goldstein et al.,
2009). We modeled that the concentration of anthropogenic SOA is on the order of 0.1 $\mu g\ m^{-3}$ for
our datasets (Fig. S8). Even if we double the SOA yields of anthropogenic VOCs to account for
the potential vapor wall loss in laboratory studies (Zhang et al., 2014), the concentration of SOA
from anthropogenic VOCs oxidation is still negligible compared to $SOA_{MT+SQT}$. The low modeled
concentration of anthropogenic SOA is consistent with Zhang et al. (2018), who showed that the
measured tracers of anthropogenic SOA only account for 2% of total OA in Centreville, AL.
Secondly, other reaction pathways, like aqueous-phase chemistry or some unexplored reaction,
may contribute to LO-OOA. However, the consistency between modeled $SOA_{MT+SQT}$ and LO-
OOA suggests that LO-OOA can be reasonably represented by a model based on current
knowledge and it is not necessary to invoke any unexplored mechanisms. In addition, SOA
produced from aqueous-phase chemistry is generally highly oxidized (Lee et al., 2011) and may
be apportioned into MO-OOA, instead of LO-OOA. A recent study by Xu et al. (2016c) suggests
that aqueous-phase SOA is a major source of MO-OOA in China.
We limit our hypothesis that major source of LO-OOA is the oxidation of MT and SQT to
the southeastern U.S. There is clear evidence that LO-OOA factor represents different sources at
different locations. For example, radiocarbon analysis shows that 68-75% of carbon in LO-OOA
in California stems from fossil sources (Hayes et al., 2013; Zotter et al., 2014), suggesting the
contribution from anthropogenic SOA to LO-OOA. Also, in the wintertime of many locations,
LO-OOA and MO-OOA are not separated and a single OOA factor is resolved (Xu et al., 2016b;
Lanz et al., 2008).
**3.6 Connection between laboratory and field studies**
Due to the difficulties associated with accurately measuring complex chemical processes in the
atmosphere, laboratory studies have been an integral part in our understanding of atmospheric
chemistry (Burkholder et al., 2017). However, the representativeness of laboratory studies under
simplified conditions with respect to the complex atmosphere is difficult to evaluate. One unique
feature of our lab-in-the-field approach is that the VOC oxidation and SOA formation proceed
under realistic atmospheric conditions. Taking advantage of this, we provide a direct link between
laboratory studies and ambient observations. Previous laboratory studies have shown that NO can
affect SOA composition by influencing the fate of organic peroxy radical ($RO_2$, a critical radical
intermediate formed from VOC oxidation) (Kroll and Seinfeld, 2008; Sarrafzadeh et al., 2016;
Presto et al., 2005). To evaluate the representativeness of laboratory studies and investigate the
effects of NO on SOA composition, in Fig. 7, we compare the chemical composition of α-pinene
SOA formed in laboratory studies under different NO conditions (denoted as $SOA_{lab}$) with those
in α-pinene ambient perturbation experiments (denoted as $SOA_{ambient}$). The degree of similarity in
OA mass spectra (i.e., evaluated by the correlation coefficient) between laboratory α-pinene SOA
generated under NO-free condition (i.e., denoted as $SOA_{lab,NO-free}$, using $H_2O_2$ photolysis as oxidant
source) and $SOA_{ambient}$ shows a strong dependence on ambient NO concentration, under which the
$SOA_{ambient}$ is formed. The degree of similarity in mass spectra decreases rapidly when ambient NO
increases from 0.1 to 0.2ppb, and then reaches a plateau at ~0.3ppb NO. The opposite trend is
observed when laboratory α-pinene SOA generated in the presence of high NO concentrations (i.e.,
denoted as $SOA_{lab,high-NO}$, using the photolysis of $NO_2$ or nitrous acid as oxidant source) are
compared with $SOA_{ambient}$. These observations show the transition of $RO_2$ fate as a function of NO
under ambient conditions. For the perturbation experiments performed when ambient NO is below
~0.1ppb, the mass spectra of $SOA_{ambient}$ are similar to $SOA_{lab,NO-free}$, consistent with that $RO_2$
mainly reacts with hydroperoxyl ($HO_2$) or isomerizes. In contrast, for the perturbation experiments
performed when ambient NO is above ~0.3ppb, the mass spectra of $SOA_{ambient}$ are similar to
$SOA_{lab,high-NO}$, consistent with that the $RO_2$ fate is dominated by NO. This NO level (~0.3ppb) is
consistent with the NO level required to dominate the fate of $RO_2$ in the atmosphere, as calculated
by using previously measured $HO_2$ and kinetic rate constants (section S8 of Supplement). These
observations also illustrate that the SOA composition from laboratory studies can be representative
of atmosphere. We note that the mass spectra of $SOA_{ambient}$ are generally more similar with that of
laboratory SOA generated using $NO_2$ photolysis as oxidant source than using nitrous acid
photolysis. This suggests that laboratory experiments using $NO_2$ photolysis as oxidant source
better represent ambient high NO oxidation conditions in the southeastern U.S. than experiments
using nitrous acid do. Possible explanations are discussed in section S7 of Supplement. This
finding provides new insights into designing future laboratory experiments to better mimic the
oxidations in ambient environments.

## 4 Implications

In this study, we performed lab-in-the-field perturbation experiments and provided objective
evidence that the majority of fresh SOA from the oxidation of MT and SQT contributes to LO-
OOA. Based on weight of evidence, we propose that LO-OOA can be used as a surrogate of fresh
SOA from MT and SQT in the southeastern U.S. We showed that modeled $SOA_{MT+SQT}$ could
reasonably reproduce both the magnitude and diurnal variability of LO-OOA at different sites and
in different seasons. Based on the model simulation, we estimate that the annual concentration of
$SOA_{MT+SQT}$ in $PM_{2.5}$ in the southeastern U.S. is ~2.1 µg m$^{-3}$ (i.e., average concentration over the
six sampling periods and over the southeastern U.S. in the updated simulation). This accounts for
21% of World Health Organization $PM_{2.5}$ guideline (i.e., 10 µg m$^{-3}$ annual mean) and indicates a
significant contributor of environmental risk to the 77 million habitants in the southeastern U.S.
Also, the estimated abundance of $SOA_{MT+SQT}$ is substantially larger than represented in current
models (Lane et al., 2008; Zheng et al., 2015), but in line with the conclusion from Zhang et al.
(2018). Zhang et al. (2018) used a different methodology, characterization of molecular tracers of
MT SOA at Centreville, AL (a site included in our study as well), to conclude that monoterpenes
are the largest source of summertime organic aerosol in the southeastern United States. The
oxidation of MT and SQT is likely an under-estimated contributor to PM in the present day and
perhaps during the pre-industrial period, which determines the baseline state of atmosphere and
the estimate of climate forcing by anthropogenic emissions (Carslaw et al., 2013). Models need to
improve the description of the MT and SQT oxidation to reduce the uncertainties in estimated OA
budget and subsequent climate forcing.

Using LO-OOA as a surrogate of $SOA_{MT+SQT}$ in the southeastern U.S., our ambient ground

measurements suggest that at least 19-34% of OA in the southeastern U.S. is from the oxidation
of biogenic monoterpenes and sesquiterpenes (Xu et al., 2015a). The fraction of biogenic OA in
the southeastern U.S. is even larger if we consider that isoprene-OA could account for 21-36% of
OA in summer (albeit potential interferences of SOA from monoterpenes oxidation) and that MO-
OOA (24-49% of OA) likely contains SOA from long-term photochemical oxidation of biogenic
VOCs. The dominant biogenic origin of SOA poses a challenge to control its burden in the
southeastern U.S., if the roles of anthropogenic oxidants and other controlling factors are not
recognized. Previous studies have shown that the SOA formation from biogenic VOCs can be
mediated by anthropogenic emissions, such as nitrogen oxides and sulfur dioxide (Hoyle et al.,
2011; Goldstein et al., 2009; Surratt et al., 2010; Rollins et al., 2012; Xu et al., 2015a). Thus,
regulating anthropogenic emissions could help reduce SOA concentration (Lane et al., 2008; Pye
et al., 2015; Zheng et al., 2015). For example, as observed in our ambient perturbation experiments,
one controlling parameter of α-pinene SOA formation is the concentration of atmospheric oxidants
($O_3$, OH, and $NO_3$), which are known to strongly depend on $NO_x$ concentration. As it has been
shown that anthropogenic emissions exert complex and non-linear influences on biogenic SOA
formation (Zheng et al., 2015), the effectiveness of regulating anthropogenic emissions on
biogenic SOA burden requires careful investigations.

The lab-in-the-field perturbation experiments provide insights into the OA factors. This

experimental approach can be easily adapted. Future experiments conducted under various
ambient environments and with diverse SOA precursors would facilitate the understanding of OA
factors in other regions of the world.

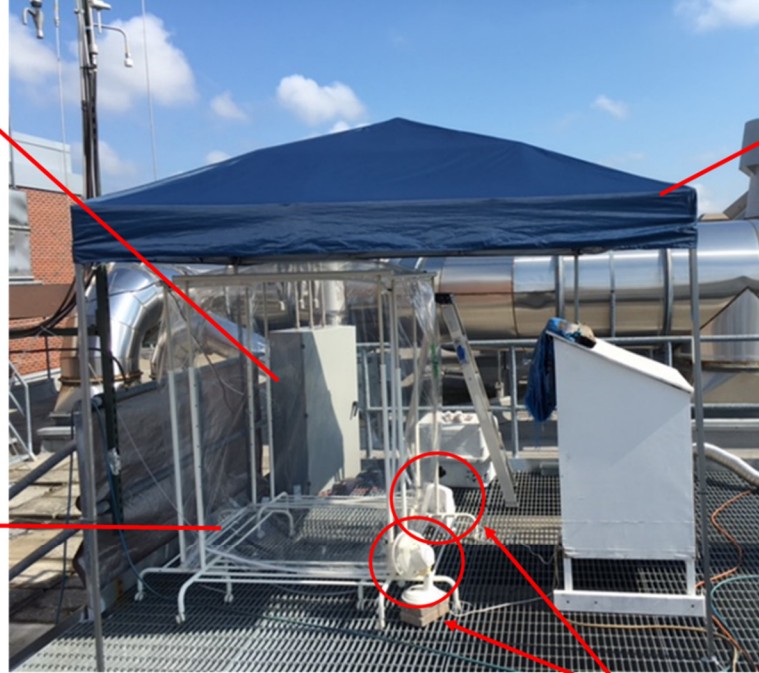

Instruments are located inside the lab (not shown). Particle phase: AMS, SMPS Gas phase: CIMS, $O_3$, $NO_x$

The tent is removed during the perturbation experiments.

The chamber volume is ~2 $m^3$. Eight corners are open.

Two fans are used to flush the chamber. The fans are turned off after VOC injection. After turning off the fans, flow rate of air going into the chamber is equal to the instruments pulling flow rate.


Fig. 1. The instrument setup for ambient perturbation experiments.

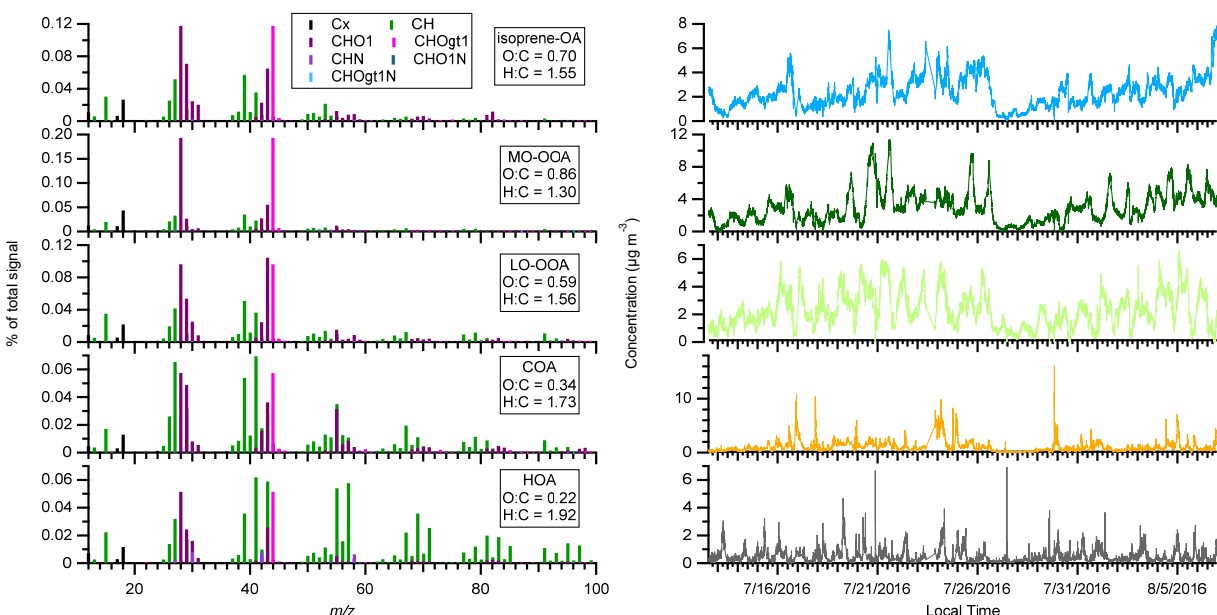


Fig. 2. The mass spectra and time series of OA factors in perturbation study. The time series
includes both the ambient data and perturbation experiments data.


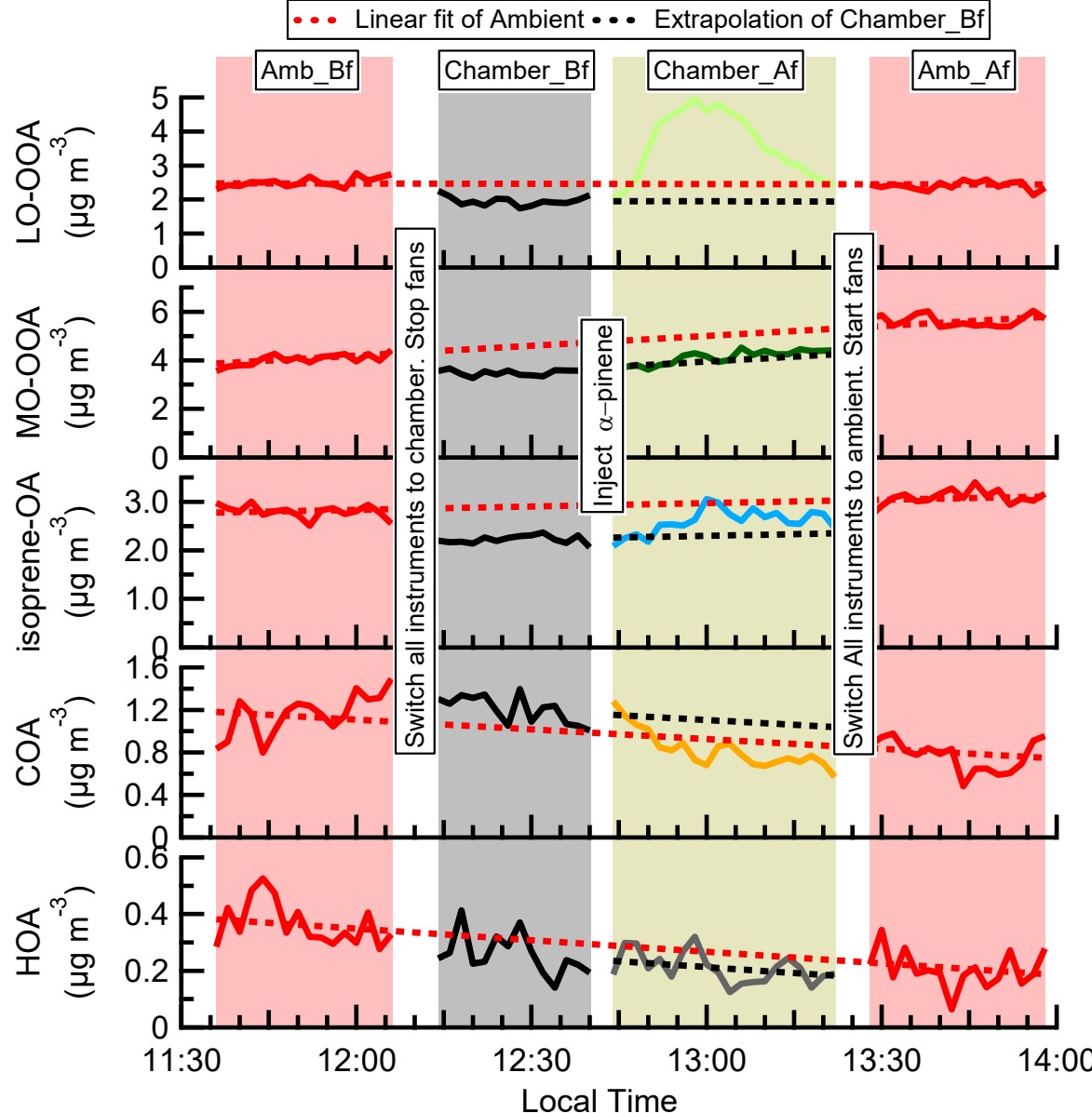


Fig. 3. The time series of OA factors in an α-pinene perturbation experiment (Expt ID: ap_0801_1).
Each perturbation experiment includes four periods: Amb_Bf (~30min), Chamber_Bf (~30min),
Chamber_Af (~40min), and Amb_Af (~40min). "Amb" and "Chamber" represent that instruments
are sampling ambient and chamber, respectively. "Bf" and "Af" stand for before and after
perturbation, respectively. The solid lines are measurement data. The dashed red lines are the linear
fits of ambient data (i.e., combined Amb_Bf and Amb_Af). The slopes are used to extrapolate
Chamber_Bf data to Chamber_Af period (i.e., dashed black lines). The validity of the linearity
assumption is discussed in Appendix A. The difference between measurements (i.e., solid lines)
and extrapolated Chamber_Bf (i.e., dashed black lines) represents the change caused by
perturbation.

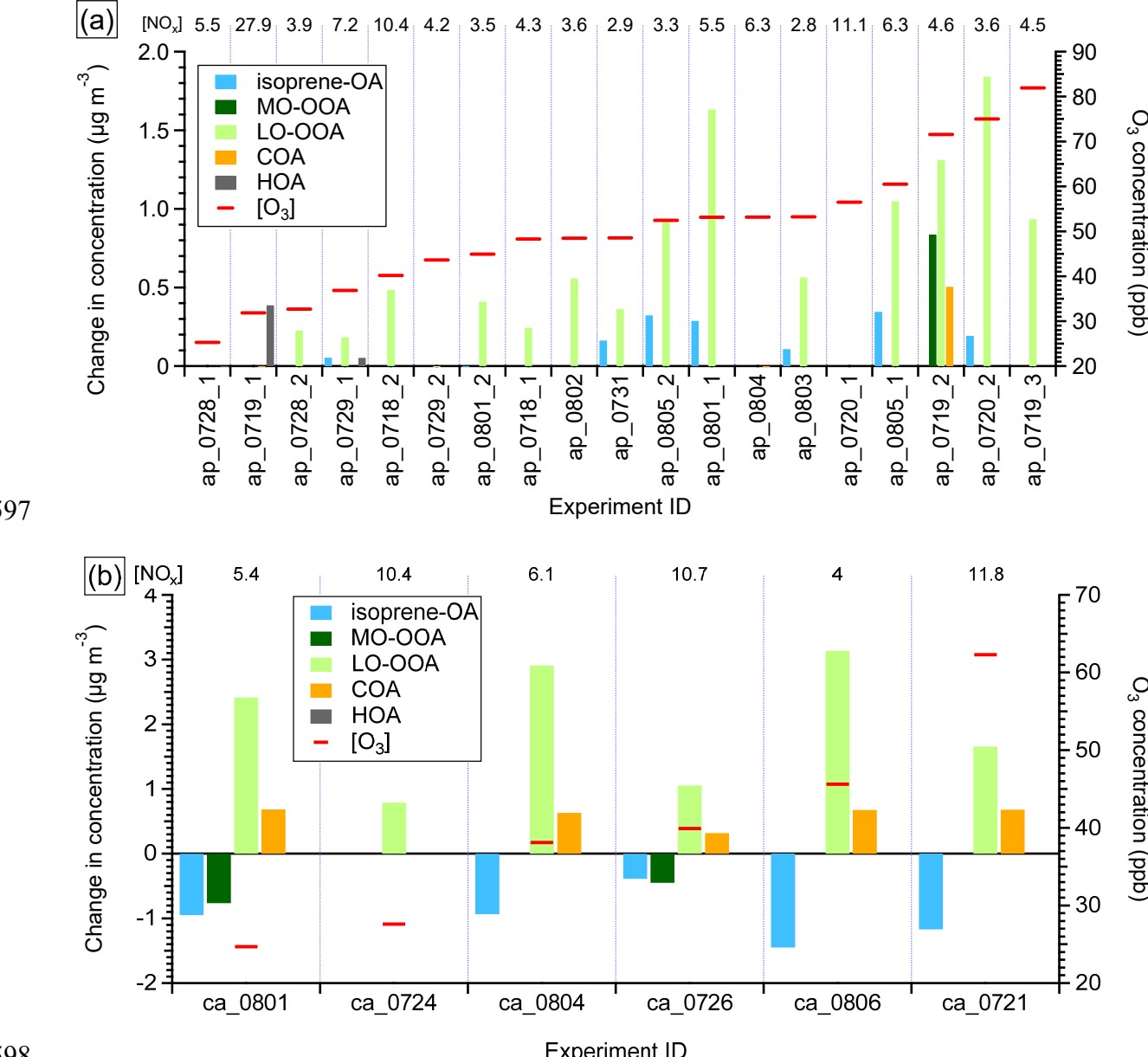

Fig. 4. The statistically significant changes in the concentrations of OA factors after perturbation by (a) α-pinene and (b) β-caryophyllene. The experiments are sorted by average [O$_3$] during Chamber_Af. The average [NO$_x$] during Chamber_Af are shown on top of the figure. The changes in concentration are the differences between measurements during Chamber_Af and extrapolated Chamber_Bf (Appendix A). A set of criteria are developed to evaluate if the changes are statistically significant and if the changes are due to ambient variation (Appendix A). Isoprene-OA decreases after β-caryophyllene injection. The reason for this decrease is unclear, but likely due to the limitations of PMF analysis, which assumes constant mass spectra of OA factors over time (section S3 of Supplement).

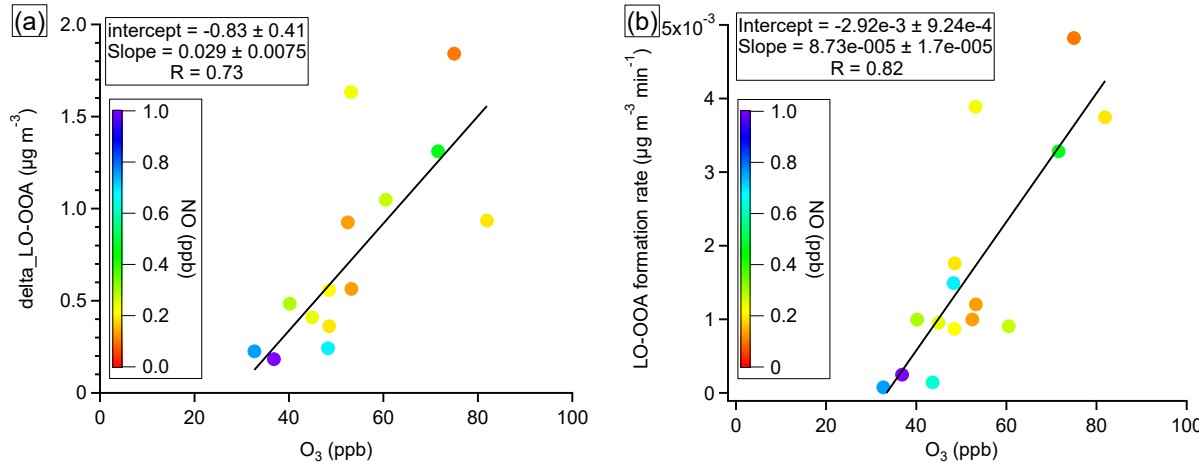


Fig. 5. Observations of trends in (a) LO-OOA enhancement amount and (b) LO-OOA formation
rate with O₃ concentration in α-pinene perturbation experiments. The data points are colored by
average NO concentration during Chamber_Af period. The slopes, intercepts, and correlation
coefficients (R) are obtained by least square fit.

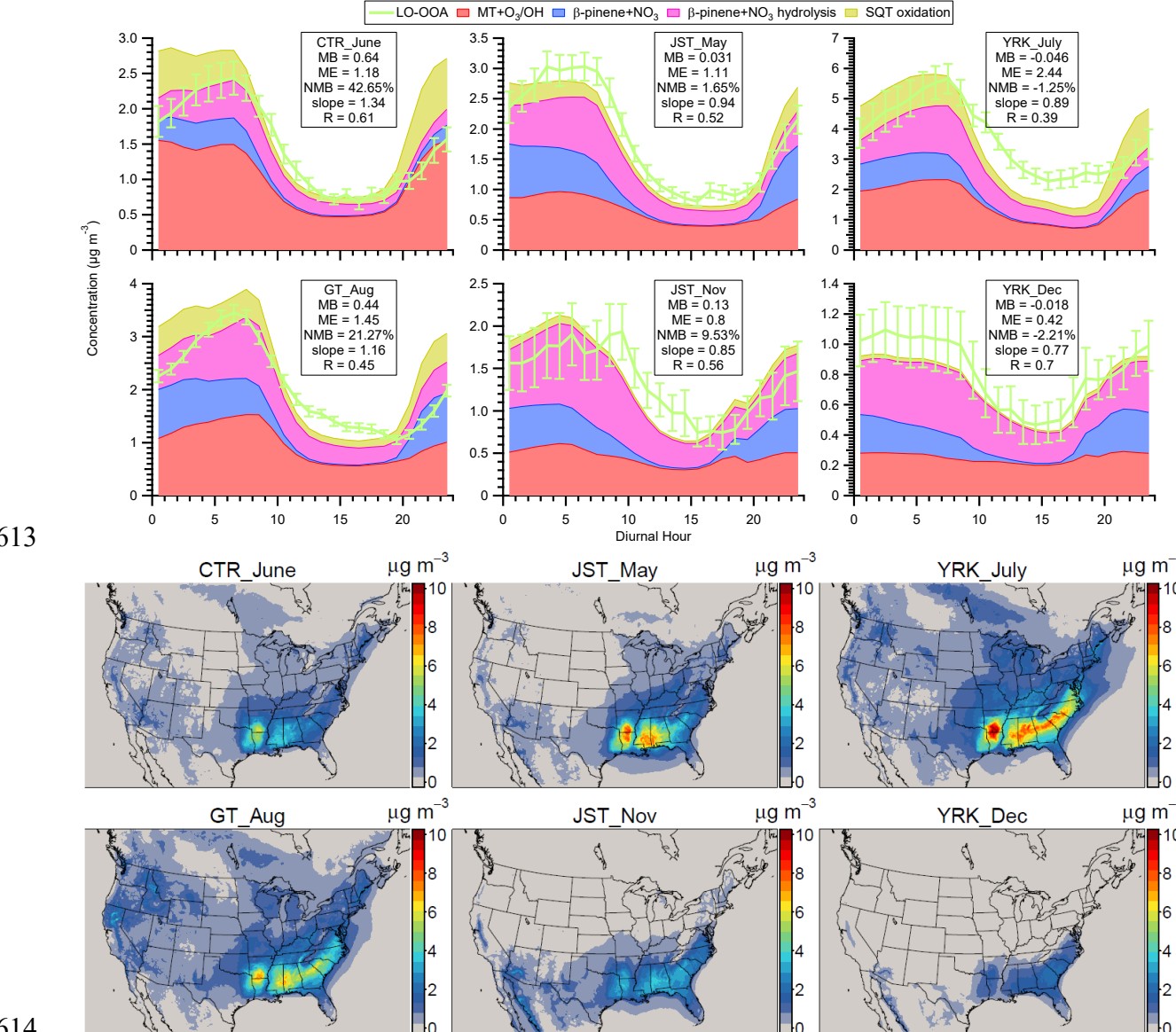



Fig. 6. (a) top panel: the diurnal trends of LO-OOA and modeled SOA from monoterpenes and sesquiterpenes (SOA$_{MT+SQT}$) at different sampling sites in the southeastern U.S. (b) bottom panel: maps of modeled ground-level SOA$_{MT+SQT}$ concentration coinciding with the time periods of intensive ambient sampling. Model results shown here are from the updated simulation. Abbreviations correspond to Centreville (CTR), Jefferson Street (JST), Yorkville (YRK), Georgia Institute of Technology (GT). Detailed sampling periods are shown in Table S1. In panel (a), since the perturbation experiments show that 16% of SOA from α-pinene oxidation is apportioned into isoprene-OA (Fig. S5a), we only include 84% of modeled SOA from MT+O$_3$/OH when comparing with LO-OOA for the sites with isoprene-OA. The mean bias (MB), mean error (ME), and normalized mean bias (NMB) for each site are shown in each panel. The slopes and correlation coefficients (R) are obtained by least square fit. The error bars indicate the standard error. In panel (b), average SOA$_{MT+SQT}$ concentration in PM$_{2.5}$ during each sampling period is reported.

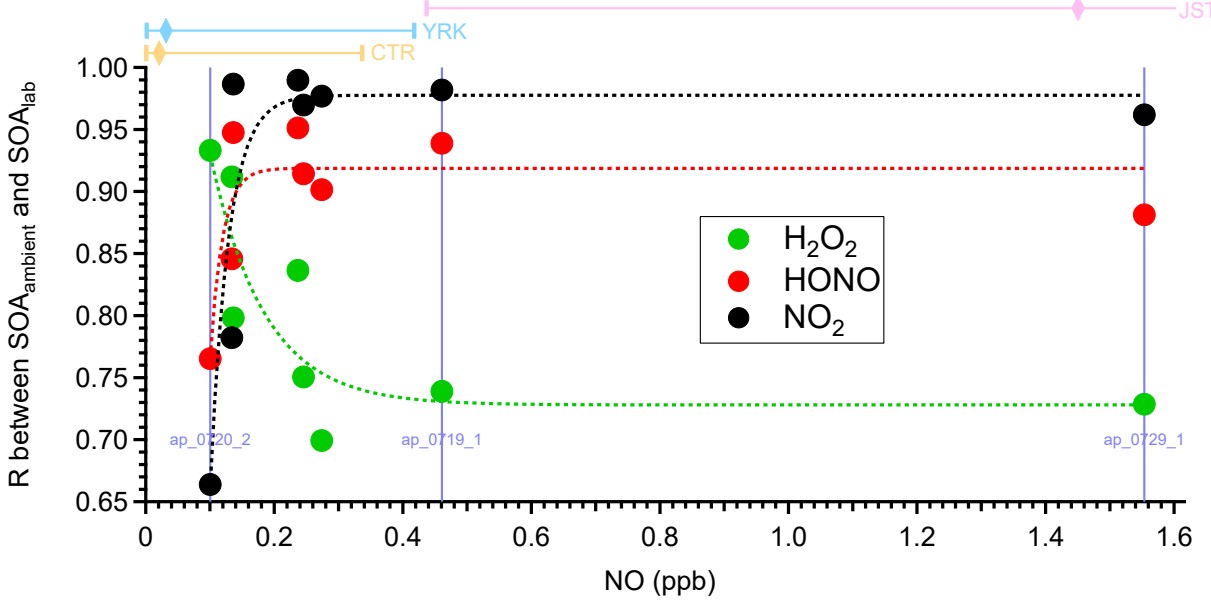

Fig. 7. The correlation coefficients between the mass spectra of OA formed in laboratory under different NO conditions ("$SOA_{lab}$") and those of OA formed in ambient α-pinene perturbation experiments ("$SOA_{ambient}$"). The subscripts "lab" and "ambient" indicate the SOA formed under laboratory conditions and ambient conditions, respectively. Three different oxidant sources (i.e., $H_2O_2$, HONO, and $NO_2$) are used to create different NO concentrations in laboratory studies. The mass spectra of "$SOA_{ambient}$" are calculated by comparing the mass spectra of OA during Chamber_Af and those of extrapolated Chamber_Bf (section S7 of Supplement). To calculate reliable mass spectra of "$SOA_{ambient}$", only the experiments with significant OA enhancement are analyzed and shown here (Appendix A). The x-axis is the average NO concentration during each perturbation experiment. The data points on the same vertical line (i.e., the same NO concentration) are from the same perturbation experiment, but compared to three different laboratory experiments. The dashed lines are used to guide eyes. The bars on top of the figure represent the 10th, 50th, and 90th percentiles of NO concentration for CTR (Centreville, AL), YRK (Yorkville, GA), and JST (Jefferson Street, GA) in 2013. The NO concentration is measured by the SouthEastern Aerosol Research and Characterization (SEARCH) network. The 90th percentile of NO concentration in JST is 14.8 ppb, which is not shown in the figure.

**Acknowledgments**

L.X. and N.L.N. acknowledged support from US Environmental Protection Agency (EPA) STAR Grant RD-83540301, and National Science Foundation (NSF) grants 1555034 and 1455588. The HR-ToF-CIMS was purchased with NSF Major Research Instrumentation (MRI) grant 1428738. HOTP contributions were supported by a Presidential Early Career Award for Scientists and Engineers (PECASE). The authors thank R. J. Weber and M. R. Canagaratna for helpful discussions, the SEARCH personnel for their many contributions, the CSRA for preparing emissions and meteorology for CMAQ simulations. The US EPA through its Office of Research and Development supported the research described here. It has been subjected to Agency administrative review and approved for publication but may not necessarily reflect official Agency policy.

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

## Appendix A. Data Analysis Method for Perturbation Experiments

The most challenging and important analysis is to determine if the perturbation results in a statistically significant change in the mass concentration of OA factors. We perform the following analysis to calculate the changes in the mass concentration of OA factors after perturbation, to determine if the change is significant, and to evaluate if the change is simply due to ambient variation.

The duration of one perturbation experiment is about 130min, including four periods: Amb_Bf (~30min), Chamber_Bf (~30min), Chamber_Af (~40min), and Amb_Af (~30min), as illustrated in Fig. A1. Firstly, we assume that the ambient variation is linear during both the Chamber_Bf and Chamber_Af periods (i.e., when instruments are connected to chamber and not sampling the ambient aerosol) and that the ambient variation can be represented by interpolating Amb_Bf and Amb_Af. The validity of this assumption will be discussed shortly. To obtain the slope of ambient variation, we analyze the combined Amb_Bf and Amb_Af data and use Theil-Sen estimator(Sen, 1968). The Theil-Sen estimator is a method to robustly fit a line to a set of two-dimensional points (i.e., concentration "$C$" and time "$t$" in this study). This method chooses the median of the slopes $(C_j-C_i)/(t_j-t_i)$ determined by all pairs of sample points. Compared to simple linear regression using ordinary least squares, the Theil-Sen estimator is robust and insensitive to outliers. Unless specifically noted, the slope is Appendix A is calculated from Theil-Sen estimator. Secondly, we use the slope to extrapolate the Chamber_Bf data to estimate aerosol concentration inside the chamber during the Chamber_Af period if there were no VOC injection. We refer to this estimated aerosol concentration as "extrapolated Chamber_Bf" and use it as the reference to calculate the change in aerosol mass concentration after perturbation. We extrapolate the Chamber_Bf data, instead of ambient data, because the OA concentration in chamber is lower than that in the atmosphere due to wall loss. Thirdly, we calculate the changes in the concentration of OA factors based on the difference between measured Chamber_Af data and "extrapolated Chamber_Bf".

For each perturbation experiment, after calculating the changes in the concentration of OA factors, we develop a set of criteria to determine if the changes are statistically significant and if the changes are simply due to ambient variation. The increase in the concentration of an OA factor

needs to satisfy all criteria to be considered as statistically significant and not due to ambient
variation.
**Criterion 1**: The difference in concentration between Chamber_Af and extrapolated Chamber_Bf
must be significant. We use T-test and 95% confidence interval.
**Criterion 2**: The slope of all data points or the first 8 data points during the Chamber_Af period
is significantly different from the slope of aerosol concentration during the Chamber_Bf period.
The rationale behind this criterion is that if the perturbation causes a substantial change in the
concentration of an OA factor, its slope during the Chamber_Af period should be different from
that during the Chamber_Bf period.

The slope of aerosol concentration during the Chamber_Af period is obtained in the
following way. We calculate the slope by using (1) all data points and (2) only first 8 data points
during the Chamber_Af period. This is because the concentration of factors firstly increases after
perturbation and then decreases due to dilution (Fig. A1). In this case, the slope obtained by fitting
all data points might be negative and will not reflect the initial increase in concentration (e.g., LO-
OOA of ap_0805_1 in Fig. S9a). Using only the first few data points during the Chamber_Af
period can avoid this issue. We select the first 8 data points in this period because the
concentrations of total OA and OA factors typically reach the highest at the 8th point (i.e., ~16min
after injection). The slope is calculated by Theil-Sen estimator.

The slope of aerosol concentration during the Chamber_Bf period is analyzed in the
following way. In order to determine if the slope in Chamber_Af is significantly different from
that in Chamber_Bf, we use bootstrap analysis (1000 times) to obtain a distribution of the slope of
Chamber_Bf. In brief, in each random resampling of Chamber_Bf with replacement, a slope is
calculated by Theil-Sen estimator. Then, 1000 times resampling provides a distribution of slope in
Chamber_Bf. The 5% and 95% percentiles of the slope distribution are compared to the slope of
Chamber_Af to determine if the slopes are significantly different. If the slope of Chamber_Af
(from either all data points or the first 8 data points) is smaller (or larger) than the 5% (or 95%)
percentile, the slopes in Chamber_Bf and Chamber_Af are significantly different.
**Criterion 3**: The slope of all data points or the first 8 data points during the Chamber_Af period
is significantly different from the slope of ambient data (i.e., combined Amb_Bf and Amb_Af).
The rationale behind this criterion is the same as the second criterion. That is, if the perturbation
causes a substantial change in the concentration of an OA factor, its slope during the Chamber_Af
period should be different from that in ambient data. The procedure to obtain a distribution of
slopes in the ambient data (combined Amb_Bf and Amb_Af) is same as Criterion 2.
As mentioned above, one critical assumption is that the ambient variation is linear during
both the Chamber_Bf and Chamber_Af periods (i.e., when instruments are connected to chamber
and not sampling the ambient aerosol) and that the ambient variation can be represented by
interpolating Amb_Bf and Amb_Af. We design the following pseudo-experiment to test the
validity of this assumption. In brief, we perform the same analysis as we did for the perturbation
experiments, but using ambient data **only** (i.e., no perturbation data). We firstly randomly select a
data point, which defines the start point of one pseudo-test. Secondly, based on the start point, we
obtain the concentration of OA factors during "Amb_Bf" period, (i.e., from start point to start point
+ 30min), "Chamber_Bf" period (i.e., from start point + 30min to start point + 60min),
"Chamber_Af" period (i.e., from start point + 60min to start point + 100min), and "Amb_Af"
period (from start point + 100 min to start point + 130min). This mimics the sampling periods in
a real perturbation experiment. Thirdly, we calculate the slope of ambient period (i.e., combined
"Amb_Bf" and "Amb_Af" periods) and the slope of chamber period (i.e., combined "Chamber_Bf"
and "Chamber_Af" periods). Fourthly, we calculate if the slope of chamber period is significantly
different from the slope of ambient period. We repeat this test 1000 times and then obtain the
probability of whether the slopes of chamber period and ambient period are significantly different.
Fig. A2a shows the probability that the slopes of chamber period and ambient period are
not significantly different for five factors. The larger this probability is, the more reliable the
linearity assumption is. The average probability is ~50% for all factors, without discernible diurnal
trends. This suggest that there is ~50% chance that the linear variation assumption is valid. Since
the linearity assumption is not perfect, we develop another criterion to constrain the potential
influence of ambient variation on the interpretation of perturbation results.
**Criterion 4**: From the above pseudo-experiment on ambient data only, we can calculate the
relative change in slope between "chamber period" and "ambient period" by
relative change in slope $= \dfrac{\text{Slope}_{\text{Chamber}} - \text{Slope}_{\text{Amb}}}{\text{Slope}_{\text{Amb}}}$  Eqn 1
In each pseudo-experiment test, we calculate a relative change in slope between "chamber period"
and "ambient period". By repeating the pseudo-experiment test 1000 times, we obtain a frequency
distribution of the relative change in slope for each OA factor (Fig. A2b). This frequency
distribution indicates the probability that certain relative change in slope occurs due to ambient
variation. Take LO-OOA as an example, the probability that the relative change in slope varies by
a factor 8 due to ambient variation is ~1%. Thus, if the relative change in slope of LO-OOA in a
α-pinene experiment is 8, the change is unlikely due to ambient variation. We use the 5% and 95%
percentiles from the frequency distribution as the fourth criterion to determine if the changes in
the concentrations of OA factors in each perturbation experiment are due to ambient variation. In
other words, if the relative change in slope between Chamber_Af and ambient data in a real
perturbation experiment falls outside of the 5% or 95% percentiles, the changes in the
concentrations of OA factors are likely due to perturbing chamber with VOC, instead of ambient
variation. This criterion strictly considers the influence of ambient variation. In general, the
comparison in slope is an optimal option to account for ambient variation, because the influence
of ambient variation is unlikely to coincide with the perturbation.
Based on these 4 criteria, the OA factors with significant changes in their mass
concentrations as a result of perturbation are shown in Fig. 4. LO-OOA is enhanced in 14 out of
19 α-pinene experiments. However, total OA is only enhanced in 8 out of 19 α-pinene experiments.
Several reasons can contribute to the different behaviors of LO-OOA and OA. Firstly, as total OA
has multiple sources, the enhancement in one factor does not guarantee an enhancement of total
OA. For instance, in some perturbation experiments, while LO-OOA is enhanced, the
concentration of other factors steadily decreases due to ambient variation. The increase in LO-
OOA and decrease in other factors compensate each other and result in a lack of enhancement in
total OA. Secondly, based on the pseudo-experiment, we note that total OA is more easily affected
by ambient variation than a single OA factor. For example, the 95% of the relative change in slope
of total OA is 3.59, which is larger than any OA factors (Fig. A2b). Thus, the criteria for the change
in total OA concentration to be considered as significant are stricter than those for a single OA
factor. Thus, some experiments with significant changes in LO-OOA do not have significant
changes in total OA.

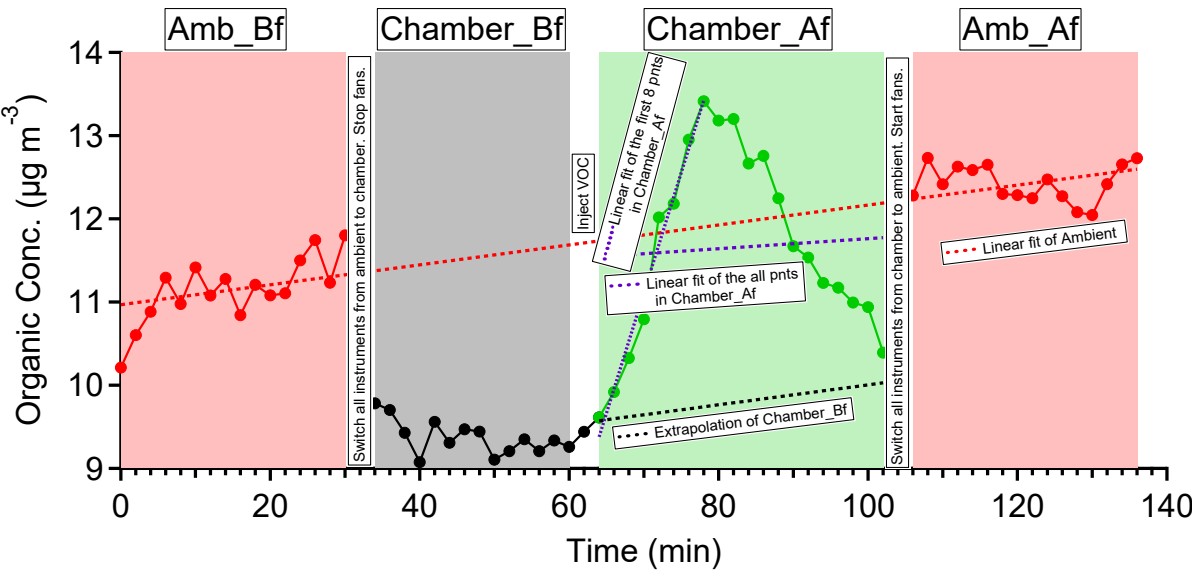


Fig. A1. Time series of OA in experiment ap_0801_1 to illustrate the analysis method. Each
perturbation experiment includes four periods: Amb_Bf (~30min), Chamber_Bf (~30min),
Chamber_Af (~40min), and Amb_Af (~40min). "Amb" and "Chamber" correspond to the periods
when the instruments are sampling ambient and chamber, respectively. "Bf" and "Af" stand for
before and after perturbation, respectively. The solid lines are measurement data. The dashed red
lines are the linear fit of ambient data (i.e., combined Amb_Bf and Amb_Af). The slope is used to
extrapolate Chamber_Bf data to Chamber_Af period (i.e., black dashed line). The dense dashed
purple line is the linear fit of the first 8 points during the Chamber_Af period. The sparse dashed
purple line is the linear fit of all data points during the Chamber_Af period. During this period, the
difference between measurements (i.e., solid green data points) and extrapolated Chamber_Bf (i.e.,
dashed black line) represents the change in organic concentration caused by perturbation.

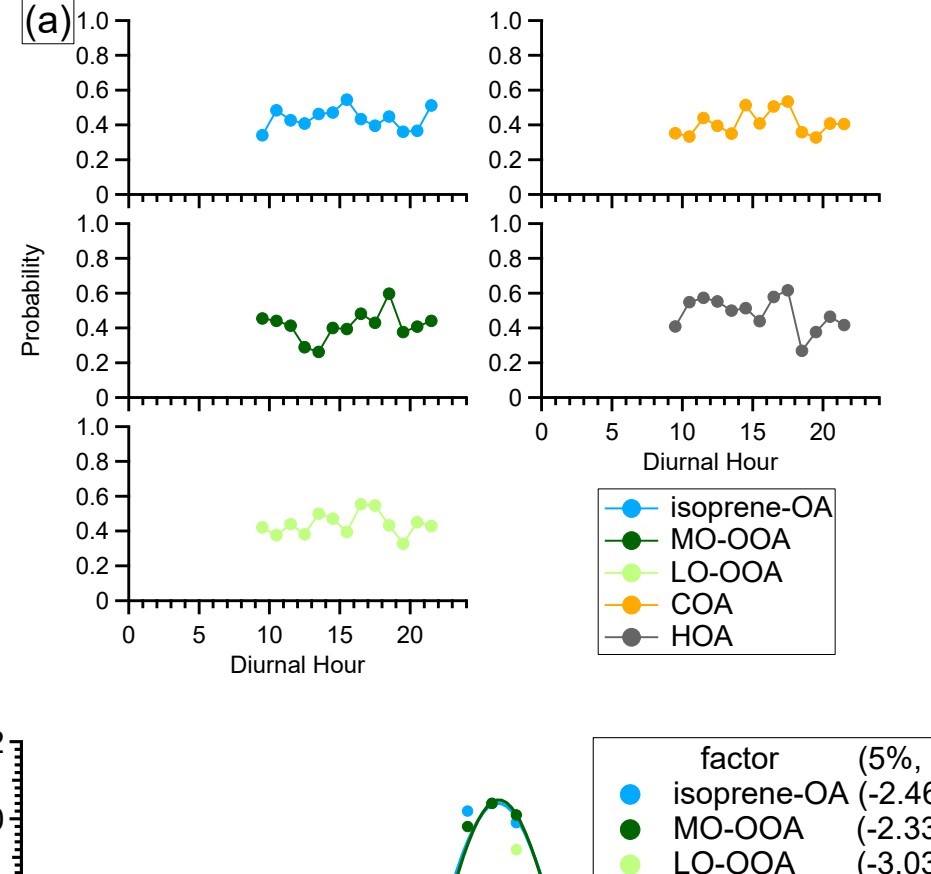


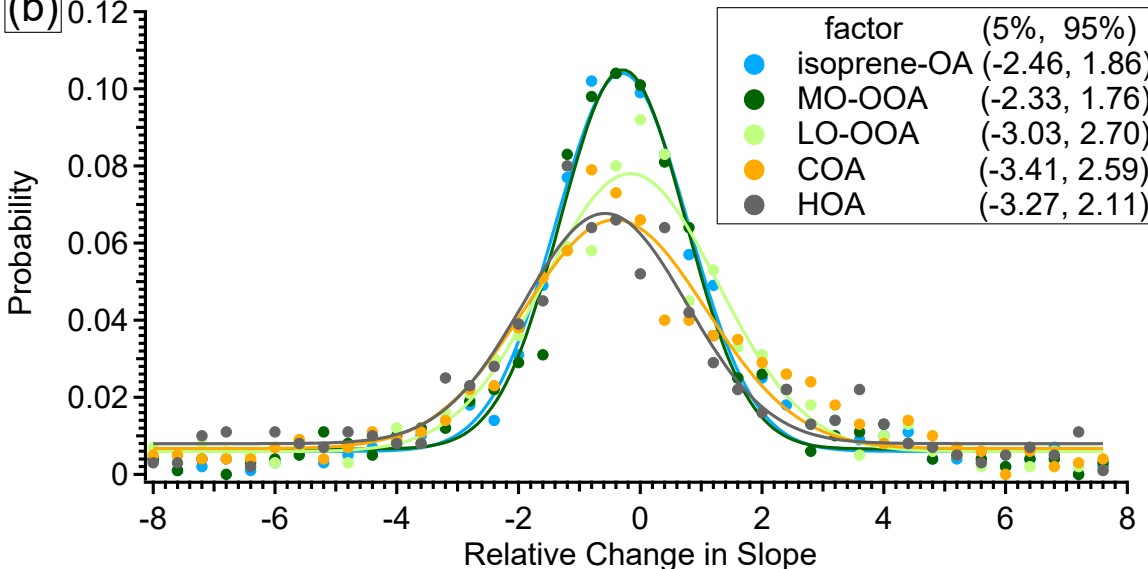


Fig. A2. (a) The diurnal trends of the probability that the slopes between ambient periods (i.e., Amb_Bf and Amb_Af periods) and chamber periods (i.e., Chamber_Bf and Chamber_Af periods) are not significantly different in the pseudo-experiment. (b) The frequency distribution of the relative change in slope. The data points are fitted using Gaussian function. The numbers in the box represent the 5% and 95% percentile of the Gaussian fit.

## Appendix B. Ambient Perturbation Experiments with Acidic Sulfate Particles

Previous field observations showed strong correlation between isoprene-OA and sulfate (Xu et al., 2015a; Xu et al., 2016a; Budisulistiorini et al., 2015). Moreover, airborne measurements over power plant plumes in Georgia, U.S. observed enhanced isoprene-OA formation in the sulfate-rich power plant plume (Xu et al., 2016a). To probe the relationship between isoprene-OA and sulfate, we conducted perturbation experiments in August 2015 by injecting acidic sulfate particles (i.e., a mixture of $H_2SO_4$ and $MgSO_4$) into the 2 $m^3$ Teflon chamber. This mimics the airborne measurements over power plants, which introduce sulfate into the atmosphere (Xu et al., 2016a).

The experimental procedure in 2015 experiments is generally similar to those in 2016 experiments, but has the following modifications. Firstly, in order to avoid the depletion of species which can uptake to sulfate particles, we kept one fan on during the Chamber_Bf and Chamber_Af periods to enhance the air exchange between chamber and atmosphere. Secondly, considering the fan is on during sulfate injection to enhance mixing chamber air with ambient air, we only use the Chamber_Bf and Chamber_Af periods to calculate the changes in OA factors. The Criteria (1)(2)(4) are applied in 2015 experiments. Thirdly, the Chamber_Bf period is ~40 min in 2015 experiments, which is slightly longer than the 30 min in 2016 experiments. Fourthly, the HR-ToF-CIMS was not deployed in 2015 experiments.

The acidic sulfate seed particles were introduced into chamber by atomizing 0.88mM $H_2SO_4$ + 0.48mM $MgSO_4$ mixture solution from a nebulizer (U-5000AT, Cetac Technologies Inc., Omaha, Nebraska, USA). One important interference in these sulfate perturbation experiments is the trace amount of organics in solvent water [i.e., HPLC-grade ultrapure water (Baker Inc.)], which is used to prepare the $H_2SO_4$+$MgSO_4$ solution. These organics were injected into chamber together with sulfate. We utilize the multilinear engine solver (ME-2) to constrain the organics from solvent water (i.e., $H_2O$-Org). Unlike the PMF2 solver which does not require any a priori information of mass spectrum or time series, the ME-2 solver uses a priori information to reduce rotational ambiguity among possible solutions(Canonaco et al., 2013; Paatero, 1999). We obtained the reference spectrum of organic contamination (i.e., the a priori information for ME-2 solver) by atomizing the $H_2SO_4$+$MgSO_4$ solution directly into AMS. The ME-2 solver successfully extracted a factor (i.e., denoted as $H_2O$-Org factor, Fig. B1), which showed a clear enhanced concentration during atomization (Fig. B2).

A total of four experiments were performed and details are summarized in Table B1. As

shown in Fig. B2, the isoprene-OA factor increases in all three daytime experiments, but not the
nighttime experiment. Based on current understanding of isoprene-OA factor, this enhancement is
likely due to the reactive uptake of IEPOX. The lack of enhancement in nighttime experiment is
consistent with low IEPOX concentration at night (Hu et al., 2015). Our results provide direct
observational evidence that acidic sulfate particles lead to increase in isoprene-OA, which supports
results from previous studies (Xu et al., 2015a; Xu et al., 2016a; Budisulistiorini et al., 2015). Due
to lack of measurements of gas-phase organic compounds, we are unable to identify the reactive
species. Other species, such as glyoxal (Kroll et al., 2005), isoprene hydroperoxides (Liu et al.,
2016), and HOMs (Ehn et al., 2014), also have the potential to uptake to acidic sulfate particles
and form SOA. Future experiments with comprehensive measurements of gas-phase organic
compounds can provide more insights into the identities of reactive uptake species.

We note that in non-atomizing period, the concentration of $H_2O$-Org factor is close to zero,

but not zero. Since $H_2O$-Org arises from the atomizing solution, it should only exist during
atomizing periods. Thus, the non-zero concentration suggests the limitation of the ME-2 solver
and cautions are required when using ME-2 solver to resolve one factor based on a specific mass
spectrum. This limitation does not affect the conclusion that the enhancement in isoprene-OA is
likely due to the reactive uptake of organic species, as we further verify that the organic increase
in three daytime perturbation experiments with sulfate particles cannot be solely explained by the
organic contamination in atomizing water, from the following two aspects. For example, we
atomize the solution directly into AMS and find that the Org/$SO_4$ ratio is 0.025. This value is
significantly lower than the Org/$SO_4$ ratio in the three daytime sulfate perturbation experiments
(i.e., 0.048-0.059), but close to the nighttime sulfate perturbation experiment (i.e., 0.022) (Fig. B4).

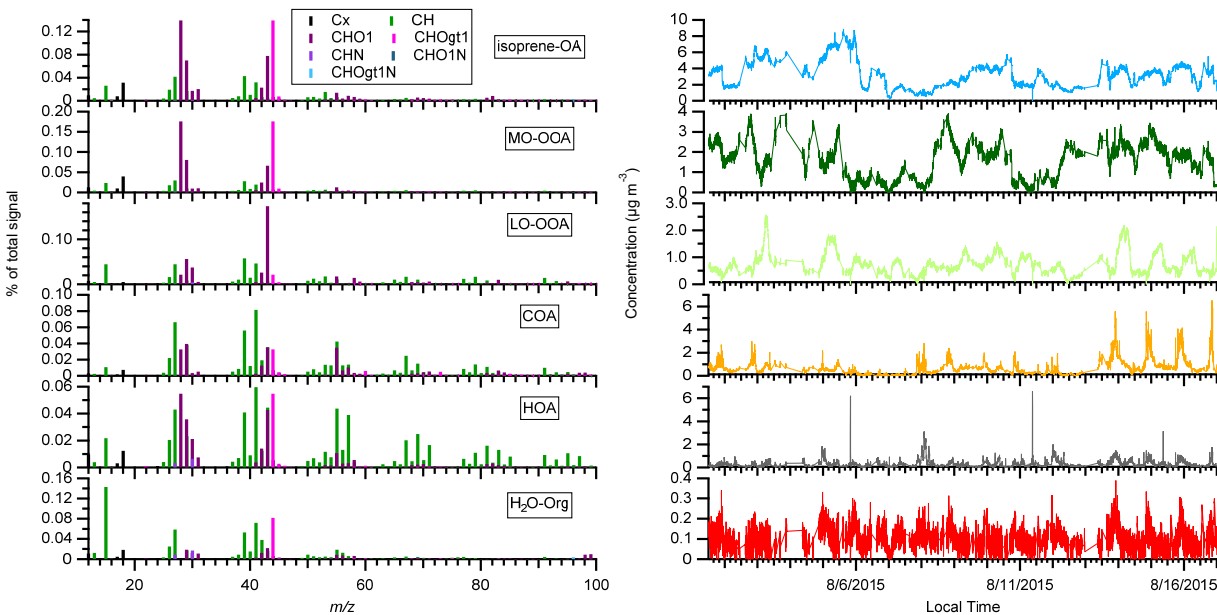


Fig. B1. The mass spectra and time series of OA factors in the 2015 acidic sulfate particle
perturbation measurements. Note that the perturbation periods are included in the time series.

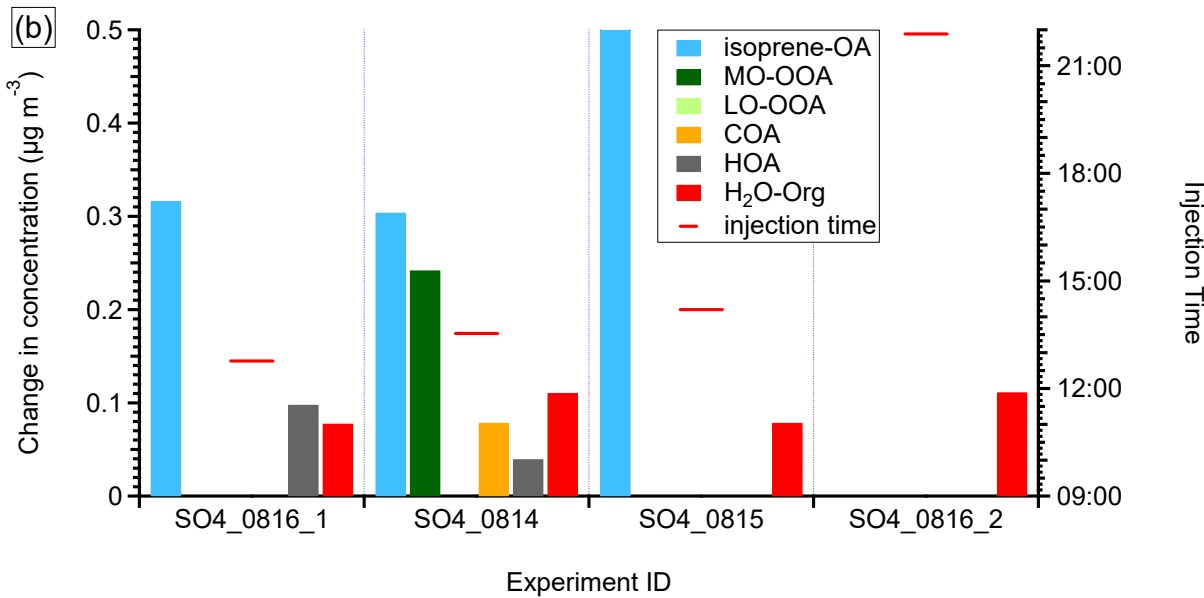


Fig. B2. The statistically significant changes in the concentrations of OA factors after perturbation
by acidic sulfate particles. The experiments are sorted by perturbation time. The changes in
concentration are the difference between measurements during the Chamber_Af period and mass
concentration extrapolated from the Chamber_Bf period. A set of criteria are developed to evaluate
if the changes are significant and if the changes are due to ambient variation (Appendix A). $H_2O$-
Org factor in these sulfate perturbation experiments represents organic contaminations in
atomizing water.

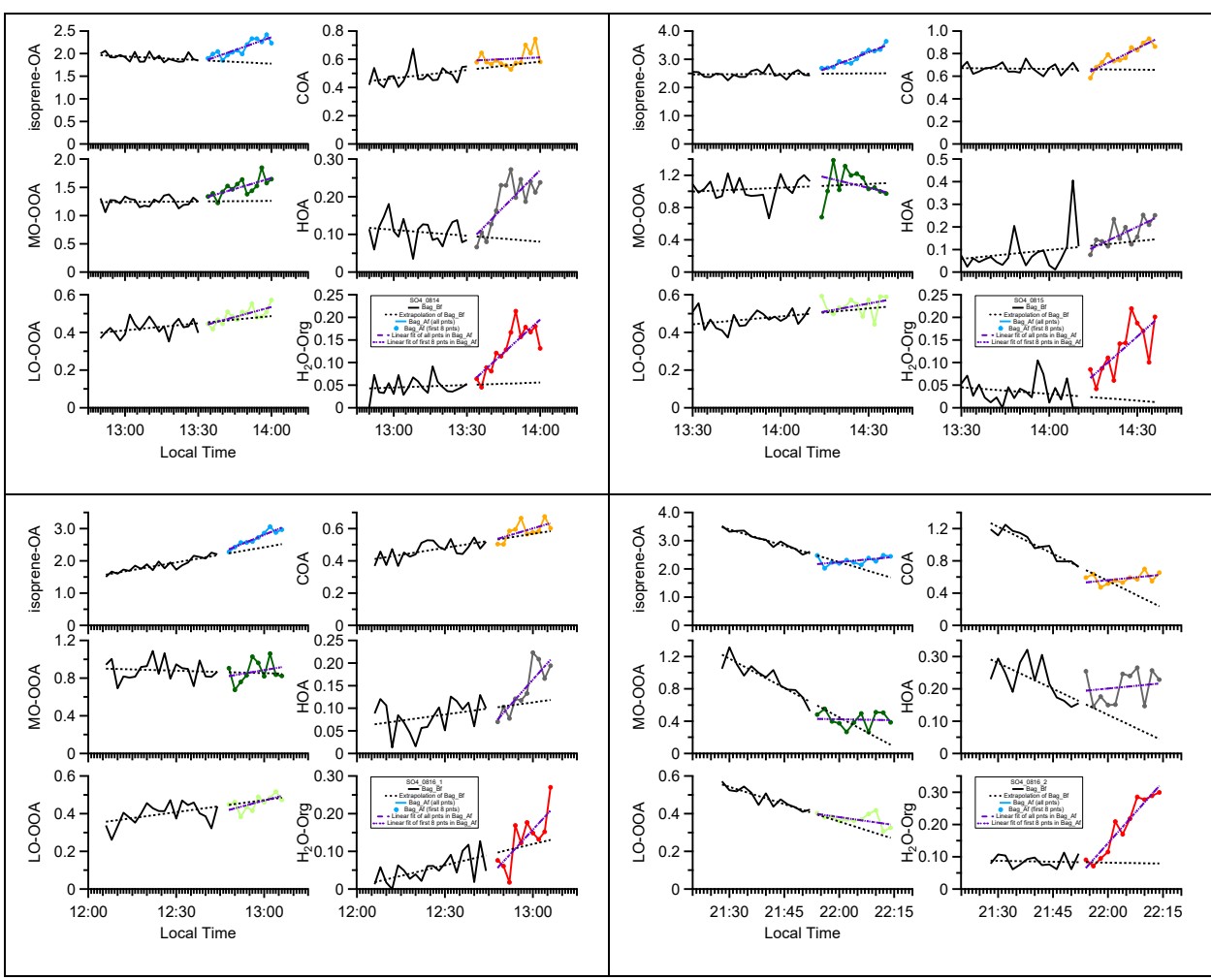

Fig. B3. Time series of OA factors in each sulfate perturbation experiment.









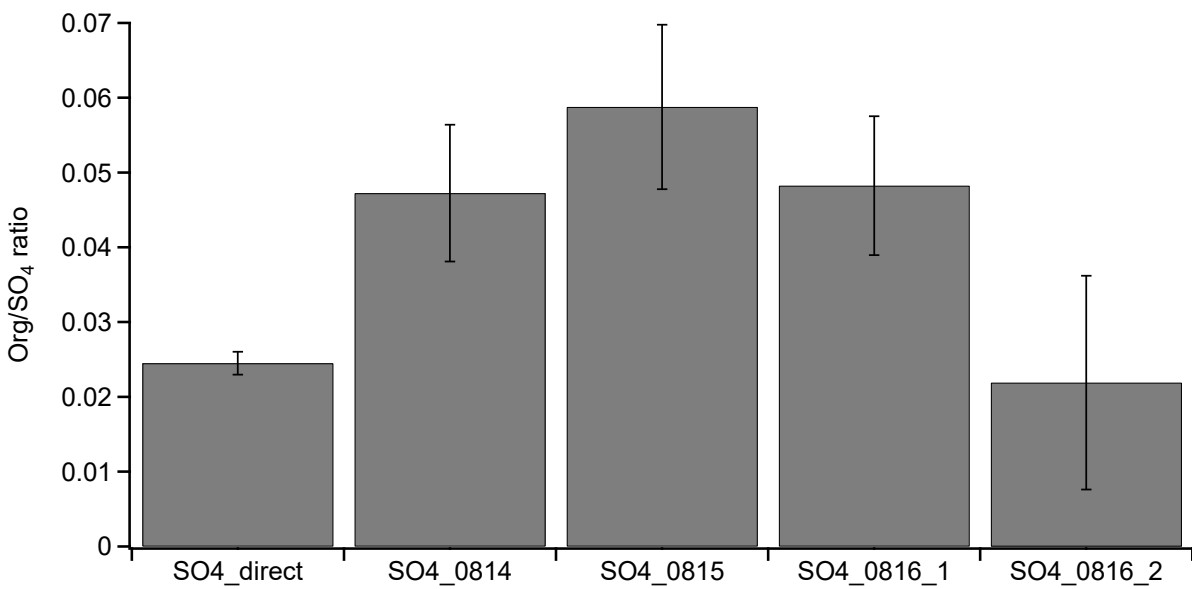


Fig. B4. The Org/SO4 ratio in sulfate perturbation experiments and laboratory tests by directly
atomizing $H_2SO_4$ + $MgSO_4$ mixture solution into AMS (i.e., SO4_direct).

Table B1. Experimental conditions for sulfate perturbation experiments.

| Perturbation | Expt ID[a] | Date | Injection Time | Perturbation Amount[b] | NO[c] (ppb) | NO$_2$[c] (ppb) | O$_3$[c] (ppb) |
|---|---|---|---|---|---|---|---|
| sulfate | SO4_0814 | 8/14/2015 | 13:32 | 16.29 | 0.51 | 5.86 | 59.8 |
| | SO4_0815 | 8/15/2015 | 14:12 | 14.33 | 0.18 | 4.79 | 63.0 |
| | SO4_0816_1 | 8/16/2015 | 12:46 | 14.52 | 0.36 | 4.08 | 53.2 |
| | SO4_0816_2 | 8/16/2015 | 21:53 | 13.92 | 0.03 | 5.40 | 35.6 |

[a]Expt ID is named as "perturbation species + date + experiment number". For example,
SO4_0816_1 represents the first sulfate perturbation experiment on 08/16.
[b]The unit for the perturbation in sulfate experiments is $\mu g \ m^{-3}$. The perturbation amounts of sulfate
are calculated from Chamber_Af – extrapolated Chamber_bf.
[c]Average concentration during the Chamber_Af period.













