# Peer review of "Experimental and Model Estimates of the Contributions from Biogenic Monoterpenes and"

_Atmospheric Chemistry and Physics, 2017_

## Referee Comment (RC1) · Anonymous Referee #1 · 30 Jan 2018

This paper shows some interesting results from a novel experiment involving taking ambient air in an urban environment and conducting a chamber experiment after enhancing the concentrations of VOCs. The main emphasis of this paper is the use of $\alpha$-pinene and $\beta$-caryophylene, the characterisation of the SOA produced using aerosol mass spectrometry and inferences are drawn regarding their contribution to particulate concentrations in the region.

Overall, this is a nice piece of work and well within the journal's remit. However, I do think that the significance is a little overblown in places and the authors need to express more caution in how they interpret some of the results. In spite of their statements

otherwise, this is not a true simulation of atmospheric processes (see below), there are a few PMF-specific subtleties that aren't taken account of and how reliably this can be projected onto the wider world is open to question on a number of levels. But in spite of these issues, the conclusions are largely sound and this deserves to get published. I therefore recommend publication with minor revisions.

General Comments:

This paper makes the assumption of $\alpha$-pinene and $\beta$-caryophylene being representative of monoterpenes and sesquiterpenes respectively. While these are common assumptions made in the community and the VOCs are both very well studied, their overall representativeness is in question because the level of oxidation in SOA from different precursors are known to vary substantially between compounds (Alfarra et al., Atmos. Chem. Phys., 13, 11769-11789, 10.5194/acp-13-11769-2013, 2013). This is especially true of the sesquiterpenes, as difficulties in working with a number of these compounds means that we lack data on a large subset of these. This should be discussed and any evidence to support this assumption properly cited.

Throughout the manuscript, there is a general tendency to treat LO-OOA and MO-OOA as defined chemical entities, whereas the truth is that these represent reductions of highly complex chemical systems and the exact factors reported are known to vary dataset to dataset. While common trends have been noted in terms of behaviour and mass spectral profile, their exact mass spectral nature depends on the measurement location and season and in many cases (particular in the wintertime), PMF will fail to separate them at all, instead returning a single OOA factor. This must be discussed in a meaningful manner in the introduction and discussion because it adds an extra layer of nuance to the results.

Related to the above point, there seems to be a general assumption that PMF had adequately accounted for the new SOA being formed, but in my mind, the decrease in MO-OOA in response to the $\beta$-caryophylene experiments in particular raises a number

of red flags because this implies that the data model didn't hold and the factorisation wasn't sound. The authors need to pay much closer attention to whether the factorisations 'worked' or not; a good starting point would be inspecting the residuals (e.g. Q/Qexp) as a function of time over the course of the experiments and if they positively correlate at all with the amount of additional SOA, this would imply that PMF has failed to capture the chemistry perfectly.

The modelling work presented in section 3.6 left me slightly confused as to what the authors were trying to achieve and how. The text in the main article would suggest that an updated scheme is being compared with a very old one, but the supplement seems to say that specific mechanisms have been added here. This must be clarified. Also, as pointed out later in this section, this work does not directly preclude that other precursors may be contributing and the discussion dealing with this relies heavily on inferences drawn from the literature, so this work isn't really that dramatic a result in how it is presented now. I would suggest a more defined modelling experiment is constructed around a clear working hypothesis. This could just be a case of making the work shown here clearer and moving material from the supplement to the main article.

More generally, I noted a very odd tendency to leave certain pertinent (and in many cases interesting) details in the supplement that maybe should have been given more prominence or at least linked to the main article better. For example, the box modelling described in section S6 was very interesting, but it wasn't clear at all how this fit into the narrative of the main article. I also had a hard time reconciling the information about the CMAQ runs in the main article and the supplement as well (see above). I would revise what information goes where, using the main article for the discussions relating to the scientific arguments and making sure the material in the supplement is purely technical detail in support of this.

Specific comments:

[Figure]

Line 95: When saying 'representative urban', please be specific about what type of urban site this (e.g. background) and how you qualify this statement.

Line 115: What counts as 'too much' SOA and why?

Line 265: It has long been shown that $\alpha$-pinene SOA produced in chambers produces a mass spectrum that is similar to LO-OOA and given that this mass spectral profile is also seen in the presence of strong emitters of this VOC (e.g. temperate and boreal forests), the case for $\alpha$-pinene SOA being a strong contributor to LO-OOA has never really been in doubt in this reviewer's opinion. Why is the evidence presented here any more 'direct' than those published previous? While the perturbation experiment does indeed produce LO-OOA as retrieved using PMF, this retrieval is still based solely on mass spectral similarity, so I would contend that this does not really present any new evidence to this effect.

Line 280: The fact that the oxidation rate of VOCs is dependent on oxidant concentration is very well established in kinetics. The discussion regarding this observation would be considered pointing out the obvious to many. It would be far more useful if a quantitative relationship with ozone concentration could be reported here.

Line 286: An alternative explanation here is that the experimental set-up here was not conducive to HOM formation for whatever reason. This should be added as a caveat.

Line 309: There is a major problem with this statement; the results indicate that the $\beta$-caryophylene SOA spectrum to be represented by PMF as a combination of the LO-OOA and COA mass spectra, but it would be a mistake to imply in any way that it is producing two 'types' of OA (this is clarified later in the manuscript but it is ambiguous here). Issues about the quality of the PMF retrieval aside (see above), in the hypothetical situation that there is an environment with a mixture of cooking and biogenic SOA, PMF will likely still separate these because it determines factors not just by mass spectral profile but by temporal profile, so would still return factors corresponding to cooking and an average of biogenic SOA from all sources. The only situation I could

think where this would be a problem is if monoterpene and sesquiterpene SOA formation were not well matched temporally, in which case I could see how the COA-like component of sesquiterpene SOA would manifest as 'mixing' between the cooking and biogenic SOA factors, but this would be evident in the temporal profiles.

Line 350: How much more 'realistic' is this? While this would give a more life-like oxidant and NOx background, given that the chamber walls will act as a sink of VOCs, radicals and particles, I would still expect that the precursor perturbations would have to be higher than typical atmospheric concentrations to achieve realistic SOA concentrations and consequently have a higher VOC:NOx ratio. This must be discussed in an objective manner and while some of this is touched on in the supplement, it's kind of glossed over in the main article.

Figure 6: The caption of this figure is excessively long.

Line S477: This doesn't make sense. Why would the solver reduce the concentration of MO-OOA because it had been added to? I find it more likely that there was a breakdown in the data model and mass was being erroneously rotated out of the factor. This is undesirable, but also feeds into the discussion above regarding the relationship with COA.

Line S480: What other studies?

Technical comments:

Line 113: Please be more specific over which VOCs are anthropogenic vs biogenic. The word 'respectively' does not work when four are listed.

Line 179: Why not saturated fatty acids?

Line 119: Correct 'concentration' to 'concentrations'.

Line 184: What are the oxidation states in each instance?

Section 2.2: Please specify the materials used for the aerosol and gas sampling lines.

[Figure]

Was the gas sampling line filtered or heated?

Line 200: The CMAQ grid cell depends on the exact model set-up being used. Is this referring to the activity specified in section 2.6? If so, this should be stated here.

Section 2.6: Please provide references or web links for the various inventories used.

Line S909: Pearson's R is not dependent on any numerical fitting method; it is merely a comparison between the two datasets.

---

## Referee Comment (RC2) · Anonymous Referee #2 · 31 Jan 2018

The manuscript presented by Xu et al. proposes an interesting study on the contribution of the oxidation of alpha-pinene and caryophyllene to the SOA mass observed in the S.E.-US. The characterization of SOA generated in the lab-in-the-field smog chamber was performed using an aerosol mass spectrometer. Overall the work performed in this study is good and fall within the scope of the journal. However, I think the conclusions proposed from the PMF analysis/chamber experiments are not always well sustained and more caution should be taken when extrapolating the results.

General comments: The authors should carefully review their paper and avoid the repetition between the main text and the SI. At many places, sentences are duplicated

and are not useful. However, some important details are left within the SI and should be moved to the main manuscript.

The authors should provide more information in the PMF analysis and provide the elementary checks to validate their analysis. For instance, it is a bit surprising that the factors don't change throughout the experiments (i.e. bf vs af) while significant perturbation has been made to the system. Or do the authors consider/claim that most of the SOA sampled in the ambient are formed from the oxidation of alpha-pinene or caryophyllene? In addition, we could expect that the fresh LO-OOA (formed within a few minutes, without lights) would have different signatures that LO-OAA formed in the atmosphere (aged SOA, formed from different chemistry,...). How do the factors correlate throughout the experiments: e.g. LO-OOA_Amb_Bf vs LO-OOA_Chamber_Af? How do the identified factors correlate with the reference MS? How do the residuals evolve throughout an experiment? How does alpha-pinene-derived LO-OOA correlate with caryophyllene derived LO-OOA? Overall, the authors should provide more statistical analyses in order to give a robust validation of the analysis.

The authors should report the concentration of the inorganics in their experiments and in case of significant concentrations of sulfate estimate the aerosol acidity. Indeed, the presence of acidic aerosols can lead to multiphase reactions (e.g. reactive uptake of IEPOX) that could greatly impact the SOA composition. In addition, an estimation (modeling?) of the concentrations of other VOCs would be interesting (especially isoprene). Ozonolysis of alpha-pinene leads to the formation of OH radicals, which could further react and oxidize alpha-pinene but also other VOCs present in the ambient air. The authors should discuss this possibility and provide more information in the background of the chamber/ambient air. As it is, the conclusions proposed in the paper on the potential increase of the IEPOX-OA or COA factors from the oxidation of alpha-pinene and caryophyllene, respectively appear speculative (correlations are not sufficient to validate such trend: r ~ 0.5). For instance, the authors could estimate the amount of IEPOX (thus isoprene) formed in the chamber to explain the formation of

IEPOX-OA and check if the numbers make sense or not.

Specific comments: Lines 104:111. Did the authors characterize the chamber? Mixing, wall losses,...

Lines 141: The authors claim that by having an overflow, it suppressed the particle loss. Did they mean reduce? Have you done some tests to validate such statement?

Lines 217-220: Why not using the outdoor chamber to do such experiments? Can the authors discuss the strategy here?

Lines 266-268: The decay of LO-OOA is quite fast and I do not think it can only explain by the dilution and or dead-volume. The residence time in the chamber is $\sim$ 100 min. Where were located the sampling inlets?

Lines 278:284: It is quite expected. What is the point of the authors?

SI Line 150:157: These results are a bit intriguing. The data reported for the boreal forest do not exhibit prominent ions at m/z 53 or 82. The authors suggest that alpha-pinene/monoterpene can contribute to IEPOX-OA but according to Fig S7 the correlation is far to be obvious strong. The authors should compare the MS obtained in their study with other PMF data obtained from monoterpene-dominated areas (e.g. boreal forest).

---

## Referee Comment (RC3) · Anonymous Referee #3 · 27 Feb 2018

This paper presents results from experiments and model runs focusing on the monoterpene contribution to biogenic SOA in the SE US. A small Teflon reactor was used to oxidize ambient air to which single VOC precursor was added. Based on simple PMF analysis and simple CMAQ model runs, it is concluded that monoterpenes are major contributors to ambient OA in the SE US. The authors are well-known in the field and have published much excellent work, the paper falls within the scope of ACP, and has some interesting aspects. However, in my opinion the new evidence is weak, partially supported with circular logic, and is very overinterpreted. The new evidence is very insufficient to support the very strong conclusions. I don't see how this paper can be published in ACP in anywhere near its present form. I recommend that the authors go back to the drawing board and summarize the new experimental aspects into a paper whose conclusions are actually supported by the evidence presented. For example, the results on Appendix B seem more novel to this reviewer than the ones that are described in the main paper.

Note that I made this recommendation already in the access review, with the concurrence of the previous Editor, and hoping to avoid having to post this review in public. However, after an appeal by the authors, it was decided to publish the paper in ACPD anyway without significant revisions.

**Brief statement of the major issue**

1) The main problem of this paper is that the evidence presented does not support the conclusions. The conclusions are summarized in the paper title "Large Contributions from Biogenic Monoterpenes and Sesquiterpenes to Organic Aerosol in the Southeastern United States." Or L80-84: "We provide direct evidence that newly formed SOA from α-pinene […] and β-caryophyllene (representative sesquiterpene) dominantly contributes to LO-OOA in the southeastern U.S."

The new evidence presented in this manuscript has two parts:

1.a) Some interesting, but incomplete, experiments with an ambient reactor, that have been analyzed using PMF. What the authors have really shown with these experiments is (in their own words) "to qualitatively understand which OA factors the α-pinene SOA is apportioned into" (Supp. Info. L349-350). Most importantly, experiments with other precursors failed, and experiments using for example cloud pathways were not even attempted. It is highly likely (based on prior published work) that spectra from other precursors and pathways would also have looked similar to LO-OOA, so it is very problematic that those experiments failed or where not even attempted.

The authors do acknowledge (L415) that based on the experimental work alone "we do not conclude that LO-OOA arises exclusively from MT and SQT." Next, they do acknowledge that their title is just a hypothesis, after all the experimental evidence has been considered: (L387-391): "we propose that the major source of OOA in this region is the oxidation of MT and SQT by various oxidants (O3, OH, and NO3). To test this hypothesis, we use CMAQ to simulate pollutant concentrations across the southeastern U.S."

So given how weak the evidence from (1.a) is in terms of supporting the paper's conclusions, how strong is the complementary evidence from the model?

1.b)  The CMAQ model is run with a different parameterization for terpene SOA that has higher yields than a very old one. Not surprisingly, the terpene contribution to SOA in the SE US goes up in CMAQ. I understand that **the parameterization is in principle improved compared to some older ones, but how do we know it is really accurate?** The parameterization is still very simple, and does not included detailed chemistry such as a HOM mechanism. The justification about why this parameterization would be accurate when implemented in as complex a region as the SE US is very light and not satisfactory (sect. 2.6). I agree the model is "upgraded" (L25), but **not that it has been shown to be accurate**. There is a long history of simple parameterizations (after various "upgrades") being wrong when compared to ambient air. I do not see sufficient evidence that supports the accuracy of these model runs, so that strong conclusions about MT + SQT contributions (that could not be reached with the experiments alone) could now be reached with certainty.

In addition, the use of the CMAQ results suffers from circular logic. Section 1.a. ended with a *hypothesis* that LO-OOA was arising from MT & SQT. Now the fact that the CMAQ results are of the same order as LO-OOA is used as confirmation that indeed LO-OOA comes from MT & SQT. But we did not know that LO-OOA came from MT & SQT yet, that was only a hypothesis. The logical structure here is therefore problematic.

Other evidence from the literature is mentioned, such as high emissions of MT & SQT in the SE US, and the fact that some previous results suggest that anthropogenic SOA is not a major contributor in this region. The first one has been known for decades. The second result is not quite consistent with some modeling studies and measurements of fossil carbon (when taking into account that urban pollution has a significant fraction of non-fossil carbon, see Kim et al. (2015) and references therein). Together, the combined evidence is still not sufficient for the sweeping conclusions.

Ten years ago we didn't know about the IEPOX-SOA pathway (Paulot et al., Science 2009), that has since proven to be a major contributor to SOA over the SE US. A paper written in 2008 may have used the existing literature to conclude that isoprene was a minor contributor to SOA in the SE US, and would have been sorely wrong in doing so. We also didn't know till very recently about autoxidation being important in the atmosphere (Ehn et al., 2014). The importance of autoxidation for urban emissions in the US has recently been demonstrated (Praske et al., PNAS 2018), and is not included in the CMAQ runs or literature studies cited here. SOA formation in clouds is also highly uncertain (Ervens et al., 2011), and could also lead to LO-OOA through various mixtures of precursors and pathways (which remain almost completely unexplored). For example perhaps there is an isoprene SOA pathway through clouds that has not been discovered yet and that contributes half of the LO-OOA in the SE US. ***The present paper is implicitly saying that other precursors and pathways are not important in the SE US*, and risks shutting down research on other alternatives. Is that justified? In my opinion it is not, and the evidence is this paper is not anywhere near sufficient to justify its title and sweeping conclusions.**

**A little more detail on the major issues**

2) A very important problem with the manuscript is the logic of the PMF analysis (with both of the other reviewers explicitly pointing out serious problems in it). Indeed the a-pinene SOA is most similar to LO-OOA (also referred to as SV-OOA in older works). This has been known for a decade, see for example Fig 2C of Jimenez et al. (2009) (cited in the manuscript) and several other papers. Or in the words of reviewer 1: "the case for a-pinene SOA being a strong contributor to LO-OOA has never really been in doubt in this reviewer's opinion. Why is the evidence presented here any more 'direct' than those

published previous?" The same conclusion can be deduced quickly by comparing spectra of chamber a-pinene SOA and ambient LO-OOA from the AMS spectral databases (much like the authors do again in their Fig. 6). So this it is not a new finding of this work. Here an interesting (though not completely new) experimental procedure is used to reach once again a conclusion that was already firmly established in the literature. See for example the figure below (Kiendler-Scharr et al., 2009, as an example, but there are quite a few such comparisons in the earlier AMS literature. Note that the Chebogue BSOA represents the outflow for the E US, similar to the work in the present paper):

[Figure]

FIGURE 3. Mass spectra comparisons of average plant chamber BSOA (a) vs α-pinene SOA (b), a BSOA component derived from field data (c)

Therefore the conclusion is not really new and has been known for over a decade. So it seems surprising that suddenly the same specific result allows the present authors to reach far more general conclusions. **The critical flaw is in the logic flow. It is true that (a) a-pinene SOA is most similar to LO-OOA. However, that cannot be used to conclude that (b) ambient LO-OOA in the SE US is mostly from a-pinene. Hypothesis (a) being true is a necessary, but not sufficient condition for the converse hypothesis (b) to be true. Much more evidence is needed to prove hypothesis (b) and to disprove alternative hypotheses.**

A critical piece of additional evidence would include proving that spectra from other sources of SOA likely to be present in the region are not spectrally similar to LO-OOA. This is not shown in the manuscript. Experiments with other precursors were attempted (L313): "by injecting isoprene, m-xylene, or naphthalene, which are major biogenic and anthropogenic emissions, respectively. However, the SOA formation from these VOCs is not detectable." **So the experiments failed, and as a consequence no spectra from other likely sources are available to establish that spectra from those sources do not look like LO-OOA.** A look at the AMS literature and the AMS databases suggest that

spectra from these and other precursors have spectra which are indeed very similar to ambient LO-OOA. So it is very problematic that the experiments did fail.

Inexplicably the authors do not use the publicly available database spectra, nor perform regular chamber experiments for other precursors, and have a very handwavy section (sect. 3.4) trying to justify that. This is simply not acceptable. I do not agree that the ambient perturbation experiments are better than regular chamber experiments, but I would agree that they can be just as good. Adding a-pinene to ambient air, and using the oxidants (O3 and OH) and OA seed from ambient air, is not significantly different to injecting O3 in a chamber and using either a sulfate seed or no seed. If the authors expect that the spectra would be significantly different, they should explain why this would be. **But their own Figure 6 indicates that the spectra from the ambient perturbation experiments and a regular chamber are indeed very similar. Why then not use chamber experiments (either from the literature database, or the authors' chamber) to obtain spectra from other precursors and pathways, and see whether they are similar to the ambient LO-OOA or not?**

3) Similar experiments and analysis (VOC addition to ambient air in a flow reactor, followed by spectral comparison) have already been conducted and published by Palm et al. (2018) as part of the GoAmazon campaign, an area comparable to the SE US with high biogenic impact, but also other sources. In that work multiple VOCs were added to ambient air individually, and oxidized to form SOA in an oxidation flow reactor, which was then measured by an HR-AMS. Those authors were able to oxidize b-caryophyllene, longifolene, limonene, b-pinene, α-pinene, toluene, and isoprene, and to obtain yields and HR-AMS mass spectra for all of them. They further state: "The mass spectrum of the SOA formed from OH oxidation [of ambient air] was correlated (R2=0.72-0.93; shown in Fig. S12) with spectra of the SOA formed from the injected VOCs from the standard injection experiments in Sect. 3.4. These correlations show that the SOA formed from OH oxidation of ambient air appeared similar to SOA from known precursors, but **the spectra from the different precursors appear too similar to be able to differentiate the SOA sources in ambient air from the spectrum alone.**" Fig. S12 of that work is reproduced below for reference.

Thus the difficulty of apportioning the SOA in ambient air through this type of experiments, and given the high fragmentation and limited information content of AMS spectra is clear and has been previously documented. If the authors disagree, the burden of proof is on them to show that they can unequivocally associate ambient air spectra with those of specific VOC precursors, including *disproving that ambient LO-OOA may have major contributions from other precursors and pathways.* **Let's imagine that Palm et al. had only conducted experiments with limonene, and the rest of the experiments had failed. Then they would have observed R2 of 0.9 between SOA of those experiments and SOA from ambient air. Would that have been sufficient evidence to justify the title "Large Contributions from limonene to Organic Aerosol in the Amazon"?** That paper performed additional analyses, and concluded instead that biomass burning and anthropogenic precursors were also important contributors at that particular location, in addition to biogenics. Therefore one has to avoid making expansive conclusions based on narrow evidence.

[Figure]

**Fig. S12.** Correlation coefficients between the SOA formed from OH oxidation of ambient air during the wet and dry with the SOA formed when injecting individual VOCs into the OFR. All spectra were calculated as the difference between the average spectrum of OA after OH oxidation in the OFR minus the average spectrum of concurrent ambient OA. For the spectrum of SOA formed form OH oxidation of ambient air, only data in the range of maximum SOA formation with >1 $\mu g\ m^{-3}$ SOA formation were used.

4) Some results of the PMF analysis appear to have been misinterpreted. The ambient data had 10 times as many points in time as the perturbation experiments. In such a situation, the ambient data effectively "dictates" the spectra due to its much higher fractional contribution to the Q value (weighed residual that PMF is minimizing). As expected in this situation (L167) "the perturbation experiments do not create a new factor that does not already exist in the ambient data." Under that setup, **the PMF results for the chamber time periods are similar to multiple linear regression onto the spectra already pre-determined for ambient air (since the chamber results have a low contribution to Q, and then are unable to change the factor spectra much).**

**As the spectra of SOA from the chamber are being forced to be** *represented by a linear combination of a limited set of ambient spectra* **(which they cannot influence in practice), it is normal that some SOA spectra from the chamber VOC additions "project" onto more than one ambient spectrum. That is, if an SOA spectrum produced in the chamber is not exactly similar to one of the ambient spectra, PMF may be able to reduce the residual by representing those experimental spectra as a linear combination of two of the "basis" spectra that were determined primarily from the ambient data.**

**Those results are expected and not surprising. They do not "clearly demonstrate" at all that if a given type of SOA was present in an ambient dataset, it would be split into two factors in the same way. Mathematically these are two very different situations, with very different structures and residual contributions.** For example an ambient cooking OA (COA) factor would have a different time variation,

which PMF would also exploit in extracting the factors, but such structural difference in the variance is not present here. Experiments (for example using simulated data) could be carried out to investigate the interference point for ambient data alone. Unfortunately, the authors misinterpret their results for a very specific PMF situation for non-ambient data, into completely unsupported general statements for PMF analyses of ambient data (that for example ambient COA may contain caryophillene SOA, or that isoprene SOA may have interferences from a-pinene SOA), even though such analyses have not been performed in this work! Perhaps those interferences exist, but they have not been proven by this study. Rather here the results of a complex PMF analysis are being misinterpreted.

In fact, the result that some of the a-pinene and carophyllene SOA apportions to other factors in the authors' methods weakens the main stated conclusions further. Even these types of SOA are not quite well represented by LO-OOA and need to "lean" on other ambient factors to reduce the residual in PMF. Therefore making the conclusion that ambient LO-OOA is mainly from MT & SQT is even more weakly supported.

5) No uncertainty analysis, such as from bootstrapping, is performed for the PMF results. This is more glaring given that very strong conclusions about the identity of a major fraction of the ambient OA, and of potential interferences between factors are made, but we are not shown that the results are even statistically significant or what the uncertainties in the analysis may be.

6) The statements about the novelty of the approach (L19, L21, L87-88, L457-458) are exaggerated. As cited in the paper, both Leungsakul et al. (2005) and Palm et al. (2017) have already published results from very similar experiments. The only difference between the present experiments and those previous ones is (in the authors' words, L22-123) that "no extra oxidant precursors were added into the chamber." That is correct, but **would the authors expect that ambient O3 or OH produced from ambient air would be that different than those produced in other ways? I am sure that the a-pinene molecules don't care about how the O3 or OH colliding with them were formed.** If the authors expected that adding a-pinene to ambient air but using ambient oxidants was going to significantly change the results compared to standard chamber experiments, the reasons for such expectations should be discussed in detail.

Otherwise the experiments are interesting, but follow on prior publications. I mention this because in some places (e.g. abstract L20-25, also in L87-88, L457-458) the statement that the experiments were "novel" is somehow used to prop the weak conclusions. As if somehow the "novel" experiments would have allowed the authors to reach some conclusions that were not reachable by previous authors. **But the novelty is very minor. And the one reason why the experiments are a little different from previous experiments (no added oxidants)** *is the reason why the experiments failed in multiple cases!* **Therefore the "novelty" does not provide any real support for the conclusions.**

7) Statements such as (L316-318) **"The perturbation experiments with other VOCs confirm the stronger ability of α-pinene and β-caryophyllene to produce SOA" are misleading, and frankly just perplexing.** A lot was already known about the relative potential of different VOCs to make SOA before this paper, and

nothing new is learned from the experiments here about this point. The low SOA observed for other precursors is attributed (SI, L405-406 "to the low SOA yields or slow oxidation rates of these VOCs (Ng et al., 2007). Yields for naphthalene are certainly not low, but are higher than for a-pinene (Chan et al., 2009). Later it is acknowledged (SI L412-414) that "isoprene oxidation products which form SOA are mostly second or higher generation products. They are not formed in large amount in the relatively short perturbation experiments (i.e., 40min)." The main issue is that a-pinene has a lifetime of ~1 h under the conditions of the ambient perturbation experiments (using well-known rate constants) and caroyphyllene has an even shorter lifetime. m-xylene has a lifetime of ~10 hrs, and therefore it reacts too slowly under the conditions of these experiments with no added oxidants. And for some precursors (other than isoprene) second or higher generation products are also needed to make SOA, which results in even longer time constants. However, those timescales are readily accessible in the atmosphere.

8) The discussion in the introduction about MO and LO OOA is unclear. Those factors do not represent the same sources or pathways in different studies. In some locations and times they are tied to biomass burning, in others to urban emissions, in others to biogenic SOA, and in others to various combinations. In older studies in which Isoprene-SOA (or "IEPOX-SOA) was not separated, it was necessarily part of MO and LO OOA. Care should be taken to clearly communicate that any interpretations about sources contributing to MO or LO OOA are specific to a given location and time period. And the possibility should be considered (including in the SE US) that different periods may have larger contributions to these factors from different sources, for example BB during one period, pollution during another, and biogenics during another (if the sources impacting a site change substantially in time due to air mass changes or other causes). See e.g. Palm et al. (2018) for an example of such a situation.

Other points

9) L71-72: "The assumption that LO-OOA represents fresh SOA has yet to be directly verified." Fig 2C, 2D, 2E in Jimenez et al. (2009) (where the older terminology SV-OOA was used for LO-OOA), and similar results in other publications, would appear to have directly verified that long ago.

10) A paper with a similar title to this one, but using different lines of evidence, has been recently published (Zhang et al., 2018).

References (not already cited in the paper):

Chan, A. W. H., Kautzman, K. E., Chhabra, P. S., Surratt, J. D., Chan, M. N., Crounse, J. D., Kürten, A., Wennberg, P. O., Flagan, R. C., and Seinfeld, J. H.: Secondary organic aerosol formation from photooxidation of naphthalene and alkylnaphthalenes: implications for oxidation of intermediate volatility organic compounds (IVOCs), Atmos. Chem. Phys., 9, 3049-3060, https://doi.org/10.5194/acp-9-3049-2009, 2009.

Ervens, B., Turpin, B. J., and Weber, R. J.: Secondary organic aerosol formation in cloud droplets and aqueous particles (aqSOA): a review of laboratory, field and model studies, Atmos. Chem. Phys., 11, 11069-11102, https://doi.org/10.5194/acp-11-11069-2011, 2011.

Kiendler-Scharr, A., Q. Zhang, T. Hohaus, E. Kleist, A. Mensah, T.F. Mentel, C. Spindler, R. Uerlings, R. Tillmann, and J. Wildt. Aerosol Mass Spectrometric Features of Biogenic SOA: Observations from a Plant Chamber and in Rural Atmospheric Environments. Environmental Science & Technology, 43(21), 8166-8172, 2009. DOI: 10.1021/es901420b. https://pubs.acs.org/doi/pdf/10.1021/es901420b

Eric Praske, Rasmus V. Otkjær, John D. Crounse, J. Caleb Hethcox, Brian M. Stoltz, Henrik G. Kjaergaard and Paul O. Wennberg. Atmospheric autoxidation is increasingly important in urban and suburban North America. PNAS 2018 January, 115 (1) 64-69. https://doi.org/10.1073/pnas.1715540115

H. Zhang et al., Monoterpenes are the largest source of summertime organic aerosol in the southeastern United States, PNAS 2018. https://doi.org/10.1073/pnas.1717513115

---

## Author Comment (AC1) · 10 May 2018

We thank the reviewers for the detailed and insightful comments. We have addressed the reviewers' comments point by point as indicated below.

**Reviewer #1:**

*This paper shows some interesting results from a novel experiment involving taking ambient air in an urban environment and conducting a chamber experiment after enhancing the concentrations of VOCs. The main emphasis of this paper is the use of α-pinene and β-caryophylene, the characterisation of the SOA produced using aerosol mass spectrometry and inferences are drawn regarding their contribution to particulate concentrations in the region.*

*1. Overall, this is a nice piece of work and well within the journal's remit. However, I do think that the significance is a little overblown in places and the authors need to express more caution in how they interpret some of the results. In spite of their statements otherwise, this is not a true simulation of atmospheric processes (see below), there are a few PMF-specific subtleties that aren't taken account of and how reliably this can be projected onto the wider world is open to question on a number of levels. But in spite of these issues, the conclusions are largely sound and this deserves to get published. I therefore recommend publication with minor revisions.*

Response: We thank the reviewer for the thoughtful insights. To address the reviewer's major concerns, we have toned down the significance of our conclusions and expanded discussions on PMF-specific subtleties in the revised manuscript. We think that the lab-in-the-field experiments can mimic certain atmospheric processes, shed light on understanding the sources of OA factors, and reflect potential issues that PMF analysis encounters. These responses will be discussed in detail when we address the reviewer's specific comments. Meantime, we agree with the reviewer that more studies are warranted to test the reliability of our conclusions to the wider world.

*General Comments:*

*2. This paper makes the assumption of α-pinene and β-caryophylene being representative of monoterpenes and sesquiterpenes respectively. While these are common assumptions made in the community and the VOCs are both very well studied, their overall representativeness is in question because the level of oxidation in SOA from different precursors are known to vary substantially between compounds (Alfarra et al., Atmos. Chem. Phys., 13, 11769-11789, 10.5194/acp-13-*

*11769-2013, 2013). This is especially true of the sesquiterpenes, as difficulties in working with a*

*number of these compounds means that we lack data on a large subset of these. This should be*

*discussed and any evidence to support this assumption properly cited.*

Response: In this study, we selected α-pinene and β-caryophyllene as representatives due to the following reasons. Firstly, both VOCs are widely studied in the literature. Secondly, they are the most abundant species in monoterpenes and sesquiterpenes, respectively (Guenther et al., 2012;

Helmig et al., 2007). Thirdly, the mass spectra of SOA from VOCs in the same class generally share similar features. For example, the correlation coefficient (i.e., R) between the mass spectra of SOA from the β-caryophyllene and α-humulene is 0.97 (Bahreini et al., 2005). Still using the mass spectra reported in Bahreini et al. (2005), the R between α-pinene SOA and other monoterpenes SOA (β-pinene, α-terpinene, myrcene, and terpinolene) is larger than 0.9. Fourthly, in addition to the similar mass spectra, the time series of α-pinene in the southeastern U.S. is similar to that of other monoterpenes, such as β-pinene and camphene (Xu et al., 2015a). The above reasons have been added in the revised SI. We also add a caveat that future studies using other monoterpenes and sesquiterpenes are still in need.

We agree with the reviewer that many properties of SOA (i.e., yield, hygroscropicity, etc)

from different monoterpenes or sesquiterpenes are different. To be precise, we have replaced

"representative" with "important" in some sentences in the revised manuscript.

*3. Throughout the manuscript, there is a general tendency to treat LO-OOA and MO-OOA as*

*defined chemical entities, whereas the truth is that these represent reductions of highly complex*

*chemical systems and the exact factors reported are known to vary dataset to dataset. While*

*common trends have been noted in terms of behaviour and mass spectral profile, their exact mass*

*spectral nature depends on the measurement location and season and in many cases (particular*

*in the wintertime), PMF will fail to separate them at all, instead returning a single OOA factor.*

*This must be discussed in a meaningful manner in the introduction and discussion because it adds*

*an extra layer of nuance to the results.*

Response: We agree with the reviewer that OA factors from PMF analysis represent complex chemical systems. Understand what these OA factors represent is one of many motivations of this study. In the revised manuscript, we have modified the language to avoid the confusion to treat

OA factors as defined chemical entities. We also agree with the reviewer that the exact mass spectral nature of OA factors varies with locations. We would like to point out that the mass spectra of LO-OOA are highly similar (R > 0.95) across all the seven datasets in our study (Fig. R1). This is one of many evidence to suggest that LO-OOA at different datasets may have similar sources.

Regardless, we clearly limit our conclusions to the southeastern U.S.

[Figure]

Fig. R1. The correlation plot between the mass spectrum of LO-OOA for 2016 rooflab perturbation study and the LO-OOA obtained in other six ambient datasets in the southeastern U.S.

In the introduction and implications sections of the revised manuscript, we have added the discussion that a single OOA factor is resolved in some cases, especially in winter.

*4. Related to the above point, there seems to be a general assumption that PMF had adequately*

*accounted for the new SOA being formed, but in my mind, the decrease in MO-OOA in response*

*to the β-caryophylene experiments in particular raises a number of red flags because this implies*

*that the data model didn't hold and the factorization wasn't sound. The authors need to pay much*

*closer attention to whether the factorisations 'worked' or not; a good starting point would be*

*inspecting the residuals (e.g. Q/Qexp) as a function of time over the course of the experiments and*

*if they positively correlate at all with the amount of additional SOA, this would imply that PMF*

*has failed to capture the chemistry perfectly.*

Response: We agree with the reviewer that the decrease in MO-OOA after β-caryophyllene injection raises a red flag and implies that factorization is not perfect. We have examined the PMF

residual (i.e., Q/Q$_{exp}$) in α-pinene and β-caryophyllene perturbation experiments. As shown in Fig.

R2a, in α-pinene experiments, the difference in Q/Q$_{exp}$ between "Chamber_Bf" (before α-pinene injection) and "Chamber_Af" (after α-pinene injection) is not statistically significant. This suggests that PMF analysis has adequately accounted for the newly formed α-pinene SOA. This is consistent with the observation that in α-pinene experiments, no OA factors show unexpected dramatic decrease after α-pinene injection.

In contrast, in β-caryophyllene experiments, there is a clear pattern that Q/Q$_{exp}$ in

"Chamber_Af" is larger than that in "Chamber_Bf" (Fig. R2b). This is likely because of the rapid change in the subtleties of OA composition caused by the newly formed β-caryophyllene. However, the behavior of Q/Q$_{exp}$ is not quite expected because the OA mass spectra after injecting β- caryophyllene are almost identical to those before perturbation (i.e., R between Chamber_Bf and

Chamber_Af is >0.99 as shown in Fig. R3b). The larger Q/Q$_{exp}$ in β-caryophyllene experiments than α-pinene experiments may be a result of that ΔOA (i.e., the difference in OA concentration between "Chamber_Af" and "Chamber_Bf") is larger in β-caryophyllene experiments (average value 1.95 µg m$^{-3}$ over 6 experiments) than α-pinene experiments (average value 0.98 µg m$^{-3}$ over

14 experiments). Regardless, under the simulated circumstances, PMF analysis cannot adequately capture the newly formed β-caryophyllene SOA.

We would like to clarify that our conclusions are not dependent on if PMF has perfectly accounted for the newly formed SOA, mainly because similar issues could also happen in the analysis of ambient data. The β-caryophyllene perturbation experiments simulate a scenario when there is a sudden change in the OA composition caused by an airmass encountering a plume or change in OA sources due to shift in wind direction. Under these circumstances, PMF analysis may have difficulty in accurately apportioning the OA sources. The simulated scenarios and the observed PMF issues have been observed in previous studies. For example, in the Figure 9 of Sun et al. (2016), an increase of hydrocarbon-like OA (HOA) is usually accompanied by a decrease in cooking OA (COA) and verse vice. Another example is that figure 5 of Reyes-Villegas et al. (2018)

showed that a biomass burning plume leads to unexpected rapid variations in the concentration of many OA factors on the time scale of minutes. Last example is that in the figure S10 of Milic et al. (2017), the PMF residual during a biomass burning plume is orders of magnitude higher than other sampling periods. Emphasizing the limitations of PMF analysis is one goal of this study.

[Figure]

Fig. R2. The PMF residual ($Q/Q_{exp}$) during "Chamber_Bf" and "Chamber_Af" periods for (a) α- pinene and (b) β-caryophyllene perturbation experiments.

[Figure]

Fig. R3. The correlation coefficient by comparing the OA mass spectra between "Chamber_Bf"

and "Chamber_Af" in (a) α-pinene and (b) β-caryophyllene perturbation experiments.

*5. The modelling work presented in section 3.6 left me slightly confused as to what the authors*
*were trying to achieve and how. The text in the main article would suggest that an updated scheme*
*is being compared with a very old one, but the supplement seems to say that specific mechanisms*
*have been added here. This must be clarified.*

Response: We use the modelling work as one of the evidence to support our hypothesis that the
major source of LO-OOA in this region is the oxidation of MT and SQT by various oxidants. In
section 2.6, we described two simulations (i.e. "default simulation" and "updated simulation") with
different organic aerosol treatment. Compared to "default simulation", we incorporate two recent
findings in the "updated simulation". Firstly, we implemented $MT+NO_3$ chemistry to explicitly
account for semivolatile organic nitrate compounds that contribute to SOA. Secondly, we
improved the parameterization of SOA formation from $MT+O_3/OH$ based on a recent study by
Saha and Grieshop (2016). In section 3.5 of revised manuscript, we pointed out that $SOA_{MT+SQT}$
in the updated simulation reasonably reproduces the magnitude and diurnal variability of LO-OOA
for each site. The model bias is within ~20%. The reasonable agreement between modeled
$SOA_{MT+SQT}$ and LO-OOA at multiple sites and in different seasons serves as one piece of evidence
to support our hypothesis that LO-OOA largely arises from the oxidation of MT and SQT in the
southeastern U.S.

*6. Also, as pointed out later in this section, this work does not directly preclude that other*
*precursors may be contributing and the discussion dealing with this relies heavily on inferences*
*drawn from the literature, so this work isn't really that dramatic a result in how it is presented*
*now. I would suggest a more defined modelling experiment is constructed around a clear working*
*hypothesis. This could just be a case of making the work shown here clearer and moving material*
*from the supplement to the main article.*

*More generally, I noted a very odd tendency to leave certain pertinent (and in many cases*
*interesting) details in the supplement that maybe should have been given more prominence or at*
*least linked to the main article better. For example, the box modelling described in section S6 was*
*very interesting, but it wasn't clear at all how this fit into the narrative of the main article. I also*
*had a hard time reconciling the information about the CMAQ runs in the main article and the*
*supplement as well (see above). I would revise what information goes where, using the main article*

*for the discussions relating to the scientific arguments and making sure the material in the*
*supplement is purely technical detail in support of this.*

Response: We have toned down the significance of conclusions in the revised manuscript. We
have also re-organized the main text and SI, and improved the linkage between two parts. Many
details, although interesting, are kept in the Appendix or SI, because we prefer to focus on the
main message and keep the logic of manuscript easy to follow.

*Specific comments:*

*7. Line 95: When saying 'representative urban', please be specific about what type of urban site*
*this (e.g. background) and how you qualify this statement.*

Response: We have deleted the word "representative".

*8. Line 115: What counts as 'too much' SOA and why?*

Response: As discussed in response to reviewer's comment #4, when there is a sudden change in
the OA composition, PMF analysis may have difficulty in accurately apportioning the OA sources,
as shown in β-caryophellene perturbation experiments and previous studies. This is why we tried
to avoid producing too much SOA in the perturbation experiments.

It is challenging to quantify what counts as "too much". As shown in β-caryophellene
perturbation experiments, even though (1) the newly formed β-caryophyllene SOA only increases
the OA concentration by roughly 2 μg m$^{-3}$ and (2) the OA mass spectra after injecting β-
caryophyllene are almost identical to those before perturbation (i.e., R >0.99 as shown in Fig. R3b),
PMF analysis encounters difficulty in accurately apportioning the OA sources.

*9. Line 265: It has long been shown that α-pinene SOA produced in chambers produces a mass*
*spectrum that is similar to LO-OOA and given that this mass spectral profile is also seen in the*
*presence of strong emitters of this VOC (e.g. temperate and boreal forests), the case for*
*α-pinene SOA being a strong contributor to LO-OOA has never really been in doubt in this*
*reviewer's opinion. Why is the evidence presented here any more 'direct' than those published*
*previous? While the perturbation experiment does indeed produce LO-OOA as retrieved using*
*PMF, this retrieval is still based solely on mass spectral similarity, so I would contend that this*
*does not really present any new evidence to this effect.*

Response: We thank the reviewer for pointing this issue out. The "directness" of our conclusions is not clearly communicated in the original manuscript and caused some confusions. While we acknowledge that the mass spectra comparison approach largely improves our understanding of

OA factors, we think that the perturbation experimental approach provides more objective and quantitative conclusions by addressing some limitations of the mass spectra comparison approach.

The mass spectra comparison approach has the following limitations. Firstly, the similarity between two mass spectra is a subjective determination. In other words, what correlation coefficient ® value implies SOA from a certain source contributes to one OA factor? For example, the R values between laboratory-generated α-pinene SOA (using HONO as OH source) with LO-

OOA, isoprene-OA, and MO-OOA in this study are 0.96, 0.88, and 0.81, respectively. Using these

R values to imply whether α-pinene SOA contributes to a certain OA factor or not is subjective.

As another example, Jimenez et al. (2009) showed that the mass spectrum of α-pinene SOA

becomes more similar to that of MO-OOA (i.e., LV-OOA in older study) than that of LO-OOA

(i.e., SV-OOA in older study) with photochemical aging. The ability to determine when and how much α-pinene SOA is apportioned to MO-OOA based on an R value is subjective. Secondly, the conclusions from mass spectra comparison approach are qualitative. Even if the mass spectrum of

α-pinene SOA is the most similar to LO-OOA, this similarity does not guarantee that all α-pinene

SOA is apportioned into LO-OOA and this similarity does not provide information regarding what fraction of α-pinene SOA is apportioned into LO-OOA.

The perturbation experiments could address the limitations of mass spectra comparison approach and provide more objective and quantitative conclusions. Firstly, the perturbation experiments simulate a short period of time with increasing α-pinene SOA concentration. we perform PMF analysis on the combined ambient data and perturbation data. PMF analysis does not distinguish SOA from natural α-pinene vs. from injected α-pinene, so that PMF analysis can objectively apportion α-pinene SOA into factors. Thus, the conclusions from the perturbation experiments are directly drawn without any subjective judgement on the similarity in mass spectra.

Secondly, using the perturbation data, we attempt to quantify the fraction of fresh α-pinene SOA

that is apportioned into different factors (i.e., ~80% into LO-OOA, ~20% into isoprene-OA, 0%

into MO-OOA, COA, and HOA). Although further studies are required to extrapolate the conclusions from perturbation experiments to real atmosphere, a similar quantitative understanding cannot be obtained from simple mass spectra comparison approach. Thirdly, the perturbation experiments have the potential to utilize subtle differences across the entire the mass spectrum to evaluate the sources of OA factors. Based on previous laboratory study, the mass spectrum of α-pinene SOA is highly correlated (R = 0.97) with that of β-caryophyllene SOA

(Bahreini et al., 2005). Using a mass spectra comparison approach would suggest that these mass spectra are too similar to be differentiated by PMF analysis. However, perturbation experiments show different behaviors of α-pinene SOA and β-caryophyllene SOA. That is, a fraction of the fresh β-caryophyllene SOA is apportioned into COA factor, but similar behavior is not observed for α-pinene SOA. The different behaviors are likely due to the subtle differences in their mass spectra. For example, $f_{55}$ (i.e., the ratio of $m/z$ 55 to total signal in the mass spectrum) is typically higher in β-caryophyllene SOA than α-pinene SOA (Bahreini et al., 2005; Tasoglou and Pandis,

2015), and the mass spectrum of COA is characterized by prominent signal at $m/z$ 55 (Fig. 2).

Overall, the perturbation experiments provide more objective and quantitative insights into the sources of OA factors than traditional mass spectra comparison approach.

We have added the above discussions in the revised manuscript.

*10. Line 280: The fact that the oxidation rate of VOCs is dependent on oxidant concentration is*

*very well established in kinetics. The discussion regarding this observation would be considered*

*pointing out the obvious to many. It would be far more useful if a quantitative relationship with*

*ozone concentration could be reported here.*

Response: In Line 280, we are not making the conclusion that the oxidation rate of VOCs depends on oxidant concentration. Instead, we use this well-established conclusion to explain our observation that the LO-OOA enhancement amount correlates with oxidant concentration. As the

SOA formation depends on a number of variables, including temperature, relative humidity, the concentrations of oxidants, $NO_x$, and existing particles, etc, the VOCs oxidation rate is not equivalent to SOA formation amount. In this study, the α-pinene perturbation experiments were conducted at different times of day (i.e., from 9am to 9pm) and under different conditions (i.e., T,

RH, oxidant, $NO_x$, etc). Despite the large difference in reaction conditions, the LO-OOA

enhancement amount correlates well with $O_3$ concentration, suggesting that the oxidant concentration plays a more controlling role in the amount of OA formed in α-pinene experiments than other reaction variables do.

As a side note, we also want to point out that the oxidant concentration is not the sole
variable in determining the SOA formation from β-caryophyllene, which is in contrast to α-pinene
experiments. As shown in Fig. S22 in revised SI, the LO-OOA enhancement amount in β-
caryophyllene perturbation experiments is greatly affected by $NO_2$ level. For example, for two
experiments with similar $O_3$ concentration and injection time, more LO-OOA is formed in the
experiment with a lower $NO_2$ level (Fig. S22f in revised SI).

*11. Line 286: An alternative explanation here is that the experimental set-up here was not*
*conducive to HOM formation for whatever reason. This should be added as a caveat.*

Response: We agree with the reviewer and have added this caveat in the revised manuscript.

*12. Line 309: There is a major problem with this statement; the results indicate that the β-*
*caryophylene SOA spectrum to be represented by PMF as a combination of the LOOOA and COA*
*mass spectra, but it would be a mistake to imply in any way that it is producing two 'types' of OA*
*(this is clarified later in the manuscript but it is ambiguous here). Issues about the quality of the*
*PMF retrieval aside (see above), in the hypothetical situation that there is an environment with a*
*mixture of cooking and biogenic SOA, PMF will likely still separate these because it determines*
*factors not just by mass spectral profile but by temporal profile, so would still return factors*
*corresponding to cooking and an average of biogenic SOA from all sources. The only situation I*
*could think where this would be a problem is if monoterpene and sesquiterpene SOA formation*
*were not well matched temporally, in which case I could see how the COA-like component of*
*sesquiterpene SOA would manifest as 'mixing' between the cooking and biogenic SOA factors, but*
*this would be evident in the temporal profiles.*

Response: We agree with the reviewer that PMF separates factors based on features in both mass
spectrum and time series. However, we note that the temporal variations of COA and SOA from
monoterpenes and sesquiterpenes do not guarantee a clean separation between different sources.
For example, the concentrations of many oxidation products of monoterpenes (e.g. pinonic acid,
pinic acid, etc), COA, and LO-OOA show similar increasing trend near sunset (Allan et al., 2010;
Xu et al., 2015b; Zhang et al., 2018). The emissions of sesquiterpenes and diurnal variations of
sesquiterpene SOA are not well-studied. Thus, it is possible that some sesquiterpene SOA is
apportioned into COA.

As mentioned in the manuscript, we agree with that COA is mainly from cooking emissions.

However, the perturbation experiments show the possibility that COA could include β- caryophyllene SOA. There is no evidence in the literature to support that cooking OA factor is purely from cooking and does not have any biogenic SOA.

*13. Line 350: How much more 'realistic' is this? While this would give a more life-like oxidant*

*and NOx background, given that the chamber walls will act as a sink of VOCs, radicals and*

*particles, I would still expect that the precursor perturbations would have to be higher than typical*

*atmospheric concentrations to achieve realistic SOA concentrations and consequently have a*

*higher VOC:NOx ratio. This must be discussed in an objective manner and while some of this is*

*touched on in the supplement, it's kind of glossed over in the main article.*

Response: We would like to clarify that we did not want to produce realistic SOA concentration.

Instead, we aimed to produce small amount of SOA, which would not significantly perturb the composition of existing organic aerosol. Based on the chamber volume and injected α-pinene volume, we injected about 14ppb α-pinene. Most of injected α-pinene is carried out of the chamber by dilution air and we estimated that only 2-5 ppb α-pinene is reacted in the chamber. We have added this information in the main text.

*14. Figure 6: The caption of this figure is excessively long.*

Response: We prefer to keeping the caption descriptive of the figure, such that the readers could comprehend the figure after reading the caption.

*15. Line S477: This doesn't make sense. Why would the solver reduce the concentration of MO-*

*OOA because it had been added to? I find it more likely that there was a breakdown in the data*

*model and mass was being erroneously rotated out of the factor. This is undesirable, but also feeds*

*into the discussion above regarding the relationship with COA.*

Response: As stated in the SI, the reason why MO-OOA decreases after adding β-caryophyllene is unknown. One possible explanation is that since the mass spectrum of β-caryophyllene SOA is more similar to LO-OOA than MO-OOA, PMF solver somehow decreases the concentration of

MO-OOA to reduce the overall fitting residual. We agree with the reviewer that this result is undesirable. However, as pointed out in response to reviewer's comment #4, similar situation could happen in ambient measurements. This issue deserves more attention.

*16. Line S480: What other studies?*

Response: Please see response to comment#4.

*Technical comments:*

*17. Line 113: Please be more specific over which VOCs are anthropogenic vs biogenic. The word*
*'respectively' does not work when four are listed.*

Response: We have deleted the word "respectively".

*18. Line 179: Why not saturated fatty acids?*

Response: We have deleted the word "unsaturated", so that the "fatty acids" in the revised sentence
includes both unsaturated and saturated compounds. We have also cited Allan et al. (2010) in this
sentence.

*19. Line 119: Correct 'concentration' to 'concentrations'.*

Response: We have made this correction in the revised manuscript.

*20. Line 184: What are the oxidation states in each instance?*

Response: We have added the oxidation states (i.e., -0.70 to -0.34 for LO-OOA and from -0.18 to
0.71 for MO-OOA in the southeastern U.S.) in the revised manuscript.

*21. Section 2.2: Please specify the materials used for the aerosol and gas sampling lines.*

Response: Teflon tubing and stainless steel tubings were used for gas and aerosol sampling lines,
respectively.

**Reviewer #2:**

*The manuscript presented by Xu et al. proposes an interesting study on the contribution of the*

*oxidation of alpha-pinene and caryophyllene to the SOA mass observed in the S.E.-US. The*

*characterization of SOA generated in the lab-in-the-field smog chamber was performed using an*

*aerosol mass spectrometer. Overall the work performed in this study is good and fall within the*

*scope of the journal. However, I think the conclusions proposed from the PMF analysis/chamber*

*experiments are not always well sustained and more caution should be taken when extrapolating*

*the results.*

Response: We thank the reviewer for the positive comments. In the revised manuscript, we have added more caveats regarding our conclusions drawn from the perturbation experiments.

*General comments*

*1. The authors should carefully review their paper and avoid the repetition between the main text*

*and the SI. At many places, sentences are duplicated and are not useful. However, some important*

*details are left within the SI and should be moved to the main manuscript.*

Response: We have re-organized the main text and SI, and improved the linkage between two parts.

*2. The authors should provide more information in the PMF analysis and provide the elementary*

*checks to validate their analysis. For instance, it is a bit surprising that the factors don't change*

*throughout the experiments (i.e. bf vs af) while significant perturbation has been made to the*

*system. Or do the authors consider/claim that most of the SOA sampled in the ambient are formed*

*from the oxidation of alpha-pinene or caryophyllene? In addition, we could expect that the fresh*

*LO-OOA (formed within a few minutes, without lights) would have different signatures that LO-*

*OAA formed in the atmosphere (aged SOA, formed from different chemistry, ...). How do the*

*factors correlate throughout the experiments: e.g. LO-OOA_Amb_Bf vs LO-OOA_Chamber_Af?*

Response: We respond to the reviewer's comment by addressing some confusions regarding PMF

analysis. Firstly, we performed PMF analysis on the combined ambient and perturbation data. Each

OA factor has a constant mass spectrum throughout the study, regardless of ambient or chamber periods. In other words, the mass spectra of factors do not change between "Chamber_Bf" and

"Chamber_Af". Secondly, we did not introduce significant perturbation to the system. In contrast, our goal is to produce small amount of SOA, which would not cause dramatic change in the OA

composition and would not bias PMF analysis. For example, the difference in average OA concentration between "Chamber_Bf" and "Chamber_Af" is within 4 μg m$^{-3}$ for all perturbation experiments. Thirdly, we did not consider/claim that most of the SOA sampled in the ambient are formed from the oxidation of α-pinene or β-caryophyllene. This assumption is not related to our conclusions.

Regarding the last question in this comment, we believe that the reviewer suggests to compare the OA mass spectra between "Chamber_Bf (before VOC injection)" and "Chamber_Af (after VOC injection)". As shown in Fig. R3 (in response to reviewer#1), the mass spectra between two periods are almost identical, with R larger than 0.99 for all experiments. This is desired because we do not intend to introduce significant change in the OA composition after perturbation.

*3. How do the identified factors correlate with the reference MS? How do the residuals evolve throughout an experiment? How does alpha-pinene-derived LO-OOA correlate with caryophyllene derived LO-OOA? Overall, the authors should provide more statistical analyses in order to give a robust validation of the analysis.*

Response: In this study, we resolve and evaluate PMF factors according to the standard procedure outlined in Zhang et al. (2011). The detailed description on OA factors and justification of PMF results have discussed in section S3 of SI. As discussed in the SI, the PMF factors have the same features as those in the literature. More importantly, the PMF results in this study are consistent with our previous measurements (from 2012 to 2015) at the same site and in similar seasons, as shown in Fig. S14 in revised SI. The mass spectra of LO-OOA are similar (R > 0.95) across all the seven datasets in this study (Fig. R1 in response to reviewer#1's comment#3).

In α-pinene perturbation experiments, the PMF residual is not significantly different between "Chamber_Bf" and "Chamber_Af", suggesting that PMF analysis has adequately accounted for the newly formed α-pinene SOA. More detailed discussions can be found in response to reviewer#1's comment #4.

As noted in response to the reviewer's comment#2, the mass spectrum of LO-OOA is the same throughout the study, regardless of ambient vs. perturbation and regardless of α-pinene vs. β-caryophyllene experiments.

*4. The authors should report the concentration of the inorganics in their experiments and in case*
*of significant concentrations of sulfate estimate the aerosol acidity. Indeed, the presence of acidic*
*aerosols can lead to multiphase reactions (e.g. reactive uptake of IEPOX) that could greatly*
*impact the SOA composition. In addition, an estimation (modeling?) of the concentrations of other*
*VOCs would be interesting (especially isoprene). Ozonolysis of alpha-pinene leads to the*
*formation of OH radicals, which could further react and oxidize alpha-pinene but also other VOCs*
*present in the ambient air. The authors should discuss this possibility and provide more*
*information in the background of the chamber/ambient air. As it is, the conclusions proposed in*
*the paper on the potential increase of the IEPOX-OA or COA factors from the oxidation of alpha-*
*pinene and caryophyllene, respectively appear speculative (correlations are not sufficient to*
*validate such trend: r~0.5). For instance, the authors could estimate the amount of IEPOX (thus*
*isoprene) formed in the chamber to explain the formation of IEPOX-OA and check if the numbers*
*make sense or not.*

Response: We interpret that the reviewer's key question as the following: "is the enhancement in
isoprene-OA factor due to the oxidation of isoprene in the chamber after injecting α-pinene?" The
most direct evidence to rule out this hypothesis is that the concentration of IEPOX+ISOPOOH
($C_5H_{10}O_3 \cdot I^-$) and isoprene hydroxyl nitrates ($C_5H_9NO_4 \cdot I^-$), measured by $I^-$ HR-ToF-CIMS, did not
change after α-pinene injection (Fig. S3b in revised SI). This suggests that the α-pinene injection
does not introduce isoprene oxidation in the chamber.

The relatively weak correlation between Δisoprene-OA and ΔLO-OOA (where the Δ
indicates the difference in concentration between the "Chamber_Af" and "Chamber_Bf") across
all α-pinene perturbation experiments is not contradictory to the conclusion that α-pinene SOA
influences isoprene-OA factor. The weak correlation could be because α-pinene SOA in different
perturbation experiments were formed under different conditions (e.g., $NO_x$) and had different
mass spectra (Fig. 7 in revised manuscript). Thus, the fraction of α-pinene SOA apportioned into
isoprene-OA factor varies with experiments and results in the weak correlation. However, we
would like to point out that although the correlation between Δisoprene-OA and ΔLO-OOA across
all α-pinene perturbation experiments is relatively weak, the time series of isoprene-OA and LO-
OOA in the same α-pinene perturbation experiment are strongly correlated. The R is 0.88 for the
α-pinene perturbation experiment on 07/20 (Fig. R4a). It is well studied that isoprene produces

SOA slower than α-pinene, as isoprene SOA involves higher-generation products. If the
enhancement in isoprene-OA factor is due to isoprene oxidation, the enhancement is expected to
occur later than the enhancement in LO-OOA, but it is not observed in the experiments. Thus, the
strong correlation between isoprene-OA and LO-OOA in the same α-pinene perturbation
experiment serves as another evidence that the enhancement in isoprene-OA factor is due to
interference from newly formed α-pinene SOA, rather than oxidation of isoprene after injecting α-
pinene.

[Figure]

Fig. R4. (a) The correlation between isoprene-OA and LO-OOA in the "Chamber_Af" period of
one α-pinene perturbation experiment (i.e., ap_0720_2). (b) The correlation between COA and
LO-OOA in the "Chamber_Af" period of one β-caryophyllene experiment (i.e., ca_0726).

At the reviewer requested, the concentrations of inorganic species have been added into
the revised SI.

*Specific comments:*

*5. Lines 104:111. Did the authors characterize the chamber? Mixing, wall losses,*

Response: Since our goal is to qualitatively understand which OA factors the α-pinene SOA is
apportioned into, we did not characterize the mixing and wall loss of the chamber. We would like
to note that because of the continuous exchange air between chamber and ambient air, the particle
wall loss is difficult to characterize. The chamber characterization will be one focus of future work.

*6. Lines 141: The authors claim that by having an overflow, it suppressed the particle loss. Did*
*they mean reduce? Have you done some tests to validate such statement?*

Response: We used a bypass flow to reduce the particle loss in sampling line. We have replaced
"suppressed" with "reduced".

*7. Lines 217-220: Why not using the outdoor chamber to do such experiments? Can the authors*
*discuss the strategy here?*

Response: In the laboratory studies, we follow traditional chamber experimental procedure
produce α-pinene SOA under controlled conditions. Then we compare laboratory experiments with
lab-in-the-field experiments to evaluate the representativeness of laboratory studies.

*8. Lines 266-268: The decay of LO-OOA is quite fast and I do not think it can only explain by the*
*dilution and or dead-volume. The residence time in the chamber is ~100 min. Where were located*
*the sampling inlets?*

Response: The ~100min residence time is calculated with the assumption of no dead volume. The
existence of dead volume would largely decrease the residence time and change the decay rate of
LO-OOA, as shown in Fig. S21 in revised SI.

The sampling inlets were inserted into the center of the chamber, which has been specified
in the revised SI.

*9. Lines 278:284: It is quite expected. What is the point of the authors?*

Response: Please see response to reviewer#1's comment#10.

*10. SI Line 150:157: These results are a bit intriguing. The data reported for the boreal forest do*
*not exhibit prominent ions at m/z 53 or 82. The authors suggest that alpha-pinene/monoterpene*
*can contribute to IEPOX-OA but according to Fig S7 the correlation is far to be obvious strong.*
*The authors should compare the MS obtained in their study with other PMF data obtained from*
*monoterpene-dominated areas (e.g. boreal forest).*

Response: In SI Line 150-157, we suggest that monoterpenes SOA may influence isoprene-OA
factor, if the isoprene-OA factor is present. In other words, for a location without isoprene-OA
factor, the influence of monoterpenes SOA on isoprene-OA does not exist. Regarding the
reviewer's comment on Fig. S7 (i.e., Fig S5 in revised SI), we have addressed it in response to
comment#4. As the reviewer suggest, we compared the mass spectrum of LO-OOA in this study
with that obtained in a coniferous forest mountain region in Whistler, British Columbia, Canada
(Lee et al., 2016). The correlation coefficient is 0.99.

**Reviewer #3:**

*This paper presents results from experiments and model runs focusing on the monoterpene*
*contribution to biogenic SOA in the SE US. A small Teflon reactor was used to oxidize ambient*
*air to which single VOC precursor was added. Based on simple PMF analysis and simple CMAQ*
*model runs, it is concluded that monoterpenes are major contributors to ambient OA in the SE US.*
*The authors are wellknown in the field and have published much excellent work, the paper falls*
*within the scope of ACP, and has some interesting aspects. However, in my opinion the new*
*evidence is weak, partially supported with circular logic, and is very overinterpreted. The new*
*evidence is very insufficient to support the very strong conclusions. I don't see how this paper can*
*be published in ACP in anywhere near its present form. I recommend that the authors go back to*
*the drawing board and summarize the new experimental aspects into a paper whose conclusions*
*are actually supported by the evidence presented. For example, the results on Appendix B seem*
*more novel to this reviewer than the ones that are described in the main paper.*

*Note that I made this recommendation already in the access review, with the concurrence of the*
*previous Editor, and hoping to avoid having to post this review in public. However, after an appeal*
*by the authors, it was decided to publish the paper in ACPD anyway without significant revisions.*

Response: We thank the reviewer for detailed comments in both access review and ACPD stages.
We appealed the reviewer's suggestion in the access review stage because we respectfully disagree
with many of the reviewer's comments. We would like to use the open discussion to clarify a
number of issues.

*1. Brief statement of the major issue*

*1) The main problem of this paper is that the evidence presented does not support the conclusions.*
*The conclusions are summarized in the paper title "Large Contributions from Biogenic*
*Monoterpenes and Sesquiterpenes to Organic Aerosol in the Southeastern United States." Or L80-*
*84: "We provide direct evidence that newly formed SOA from α-pinene [...] and β-caryophyllene*
*(representative sesquiterpene) dominantly contributes to LO-OOA in the southeastern U.S."*

*The new evidence presented in this manuscript has two parts:*

*1.a) Some interesting, but incomplete, experiments with an ambient reactor, that have been*
*analyzed using PMF. What the authors have really shown with these experiments is (in their own*

*words) "to qualitatively understand which OA factors the α-pinene SOA is apportioned into"*

*(Supp. Info. L349-350). Most importantly, experiments with other precursors failed, and*

*experiments using for example cloud pathways were not even attempted. It is highly likely (based*

*on prior published work) that spectra from other precursors and pathways would also have looked*

*similar to LO-OOA, so it is very problematic that those experiments failed or where not even*

*attempted.*

*The authors do acknowledge (L415) that based on the experimental work alone "we do not*

*conclude that LO-OOA arises exclusively from MT and SQT." Next, they do acknowledge that*

*their title is just a hypothesis, after all the experimental evidence has been considered: (L387-391):*

*"we propose that the major source of OOA in this region is the oxidation of MT and SQT by*

*various oxidants (O3, OH, and NO3). To test this hypothesis, we use CMAQ to simulate pollutant*

*concentrations across the southeastern U.S."*

*So given how weak the evidence from (1.a) is in terms of supporting the paper's conclusions, how*

*strong is the complementary evidence from the model?*

*1.b) The CMAQ model is run with a different parameterization for terpene SOA that has higher*

*yields than a very old one. Not surprisingly, the terpene contribution to SOA in the SE US goes up*

*in CMAQ. I understand that the parameterization is in principle improved compared to some older*

*ones, but how do we know it is really accurate? The parameterization is still very simple, and does*

*not included detailed chemistry such as a HOM mechanism. The justification about why this*

*parameterization would be accurate when implemented in as complex a region as the SE US is*

*very light and not satisfactory (sect. 2.6). I agree the model is "upgraded" (L25), but not that it*

*has been shown to be accurate. There is a long history of simple parameterizations (after various*

*"upgrades") being wrong when compared to ambient air. I do not see sufficient evidence that*

*supports the accuracy of these model runs, so that strong conclusions about MT + SQT*

*contributions (that could not be reached with the experiments alone) could now be reached with*

*certainty.*

*In addition, the use of the CMAQ results suffers from circular logic. Section 1.a. ended with a*

*hypothesis that LO-OOA was arising from MT & SQT. Now the fact that the CMAQ results are of*

*the same order as LO-OOA is used as confirmation that indeed LO-OOA comes from MT & SQT.*

*But we did not know that LO-OOA came from MT & SQT yet, that was only a hypothesis. The logical structure here is therefore problematic.*

*Other evidence from the literature is mentioned, such as high emissions of MT & SQT in the SE US, and the fact that some previous results suggest that anthropogenic SOA is not a major contributor in this region. The first one has been known for decades. The second result is not quite consistent with some modeling studies and measurements of fossil carbon (when taking into account that urban pollution has a significant fraction of non-fossil carbon, see Kim et al. (2015) and references therein). Together, the combined evidence is still not sufficient for the sweeping conclusions.*

*Ten years ago we didn't know about the IEPOX-SOA pathway (Paulot et al., Science 2009), that has since proven to be a major contributor to SOA over the SE US. A paper written in 2008 may have used the existing literature to conclude that isoprene was a minor contributor to SOA in the SE US, and would have been sorely wrong in doing so. We also didn't know till very recently about autoxidation being important in the atmosphere (Ehn et al., 2014). The importance of autoxidation for urban emissions in the US has recently been demonstrated (Praske et al., PNAS 2018), and is not included in the CMAQ runs or literature studies cited here. SOA formation in clouds is also highly uncertain (Ervens et al., 2011), and could also lead to LO-OOA through various mixtures of precursors and pathways (which remain almost completely unexplored). For example perhaps there is an isoprene SOA pathway through clouds that has not been discovered yet and that contributes half of the LO-OOA in the SE US. The present paper is implicitly saying that other precursors and pathways are not important in the SE US, and risks shutting down research on other alternatives. Is that justified? In my opinion it is not, and the evidence is this paper is not anywhere near sufficient to justify its title and sweeping conclusions.*

Response: We think the reviewer over-states the "weakness" of our evidence and the "strongness" of our conclusion. In many places of reviewer's comments, the reviewer indicates "the authors draw the conclusion that LO-OOA is exclusively monoterpenes SOA based on a single evidence" and that is a significant shortcoming. Our hypothesis is that "the major source of LO-OOA in the southeastern U.S. is the fresh SOA from the oxidation of monoterpenes (MT) and sesquiterpenes (SQT) by various oxidants ($O_3$, OH, and $NO_3$)". We never argue that LO-OOA is exclusively MT and SQT SOA and never state that SOA from other sources/pathways is not important. While monoterpenes have been recognized an important SOA source for some time, until Zhang et al.

(2018), there was no evidence for them being a contributor on the order of half of the ambient OA.

Even with Zhang et al. (2018), the scientific literature lacks information on the role of monoterpenes on larger spatial (e.g. entire southeast U.S.) and temporal (different times of year)

scales which are included in our work. We support our hypothesis based on a weight of evidence provided in the literature and this study, as listed and discussed below.

(1) The large emissions of MT and SQT in the southeastern U.S. (Guenther et al., 2012), which has been established in decades and the reviewer agrees with.

(2) The majority (roughly 80%) of carbon in SOA is modern in the southeastern U.S. The reviewer suspects that this evidence is not quite consistent with some modeling studies. We beg to differ due to following reasons. Firstly, Weber et al. (2007) measured that the biogenic fraction of carbon is roughly 70-80% at two urban sites in Georgia that were also used in our study. Note that measurements in Weber et al. (2007) were performed in 2004 and the biogenic fraction is expected to be higher in 2016 than 2004, as a result of reductions in anthropogenic emissions (Blanchard et al., 2010). Secondly, we checked Kim et al. (2015) and found that the paper clearly stated that "we estimate that 18% of the total OC burden is derived from fossil fuel use. This is consistent with an

18% fossil fraction from radiocarbon measurements made on filter samples collected in Alabama during SOAS." In brief, Kim et al. (2015) is consistent with other studies (Zhang et al., 2018;

Lewis and Stiles, 2006; Weber et al., 2007).

(3) Previous studies suggest that the oxidation of β-pinene (another important monoterpene) by nitrate radicals ($NO_3$) contributes to LO-OOA in the southeastern U.S. (Boyd et al., 2015; Xu et al., 2015a) and this reaction alone cannot replicate the magnitude of LO-OOA (Pye et al., 2015).

(4) The mass spectra of LO-OOA are almost identical (i.e., R ranges from 0.95 to 0.99 in Fig. R1)

across all the seven datasets in our study. In addition, LO-OOA across all datasets also shares the same diurnal trends (Xu et al., 2015a). The similarity in LO-OOA features suggests that LO-OOA

may share similar sources across multiple sites and in different seasons in the southeastern U.S.

(5) Perturbation experiments in this study show that the majority of fresh SOA from the oxidation of MT and SQT contributes to LO-OOA. Previous studies, mainly based on mass spectra comparison, concluded that MT SOA contributes to LO-OOA, but did not quantitatively show the fraction of MT SOA that is apportioned into LO-OOA. In other words, previous studies did not show whether 100% or 50% of MT SOA is apportioned in to LO-OOA. The quantitative understanding is the basis when comparing modeled MT SOA with PMF factors. The reviewer raises concern regarding this conclusion in next comment and we will address his/her concern later.

(6) CMAQ model calculations for the region showed consistency between modeled $SOA_{MT+SQT}$

and observed LO-OOA in terms of both magnitude and diurnal trend at different sites and in different seasons when an updated monoterpene SOA parameterization was used.

The new VBS parameterization implemented in the updated simulation represents a significant scientific improvement over the Odum 2-product parameterization currently used in the public version of CMAQ (v5.2). Specifically, the VBS parameterization does promptly form low volatility species, likely from autoxidation, which were absent from the previous Odum-2 product parameterization. In addition, the new parameterization allows for enthalpies of vaporization that are more consistent with species of the specified volatility, since the parameterization was produced from a richer dataset than the original Odum 2-product representation. The work shown here is an important step in the right direction and will allow for an improved representation of monoterpene SOA in current models while mechanistic pathways are still being determined. As shown in Fig. S16 in revised SI, implementing the new parameterization of MT SOA substantially reduces the normalized mean bias (NMB) between modeled and measured OA for all six datasets.

(7) A recent study by Zhang et al. (2018), which was published after our manuscript submission, offered other evidence to support our hypothesis. Zhang et al. (2018) characterized the molecular tracers of MT SOA at Centreville, AL (a site included in our study as well) and concluded that monoterpenes are the largest source of summertime organic aerosol in the southeastern U.S.

Therefore, we use above weight of evidence to support our hypothesis. We have revised the manuscript to clarify the logic and avoid confusions.

At last, we fully acknowledge the progress already made and the need to improve our understanding of atmospheric chemistry and all the unknowns the reviewer brought up. In fact, our study is motivated by many unknowns that the reviewer brought up. For example, due to the high O:C ratio of highly oxygenated molecules (HOMs) formed during monoterpene oxidation, it is hypothesized that HOMs maybe a potential source of MO-OOA. This hypothesis challenges our current understanding that MO-OOA represents aged SOA and also raises the question if monoterpenes SOA is exclusively apportioned into LO-OOA. Although previous studies repeatedly showed the similar mass spectra between α-pinene SOA and LO-OOA, the mass spectra comparison approach cannot tell us what is the fraction of α-pinene SOA apportioned into LO-OOA vs. other factors. The limitations of mass spectra comparison approach motivate us to explore alternative approaches to understand the sources of PMF factors. Another example is still related to HOMs. As the reviewer is aware of, the formation of HOMs and the contributions of HOMs to SOA are not captured by the Odum 2-product model implemented in current regional models which do not include prompt formation of material with saturation concentrations less than 10 $\mu g/m^3$. This is one of the reasons we replaced Odum 2-product model with VBS parameterization in the updated simulation. The new parameterization based on Saha and Grieshop (2016) considers the HOMs contribution to SOA and the HOMs yield in Saha and Grieshop (2016) is consistent with recent observations.

Throughout the manuscript, we never imply that SOA sources, other than monoterpenes and sesquiterpenes, are not important. Based on our measurements, LO-OOA accounts for 19-34% of total OA in the southeastern U.S. The sources of MO-OOA, which accounts for 24-49% of OA in the southeastern U.S., are highly uncertain. Many reaction pathways the reviewer brought up are actually potential sources of MO-OOA. For example, SOA produced from aqueous-phase chemistry is generally highly oxidized (Lee et al., 2011) and is likely apportioned into MO-OOA, instead of LO-OOA. A recent study by Xu et al. (2016) suggests that aqueous-phase reaction has a dominant impact on MO-OOA in China. There are also hypotheses in the literature that the SOA formed through cloud chemistry together with long-range transport and entrainment from aloft may contribute to MO-OOA (Crippa et al., 2013; Robinson et al., 2011; Xu et al., 2015b). In brief, we never make any implication that SOA from other sources/pathways are not important in the southeastern U.S.

*A little more detail on the major issues*

*2) A very important problem with the manuscript is the logic of the PMF analysis (with both of the other reviewers explicitly pointing out serious problems in it). Indeed the a-pinene SOA is most similar to LOOOA (also referred to as SV-OOA in older works). This has been known for a decade, see for example Fig 2C of Jimenez et al. (2009) (cited in the manuscript) and several other papers. Or in the words of reviewer 1: "the case for a-pinene SOA being a strong contributor to LO-OOA has never really been in doubt in this reviewer's opinion. Why is the evidence presented here any*

*more 'direct' than those published previous?" The same conclusion can be deduced quickly by*
*comparing spectra of chamber apinene SOA and ambient LO-OOA from the AMS spectral*
*databases (much like the authors do again in their Fig. 6). So this it is not a new finding of this*
*work. Here an interesting (though not completely new) experimental procedure is used to reach*
*once again a conclusion that was already firmly established in the literature. See for example the*
*figure below (Kiendler-Scharr et al., 2009, as an example, but there are quite a few such*
*comparisons in the earlier AMS literature. Note that the Chebogue BSOA represents the outflow*
*for the E US, similar to the work in the present paper).*

*Therefore the conclusion is not really new and has been known for over a decade. So it seems*
*surprising that suddenly the same specific result allows the present authors to reach far more*
*general conclusions. The critical flaw is in the logic flow. It is true that (a) a-pinene SOA is most*
*similar to LO-OOA. However, that cannot be used to conclude that (b) ambient LO-OOA in the*
*SE US is mostly from apinene. Hypothesis (a) being true is a necessary, but not sufficient condition*
*for the converse hypothesis (b) to be true. Much more evidence is needed to prove hypothesis (b)*
*and to disprove alternative hypotheses.*

*A critical piece of additional evidence would include proving that spectra from other sources of*
*SOA likely to be present in the region are not spectrally similar to LO-OOA. This is not shown in*
*the manuscript. Experiments with other precursors were attempted (L313): "by injecting isoprene,*
*mxylene, or naphthalene, which are major biogenic and anthropogenic emissions, respectively.*
*However, the SOA formation from these VOCs is not detectable." So the experiments failed, and*
*as a consequence no spectra from other likely sources are available to establish that spectra from*
*those sources do not look like LO-OOA. A look at the AMS literature and the AMS databases*
*suggest that spectra from these and other precursors have spectra which are indeed very similar*
*to ambient LO-OOA. So it is very problematic that the experiments did fail.*

*Inexplicably the authors do not use the publicly available database spectra, nor perform regular*
*chamber experiments for other precursors, and have a very handwavy section (sect. 3.4) trying to*
*justify that. This is simply not acceptable. I do not agree that the ambient perturbation experiments*
*are better than regular chamber experiments, but I would agree that they can be just as good.*
*Adding a-pinene to ambient air, and using the oxidants (O3 and OH) and OA seed from ambient*
*air, is not significantly different to injecting O3 in a chamber and using either a sulfate seed or no*

*seed. If the authors expect that the spectra would be significantly different, they should explain*

*why this would be. But their own Figure 6 indicates that the spectra from the ambient perturbation*

*experiments and a regular chamber are indeed very similar. Why then not use chamber*

*experiments (either from the literature database, or the authors' chamber) to obtain spectra from*

*other precursors and pathways, and see whether they are similar to the ambient LO-OOA or not?*

Response: The reviewer's comment targets on why our conclusions from perturbation experiments are more "direct" than previous studies based on mass spectra comparison method. We have carefully addressed this question in response to reviewer#1's comment#9. In brief, in the authors'

opinion, the mass spectra comparison approach is subjective and qualitative. It relies on subjective judgement to determine whether lab SOA is similar to OA factor. Also, even if the mass spectrum of α-pinene SOA is similar to LO-OOA, the similarity is not equivalent to that α-pinene SOA is exclusively apportioned into LO-OOA. In contrast, in the perturbation approach, PMF analysis does not distinguish SOA from natural α-pinene vs. from injected α-pinene, so that PMF analysis can objectively apportion α-pinene SOA into factors. Further, we attempt to quantify the fraction of α-pinene SOA that is apportioned into different factors.

The reviewer also argued that it is problematic that the perturbation experiments with other

VOCs (isoprene, m-xylene, or naphthalene) failed. In fact, the results with other VOCs are expected and explainable as will be discussed in response to reviewer's comment #7.

*3) Similar experiments and analysis (VOC addition to ambient air in a flow reactor, followed by*

*spectral comparison) have already been conducted and published by Palm et al. (2018) as part of*

*the GoAmazon campaign, an area comparable to the SE US with high biogenic impact, but also*

*other sources. In that work multiple VOCs were added to ambient air individually, and oxidized*

*to form SOA in an oxidation flow reactor, which was then measured by an HR-AMS. Those authors*

*were able to oxidize bcaryophyllene, longifolene, limonene, b-pinene, α-pinene, toluene, and*

*isoprene, and to obtain yields and HR-AMS mass spectra for all of them. They further state: "The*

*mass spectrum of the SOA formed from OH oxidation [of ambient air] was correlated (R2=0.72-*

*0.93; shown in Fig. S12) with spectra of the SOA formed from the injected VOCs from the standard*

*injection experiments in Sect. 3.4. These correlations show that the SOA formed from OH oxidation*

*of ambient air appeared similar to SOA from known precursors, but the spectra from the different*

*precursors appear too similar to be able to differentiate the SOA sources in ambient air from the*

*spectrum alone." Fig. S12 of that work is reproduced below for reference.*

*Thus the difficulty of apportioning the SOA in ambient air through this type of experiments, and*

*given the high fragmentation and limited information content of AMS spectra is clear and has been*

*previously documented. If the authors disagree, the burden of proof is on them to show that they*

*can unequivocally associate ambient air spectra with those of specific VOC precursors, including*

*disproving that ambient LO-OOA may have major contributions from other precursors and*

*pathways. Let's imagine that Palm et al. had only conducted experiments with limonene, and the*

*rest of the experiments had failed. Then they would have observed R2 of 0.9 between SOA of those*

*experiments and SOA from ambient air. Would that have been sufficient evidence to justify the title*

*"Large Contributions from limonene to Organic Aerosol in the Amazon"? That paper performed*

*additional analyses, and concluded instead that biomass burning and anthropogenic precursors*

*were also important contributors at that particular location, in addition to biogenics. Therefore*

*one has to avoid making expansive conclusions based on narrow evidence.*

Response: Firstly, we do not agree with the reviewer that the analysis in Palm et al. (2018) is similar to our study. Palm et al. (2018) focused on the SOA formation from oxidizing ambient air with OH in an oxidation flow reactor. This SOA is referred to as "potential SOA". Both the mass spectra comparison and the "additional analysis" the reviewer referred to aim to understand the

"potential SOA", instead of the existing SOA in the atmosphere. As clearly stated in Palm et al.

(2018), "Importantly, this analysis does not provide information about what amounts or fractions of the preexisting (i.e., ambient) OA measured at the T3 site came from each of these sources."

Thus, the analysis in Palm et al. (2018) is not similar to our study, as our study aims to understand the sources of preexisting OA.

The reviewer quote from Palm et al. (2018) that "…*but the spectra from the different*

*precursors appear too similar to be able to differentiate the SOA sources in ambient air from the*

*spectrum alone.*". The $R^2$ in Palm et al. ranges from 0.72 to 0.93. We would like to ask the question what counts as "too similar"? We believe that the mass spectra comparison method cannot provide an objective answer to this question, but the PMF analysis as done in our study can potentially answer the question. That is, if Palm et al. (2018) performed PMF analysis on the combined ambient data and perturbation data, SOA from different precursors may be apportioned into
different factors.

Secondly, the reviewer raised one question that if Palm et al. had only conducted
experiments with limonene, would that have been sufficient evidence to justify the title "Large
Contributions from Limonene to Organic Aerosol in the Amazon"? We agree with the reviewer
that the answer is definitely no, because this conclusion is drawn based on simply mass spectra
comparison, instead of cumulative evidence as in our study. Let's imagine that Palm et al. simulate
the SOA formation from limonene in Amazon, would that match the LO-OOA in both magnitude
and diurnal trend at multiple sites and in different seasons? Is limonene the most abundant
monoterpenes in the Amazon? Is there any study to objectively show that the majority of limonene
SOA contributes to LO-OOA, instead of other factors? We state again that our hypothesis is
supported by a weight of evidence, instead of simple mass spectra comparison.

Thirdly, to support the hypothesis that LO-OOA is largely from the oxidation of
monoterpenes and sesquiterpenes in the southeastern U.S., we have shown cumulative evidence in
response to this reviewer's comment#1. Below, we list more evidence to support that LO-OOA
unlikely has major contributions from anthropogenic VOCs.

(1) m-xylene, an important and abundant anthropogenic VOC is likely apportioned to MO-
OOA. The mass spectrum of laboratory-generated m-xylene SOA (Bahreini et al., 2005) is more
similar to the MO-OOA (R = 0.97) than LO-OOA (R = 0.83), using the standard mass spectra in
Ng et al. (2010). Using the reviewer's recommendation to leverage the similarity of spectra
between laboratory experiments and the ambient measurements, m-xylene SOA contributes to
MO-OOA, instead of LO-OOA.

(2) As indicated in Fig. S8 in revised SI, the modeled concentration of SOA from
anthropogenic VOCs is on the order of 0.1 $\mu g \ m^{-3}$. Even if we double the SOA yields of
anthropogenic VOCs to account for the potential vapor wall loss in laboratory studies (Zhang et
al., 2014) and even if we assume all SOA from anthropogenic VOCs oxidation contributes to LO-
OOA, anthropogenic SOA only account for a small fraction of LO-OOA. Also, the modeled
anthropogenic SOA peaks in the day, which is different from that of LO-OOA, which reaches
daily maximum at night. This small amount of anthropogenic SOA is consistent with Zhang et al.

(2018), who performed molecular-level characterization of OA and showed that anthropogenic

SOA only accounts for 2% of total OA in Centreville, AL.

*4) Some results of the PMF analysis appear to have been misinterpreted. The ambient data had*

*10 times as many points in time as the perturbation experiments. In such a situation, the ambient*

*data effectively "dictates" the spectra due to its much higher fractional contribution to the Q value*

*(weighed residual that PMF is minimizing). As expected in this situation (L167) "the perturbation*

*experiments do not create a new factor that does not already exist in the ambient data." Under*

*that setup, the PMF results for the chamber time periods are similar to multiple linear regression*

*onto the spectra already predetermined for ambient air (since the chamber results have a low*

*contribution to Q, and then are unable to change the factor spectra much).*

*As the spectra of SOA from the chamber are being forced to be represented by a linear combination*

*of a limited set of ambient spectra (which they cannot influence in practice), it is normal that some*

*SOA spectra from the chamber VOC additions "project" onto more than one ambient spectrum.*

*That is, if an SOA spectrum produced in the chamber is not exactly similar to one of the ambient*

*spectra, PMF may be able to reduce the residual by representing those experimental spectra as a*

*linear combination of two of the "basis" spectra that were determined primarily from the ambient*

*data. Those results are expected and not surprising. They do not "clearly demonstrate" at all that*

*if a given type of SOA was present in an ambient dataset, it would be split into two factors in the*

*same way. Mathematically these are two very different situations, with very different structures*

*and residual contributions. For example an ambient cooking OA (COA) factor would have a*

*different time variation, which PMF would also exploit in extracting the factors, but such*

*structural difference in the variance is not present here. Experiments (for example using simulated*

*data) could be carried out to investigate the interference point for ambient data alone.*

*Unfortunately, the authors misinterpret their results for a very specific PMF situation for non-*

*ambient data, into completely unsupported general statements for PMF analyses of ambient data*

*(that for example ambient COA may contain caryophillene SOA, or that isoprene SOA may have*

*interferences from a-pinene SOA), even though such analyses have not been performed in this*

*work! Perhaps those interferences exist, but they have not been proven by this study. Rather here*

*the results of a complex PMF analysis are being misinterpreted.*

*In fact, the result that some of the a-pinene and carophyllene SOA apportions to other factors in*
*the authors' methods weakens the main stated conclusions further. Even these types of SOA are*
*not quite well represented by LO-OOA and need to "lean" on other ambient factors to reduce the*
*residual in PMF. Therefore making the conclusion that ambient LO-OOA is mainly from MT &*
*SQT is even more weakly supported.*

Response: As stated in the manuscript, we designed our experiments in a way that the perturbation
experiments do not influence the mass spectra of OA factors and would not create a new factor.

The reviewer argued that "*PMF results for the chamber time periods are similar to multiple*
*linear regression onto the spectra already predetermined for ambient air*". Let's put aside whether
this interpretation is correct, we think the same argument/interpretation also applies for ambient
monoterpenes SOA. Imagine that there is a short period of ambient data with increasing
monoterpenes SOA concentration. For this short period, one can also argue that this short period
has small contribution to overall Q value and thus PMF results for this period are similar to
multiple linear regression onto the spectra already predetermined for ambient air. The perturbation
experiments simulate this short period with increasing α-pinene SOA concentration. It does not
matter how PMF treats the perturbation experiments, as long as the treatment is the same for
ambient data. Therefore, conclusions drawn from perturbation experiments are applicable to
ambient data from a similar situation. The perturbation experiments point out the possibility that
isoprene-OA factor could have interference from α-pinene SOA. This interference has not been
acknowledged in previous studies and there is no study in the literature to prove that this
interference does not exist. In fact, the interference of α-pinene SOA on isoprene-OA factor helps
to address some uncertainties regarding the isoprene-OA factor in the literature. For example, Liu
et al. (2015) compared the mass spectrum of laboratory-derived IEPOX SOA with isoprene-OA
factors at some sites. The authors observed stronger correlation for isoprene-OA factors resolved
at Borneo (Robinson et al., 2011a) and Amazon (Chen et al., 2015), and weaker correlation at
Atlanta, U.S. (Budisulistiorini et al., 2013) and Ontario, Canada (Slowik et al., 2011). As another
example, the fraction of measured total IEPOX-SOA molecular tracers in isoprene-OA factor
highly varies with location, ranging from 26% at Look Rock, TN (Budisulistiorini et al., 2015) to
78% at Centreville, AL (Hu et al., 2015). To address the uncertainties in above two examples, one
possible reason is that the isoprene-OA factors resolved at different sites are not purely from
IEPOX uptake. Isoprene-OA factors likely have interference from monoterpenes SOA or other sources, but the interference magnitude varies with locations. We hope to use this study to raise the public awareness of the possible interference in OA factors.

The reviewer proposed a great suggestion to use simulated dataset to investigate the potential interference. However, a great amount of work is required to fully carry out this idea, as the creation of the simulated dataset (i.e., what mass spectrum, time series, and concentration of

α-pinene SOA should be used?) is complicated and subjective. It would be an entire study in itself.

We agree with the reviewer that the perturbation experiments do not simulate all scenarios in the atmosphere and do not consider the temporal variation. The applicability of the conclusions drawn from the specific scenario to general atmosphere warrants further exploration. We have clearly discussed the caveats of the conclusions in the revised manuscript.

We agree with that PMF separates factors based on features in mass spectrum and time series. However, we do not agree that the temporal variations of monoterpenes and sesquiterpenes

SOA and COA can guarantee a clean separation between different sources. For example, the concentrations of many oxidation products of monoterpenes (e.g. pinonic acid, pinic acid, etc),

COA, and LO-OOA show similar increasing trend near sunset (Allan et al., 2010; Xu et al., 2015b;

Zhang et al., 2018). As an attempt to test if our conclusion is affected by the temporal variation, we performed perturbation experiments at different times of day (9am to 9pm) in this study.

*5) No uncertainty analysis, such as from bootstrapping, is performed for the PMF results. This is*

*more glaring given that very strong conclusions about the identity of a major fraction of the*

*ambient OA, and of potential interferences between factors are made, but we are not shown that*

*the results are even statistically significant or what the uncertainties in the analysis may be.*

Response: As the reviewer requested, we performed 100 bootstrapping runs to quantify the uncertainty of PMF results. As shown in Fig. R5. The statistical uncertainties in the time series and mass spectra of 5 factors are small and the PMF results reported in this study are robust.

[Figure]

Fig. R5. PMF results from bootstrapping analysis. (a) Average mass spectra (sticks) with 1-σ error bars (caps). (b) Average time series and 1-σ error bars (red).

*6) The statements about the novelty of the approach (L19, L21, L87-88, L457-458) are exaggerated.*

*As cited in the paper, both Leungsakul et al. (2005) and Palm et al. (2017) have already published*

*results from very similar experiments. The only difference between the present experiments and*

*those previous ones is (in the authors' words, L22-123) that "no extra oxidant precursors were*

*added into the chamber." That is correct, but would the authors expect that ambient O3 or OH*

*produced from ambient air would be that different than those produced in other ways? I am sure*

*that the a-pinene molecules don't care about how the O3 or OH colliding with them were formed.*

*If the authors expected that adding a-pinene to ambient air but using ambient oxidants was going*

*to significantly change the results compared to standard chamber experiments, the reasons for*

*such expectations should be discussed in detail.*

*Otherwise the experiments are interesting, but follow on prior publications. I mention this because*

*in some places (e.g. abstract L20-25, also in L87-88, L457-458) the statement that the experiments*

*were "novel" is somehow used to prop the weak conclusions. As if somehow the "novel"*

*experiments would have allowed the authors to reach some conclusions that were not reachable*

*by previous authors. But the novelty is very minor. And the one reason why the experiments are a*

*little different from previous experiments (no added oxidants) is the reason why the experiments*

*failed in multiple cases! Therefore the "novelty" does not provide any real support for the*

*conclusions.*

Response: As the reviewer noted, we already referenced and acknowledged previous studies which used ambient air. However, the goals of previous studies are completely different from that of our perturbation experiments. In Leungsakul et al., the main purpose of using rural ambient air is to flush the $270m^3$ outdoor chamber reactor. In Palm et al., their main goal is to measure the SOA

yield from individual VOCs in the OFR under ambient RH and temperature conditions. Our goal to use ambient air is to examine which factor the fresh α-pinene and β-caryophyllene SOA is apportioned into by PMF analysis. With our goal in mind, we want to produce SOA only from α- pinene and β-caryophyllene. The reviewer is totally right that the α-pinene molecules don't care about where the $O_3$ or OH comes from. However, adding extra oxidants will produce SOA via a number of reactions (i.e., oxidize other existing VOCs/SVOCs/IVOCs). If so, we would not unambiguously know if the LO-OOA enhancement in the perturbation experiments arise from injected VOC or from other pathways.

We realize that the description of the novelty of our approach is not accurate in many places and we have modified the language in the revised manuscript.

*7) Statements such as (L316-318) "The perturbation experiments with other VOCs confirm the*

*stronger ability of α-pinene and β-caryophyllene to produce SOA" are misleading, and frankly*

*just perplexing. A lot was already known about the relative potential of different VOCs to make*

*SOA before this paper, and nothing new is learned from the experiments here about this point. The*

*low SOA observed for other precursors is attributed (SI, L405-406 "to the low SOA yields or slow*

*oxidation rates of these VOCs (Ng et al., 2007). Yields for naphthalene are certainly not low, but*

*are higher than for a-pinene (Chan et al., 2009). Later it is acknowledged (SI L412-414) that*

*"isoprene oxidation products which form SOA are mostly second or higher generation products.*

*They are not formed in large amount in the relatively short perturbation experiments (i.e., 40min)."*

*The main issue is that a-pinene has a lifetime of ~1 h under the conditions of the ambient*

*perturbation experiments (using well-known rate constants) and caroyphyllene has an even*

*shorter lifetime. m-xylene has a lifetime of ~10 hrs, and therefore it reacts too slowly under the*

*conditions of these experiments with no added oxidants. And for some precursors (other than*

*isoprene) second or higher generation products are also needed to make SOA, which results in*

*even longer time constants. However, those timescales are readily accessible in the atmosphere.*

Response: The confusion regarding the referred statement mainly arises from the phrase "stronger ability". We realize that this phrase is not properly defined, but we do not think this statement is fundamentally wrong. Here, the "stronger ability of α-pinene and β-caryophyllene to produce SOA"

means that under the same atmospheric conditions (i.e., oxidants level, $NO_x$, per-existing particles, etc) and the same initial VOC concentration, more SOA would be produced from α-pinene and β- caryophyllene than from other VOCs (i.e., isoprene, m-xylene) after the same oxidation time (i.e.,

40min in perturbation experiments). This conclusion is well supported by laboratory studies in the literature. This is why we used the word "confirm" in the sentence.

The "*main issue is that ...*" brought up by the reviewer is exactly the same meaning as "the low SOA yields or slow oxidation rates of these VOCs", which we wrote in the original manuscript.

The timescale required to produce SOA of other VOCs (i.e., isoprene and m-xylene) is longer than our perturbation experiments. This is why we did not detect SOA formation from these VOCs in our experimental approach. Thus, the results are expected and explainable.

After submitting the manuscript, we realize another reason for the lack of SOA formation in naphthalene experiments. We injected naphthalene by passing pure air (1 liter per min) over the solid naphthalene flakes under ambient temperature for 1 min. Due to the relatively low vapor pressure of naphthalene (23.6Pa at 30°C) and rapid dilution in the chamber, the injected naphthalene concentration could be very low. We add this possible reason in the revised SI.

*8) The discussion in the introduction about MO and LO OOA is unclear. Those factors do not*

*represent the same sources or pathways in different studies. In some locations and times they are*

*tied to biomass burning, in others to urban emissions, in others to biogenic SOA, and in others to*

*various combinations. In older studies in which Isoprene-SOA (or "IEPOX-SOA) was not*

*separated, it was necessarily part of MO and LO OOA. Care should be taken to clearly*

*communicate that any interpretations about sources contributing to MO or LO OOA are specific*
*to a given location and time period. And the possibility should be considered (including in the SE*
*US) that different periods may have larger contributions to these factors from different sources,*
*for example BB during one period, pollution during another, and biogenics during another (if the*
*sources impacting a site change substantially in time due to air mass changes or other causes).*
*See e.g. Palm et al. (2018) for an example of such a situation.*

Response: We agree that the OOA factors represent different sources or pathways in different
regions. For example, in the original manuscript, we stated that "There is evidence that LO-OOA
in California is related to the oxidation of anthropogenic VOCs, as radiocarbon analysis suggests
68-75% of carbon in LO-OOA in California stems from fossil sources (Hayes et al., 2013; Zotter
et al., 2014)." We have further emphasized these points in the revised manuscript and expanded
the discussions on OOA factors in the introduction, as the reviewer requested.

While we agree with the reviewer that interpretations about the sources contributing to
OOA factors are location- and time-specific, we would like to point out that our study includes
measurements at multiple sites in the southeastern U.S. and in different seasons. The LO-OOA
across all datasets have similar diurnal variation (Xu et al., 2015a) and mass spectra (Fig. R1).
Moreover, the modeled $SOA_{MT+SQT}$ can capture the magnitude and diurnal variation of measured
LO-OOA at all datasets. These evidence suggests a general source of LO-OOA on a regional scale.

*Other points*

*9) L71-72: "The assumption that LO-OOA represents fresh SOA has yet to be directly verified."*
*Fig 2C, 2D, 2E in Jimenez et al. (2009) (where the older terminology SV-OOA was used for LO-*
*OOA), and similar results in other publications, would appear to have directly verified that long*
*ago.*

Response: Please see response to reviewer#1's comment#9.

*10) A paper with a similar title to this one, but using different lines of evidence, has been recently*
*published (Zhang et al., 2018).*

Response: Thanks for pointing this out. We note that Zhang et al. (2018) was published after our
manuscript submission. The conclusion in Zhang et al. (2018) is consistent with our study and has
been discussed in the revised manuscript.

[revised manuscript text omitted]

---

## Author Response (AR2)

We thank the reviewers for the detailed and insightful comments. We have addressed the reviewers' comments point by point as indicated below. The reviewers' comments are in italics and changes made to the manuscript are in quotation marks. Unless otherwise noted, the numbers of sections, figures, and lines mentioned in the response refer to those in the revised manuscript after the first round review.

Reviewer #1

*While I thank the reviewers for taking on board my comments and those of the other reviewers, I feel that there is still a fundamental issue to deal with in that the core conclusion as featured in the title (among other places), in that biogenic monoterpenes and sesquiterpenes are large contributors to SOA in the southeastern US, is not supported. While the authors have toned down the discussion in places and added caveats, there is still the major issue that the paper is constructed around a hypothesis that is merely 'supported' (chiefly in the new section 3.5) without adequately discounting the null hypothesis, i.e. that other precursors are really responsible for the SOA. This is a completely essential step in presenting any hypothesis-lead paper and as such, the hypothesis is not properly tested. There are a number of technical reasons raised by myself and other reviewers (reviewer 3 was far more strident than me, but I largely agree with the points they raise) why this particular line of argument could be considered weak, so as such, this represents a line of scientific reasoning that is not befitting an ACP article. It's also not helped by the authors' tendency in the modified article to discuss the merits of the various approaches after the observations have been reported, which gives the impression of a 'shoot first ask questions later' approach to the science, rather than a properly designed experiment.*

*I don't consider this paper to be unpublishable, however I would recommend that the paper be pitched differently, focusing on the new insights offered by the novel experimental procedure, rather than treating it as strong evidence in support of an apportionment SOA to precursors. Based off the discussion, I feel that the overall methodology will require significant development before it can really be used confidently for this type of apportionment. That's not to say the authors shouldn't carry forward the data as they have done, but it should be presented as a speculative projection rather than a conclusion, awaiting further evidence in support.*

Response: We thank the reviewer for the thoughtful comments. As suggested by the reviewer, we focus on the new insights offered by the novel experimental procedure, and we changed the title correspondingly to "Experimental and Model Estimates of the Contributions from Biogenic
Monoterpenes and Sesquiterpenes to Secondary Organic Aerosol in the Southeastern United
States".

We acknowledge the importance of discounting the null hypothesis. However, there are
hundreds to thousands of SOA precursors in the atmosphere and it is impossible to test all SOA
precursors. Therefore, we use multiple lines of evidence as discussed in section 3.4 of revised
manuscript and briefly listed below, to support our hypothesis (i.e., the major source of LO-OOA
in this region is the oxidation of monoterpenes and sesquiterpenes by various oxidants) and
discount the null hypothesis (i.e., the major source of LO-OOA is from the oxidation of precursors
other than monoterpenes and sesquiterpenes). We do not draw our conclusions based on the
perturbation experiments alone. Instead, all these together supported that a large fraction of SOA
in the southeastern U.S. arises from monoterpenes and sesquiterpenes:

(1) The large emissions of MT and SQT in the southeastern U.S. (Guenther et al., 2012).

(2) The majority (roughly 80%) of carbon in SOA is modern in the southeastern U.S. (Zhang et
al., 2018; Lewis and Stiles, 2006; Weber et al., 2007).

(3) Previous studies suggest that the oxidation of β-pinene by nitrate radicals ($NO_3$) contributes to
LO-OOA in the southeastern U.S. (Boyd et al., 2015; Xu et al., 2015a) though this reaction alone
cannot replicate the magnitude of LO-OOA (Pye et al., 2015).

(4) The mass spectra and diurnal trends of LO-OOA are almost identical across all the seven
ambient datasets in our study, suggesting that LO-OOA may share similar sources across multiple
sites and in different seasons in the southeastern U.S.

(5) Perturbation experiments in this study show that the majority of fresh SOA from the oxidation
of MT and SQT contributes to LO-OOA.

(6) CMAQ model calculations for the region showed consistency between modeled $SOA_{MT+SQT}$
and observed LO-OOA in terms of both magnitude and diurnal trend at different sites and in
different seasons when an updated monoterpene SOA parameterization was used.

(7) The emissions of anthropogenic VOCs are much weaker than that of biogenic VOCs in the
southeastern U.S. (Goldstein et al., 2009).

(8) We modeled that the concentration of anthropogenic SOA is on the order of 0.1 µg m$^{-3}$ for our datasets (Fig. S8). The low modeled concentration of anthropogenic SOA is consistent with the recent publication by Zhang et al. (2018), who showed that the measured tracers of anthropogenic

SOA only account for 2% of total OA in Centreville, AL.

As discussed in the manuscript, we do not conclude that LO-OOA arises exclusively from

MT and SQT, but that they contribute substantially to LO-OOA. It is possible that the oxidation of other VOCs contributes to LO-OOA. However, based on multiple lines of evidence discussed above, these other contributions are expected to be much smaller than that from biogenic monoterpenes and sesquiterpenes in the southeastern U.S.

The southeastern US is a unique location in that there have been a large number of field studies in recent years at multiple locations and seasons throughout the region (Carlton et al., 2018;

Zhang et al., 2018; Xu et al., 2015a; Warneke et al., 2016). Results from these studies provided additional constraints for OA sources in this region. In future studies, we agree that further developments are needed if one were to use the perturbation experimental approach for source apportionments of OA at other sites, but auxiliary constraints from field measurements/lab studies/modeling are not readily available for those sites. We have added this discussion to the revised manuscript to emphasize this point.

The new insights from perturbation experiments are indeed one focus of this study. In results sections 3.1 and 3.2, we thoroughly discuss how the perturbation experiments improve our understanding on the sources of OA factors, such as the potential interference of α-pinene SOA on isoprene-OA factor and the implications of this interference. In the revised manuscript, to further emphasize the insights from the perturbation experiments, we clearly discuss the limitations of traditional mass spectra comparison approach in the introduction section and explain the advantages of perturbation experiments in the experimental section. In section 3.4 of revised manuscript, we clearly state that the large contributions from monoterpenes and sesquiterpenes to

LO-OOA in the southeastern U.S. is our hypothesis and how this hypothesis is supported by multiple lines of evidence. This structure is in line with the reviewer's suggestion.

The reviewer also raised a comment regarding the discussions on the merits of the various approaches. This comment is addressed below in response to the reviewer's specific comment #1.

*1. While I thank the authors for addressing my query about 'direct evidence', I don't understand how the modified text really addresses this. Contrary to what is implied in the introduction the fact that a-pinene is a significant contributor to SOA and LO-OOA has never been in doubt; completely asides AMS evidence, we know it is an abundant VOC that produces SOA in chamber studies and that tracers for its SOA production (e.g. pinic acid) are present in ambient samples. Also, we know that the AMS spectra produced in chamber studies bear strong resemblances to LO-OOA spectra recorded in environments with strong a-pinene sources (e.g. temperate and boreal forests). Stating that this study is 'objective' because it also produces a similar mass spectrum isn't sufficient to demonstrate novelty over previous chamber studies. The authors need to focus on what this work adds that wasn't possible before. While they go to great lengths in the modified section 3.4 to compare with the 'mass spectral comparison approach', at the end of the day, all PMF is doing is comparing mass spectra, so I don't see what their point is. It's also very strange that the authors choose to justify their approach after the results section of the paper. Wouldn't it be more logical to provide justification at the design stage? As a constructive suggestion, I would try focusing on the fact that the same instrumental set-up was used for both ambient and chamber aerosols, making for a more seamless factorisation, free of any instrument tuning issues (the kind of thing that necessitates the use of arbitrary relaxation parameters when supplying target spectra in ME-2). It also provides actual 'mixing' with ambient aerosols, which a standard chamber experiment wouldn't do, meaning the performance of the factorisation can be more directly inspected (although I wouldn't regard that as a complete success in the sesquiterpene case; see my earlier 'red flag' comment in the first stage review). Naturally, this should be with the caveat that this doesn't disprove the null hypothesis; ambient SOA from other precursors could still be misclassified if they also produce LO-OOA with a similar mass spectrum, which I still consider to be a possibility.*

Response: We originally added the detailed discussion of various approaches in response to some comments raised by a reviewer in the previous round. However, upon further reading and consideration, we agree that this paragraph distracts the flow of the main discussions in the manuscript. Therefore, in light of the reviewer's suggestion, we now provide the justification of perturbation experiments in the design stage (i.e., section 2.1). We agree with the reviewer that we shall better point out the uniqueness of our study. We have highlighted the advantages of perturbation experiments that this approach is free of any instrument tuning issues and utilizes the
actual mixing between ambient SOA and chamber SOA.

We believe that the perturbation experimental approach provides more objective and
quantitative conclusions than the mass spectra comparison approach, mainly due to the following
reasons. Firstly, the conclusions from perturbation experiments are more quantitative than those
from mass spectra comparison approach. We agree with the reviewer that the mass spectrum of
laboratory α-pinene SOA bear strong resemblances to LO-OOA. However, this similarity does not
guarantee that all ambient α-pinene SOA is apportioned into LO-OOA and does not provide
quantitative insights into the fraction of ambient α-pinene SOA that is apportioned into LO-OOA.
Based on the perturbation experiments, we attempt to quantify the fraction of fresh α-pinene SOA
that is apportioned into different factors (i.e., ~80% into LO-OOA, ~20% into isoprene-OA, 0%
into MO-OOA, COA, and HOA). Secondly, the conclusions from perturbation experiments are
objective, while the mass spectra comparison approach is a subjective determination. For example,
the R values between laboratory generated α-pinene SOA (using HONO as OH source) with LO-
OOA, isoprene-OA, and MO-OOA in this study are 0.96, 0.88, and 0.81, respectively. In the mass
spectra comparison approach, using these R values to imply whether α-pinene SOA contributes to
a certain OA factor is subjective. On the other hand, although the PMF analysis on the perturbation
experiments still compares mass spectra, this "comparison" is objective. Because PMF analysis
does not distinguish SOA from α-pinene that is already in the atmosphere vs. from injected α-
pinene, α-pinene SOA can be objectively apportioned into factors.

In the introduction section, we have added the following discussions on the limitations of
traditional mass spectra comparison approach.

"Many different sources of LO-OOA and MO-OOA have been proposed primarily based
on comparing the mass spectra between ambient OA factors and laboratory-generated SOA
(Jimenez et al., 2009; Kiendler-Scharr et al., 2009; Ng et al., 2010). While the mass spectra
comparison approach largely improves our understanding of ambient OA factors, this approach
has the following limitations. Firstly, the similarity between two mass spectra is a subjective
determination. In other words, a good correlation coefficient (R) value between the mass spectra
of an ambient OA factor and a specific type of laboratory SOA does not imply that the laboratory
SOA contributes to the specific ambient OA factor. Secondly, such subjectively-defined similarity does not provide quantitative insights into the contribution of SOA from a certain source to a
specific OA factor. For example, previous studies have shown that the mass spectrum of laboratory
α-pinene SOA is the most similar to that of LO-OOA (Jimenez et al., 2009; Kiendler-Scharr et al.,
2009; Ng et al., 2010). However, this similarity neither guarantees that α-pinene SOA is
exclusively apportioned into LO-OOA, nor provides information regarding what fraction of α-
pinene SOA is apportioned into LO-OOA in ambient environments. Thus, uncertainties associated
with the sources of these OA factors still exist. Considering the large abundance of OOA subtypes
and their use as surrogates for ambient SOA, understanding the sources of compounds composing
these two OOA subtypes is critical to constrain atmospheric models and SOA budget."

In method section 2.6, we have added the following discussions on the advantages of the
perturbation experiments.

"The perturbation experiments are designed to address some limitations of the mass spectra
comparison approach by providing objective and quantitative evaluations. By producing SOA
from a known precursor, PMF analysis allows for the apportionment of the newly-formed SOA
into various factors without any subjective judgement on the similarity in mass spectra, and
provides quantification of the fraction of the newly formed SOA that is apportioned into each
factor. The perturbation experiments utilize the actual mixing between ambient OA and newly
formed SOA from perturbation, which a standard chamber experiment would not achieve, meaning
that the performance of the factorization can be more directly inspected. In addition, as the same
instrument set-up is used for both ambient sampling and perturbation experiments, factorization
results are free of instrument tuning issues."

*2. In their justification of the choice of precursors, the authors basically confirm that they chose*
*the ones they did because they are the ones they expected to be important. This isn't necessarily*
*'bad science', but this further weakens the case for the null hypothesis being discounted, as it adds*
*an element of confirmation bias to the experiment. The authors still don't really justify their choices*
*in the amount of precursor to be added either (they don't really give quantities), with very hand-*
*waving statements about 'too much SOA' remaining in the text. I can appreciate it might be difficult*
*to describe, but they should at least state the basis for their judgements, even if they were subjective*
*and of-the-moment.*

Response: We have tried isoprene, α-pinene, β-caryophyllene, *m*-xylene, or naphthalene, which are major biogenic or anthropogenic emissions. Due to reasons discussed in section S6, the perturbation experiment in current design is not suitable to study SOA formation from isoprene,

*m*-xylene, and naphthalene.

The values for the mixing ratios were provided in section 3.1 and 3.2 of previous manuscript version. We aim to inject as low of a VOC mixing ratio as possible to be atmospherically relevant. α-pinene or β-caryophyllene was injected via a needle into the chamber.

Limited by the needle size, 0.2 μL is the minimal amount we could add with reliable accuracy. 0.2

μL corresponds to a mixing ratio of 15 ppb and 10 ppb for α-pinene or β-caryophyllene, respectively, for a chamber volume of 2 $m^3$. The injection volume has been added in the revised manuscript.

"We aim to inject as low of a VOC mixing ratio as possible to be atmospherically relevant.

If the injection amount is too large, too much SOA will be produced, which will bias subsequent analysis. We use a needle to inject liquid sample into the chamber. Limited by the needle size, 0.2

μL is selected because it is the minimal amount we could inject with reliable accuracy."

*3. The modelling section in the paper still lacks a clear objective. It currently gives the impression*

*that the authors did the simulations for the sake of it and interpreted the results a posteriori, which*

*is again not conducive to rigorous hypothesis testing. To reiterate, being able to produce biogenic*

*SOA in a model isn't a new result and neither is changing said result when moving to a different*

*SOA scheme, so the authors need to be more explicit about what it is they are trying to achieve*

*with this exercise before launching into the results.*

Response: All models shall be able produce biogenic SOA as long as they have some biogenic precursors in the schemes. However, the key issue is whether the OA concentrations predicted by models are consistent with observational data. This is particularly problematic for regions where there are substantial anthropogenic-biogenic interactions, such as the southeastern U.S. For instance, regarding monoterpenes SOA, the recent publication by Zhang et al. (2018) showed that

CMAQ with default monoterpene SOA scheme significantly under-estimates the measured monoterpene SOA tracers in the southeastern U.S. Further, Pye et al. (2015) showed that CMAQ

modeled SOA from monoterpenes oxidation by $NO_3$ radical correlates well with measured LO-

OOA, but the magnitude is lower by roughly a factor of 2.

In our study, the objective of the modeling work is to simulate the SOA from monoterpenes
and sesquiterpenes ($SOA_{MT+SQT}$) in the southeastern U.S. and to compare the simulated
$SOA_{MT+SQT}$ with measured LO-OOA. We update CMAQ by incorporating two recent findings and
find that the $SOA_{MT+SQT}$ in the updated simulation reasonably reproduces the magnitude and
diurnal variability of LO-OOA at multiple sites and in different seasons. The reasonable agreement
between modeled $SOA_{MT+SQT}$ and LO-OOA serves as one piece of evidence to support our
hypothesis that LO-OOA largely arises from the oxidation of MT and SQT in the southeastern U.S.
We have added the objective of the modeling work at the beginning of section 2.6 in the revised
manuscript.

"To test the hypothesis that a large fraction of LO-OOA originates from monoterpenes and
sesquiterpenes in the southeastern U.S., we used the Community Multiscale Air Quality (CMAQ)
atmospheric chemical transport model to simulate the SOA from monoterpenes and sesquiterpenes
($SOA_{MT+SQT}$) in the southeastern U.S. and then compared the simulated $SOA_{MT+SQT}$ with measured
LO-OOA."

*4. As regards the new discussion of isoprene SOA, a major limitation of this experiment is that it*
*is strictly conducted at ground level, whereas some evidence from aircraft studies suggests that*
*production may be strongest at the top of the boundary layer (e.g. doi: 10.1029/2006JD008147).*
*This should be added as a caveat.*

Response: We have added this caveat in the revised manuscript.

"Third, the perturbation experiments are conducted at ground level, whereas evidence from
aircraft studies suggests that production of isoprene SOA may be stronger at the top of the
boundary layer (Allan et al., 2014)."

We believe the article doi: 10.1029/2006JD008147 suggested by the reviewer is a typo,
because the mentioned article discussed biomass burning, instead of isoprene OA.

Reviewer#2

*I wish to point out that I was not a reviewer of this manuscript in the first round of review.*
*Therefore, I have not read the original version of the paper. In an attempt to remain unbiased, I*
*read the updated manuscript without consideration of previous review comments or the authors'*
*response. My review comments are based on the updated version only, with small additions after*
*consideration of the review comments and authors' response.*

*This paper addresses a topic of general interest to the atmospheric chemistry/aerosol*
*community – the contribution of biogenic VOCs to secondary organic aerosol formation in the*
*atmosphere. In that regard, it is timely (especially given other recent publications) and well suited*
*for ACP. It extensively cites previous literature. In general, the text itself is well written (though*
*with some obvious typographical/syntax errors – please edit carefully). I'm not sure that the title*
*is appropriate for what is currently contained in the manuscript, but the abstract certainly covers*
*the wide array of topics included.*

*This last point about the abstract is one of my main concerns about the paper. The paper*
*seems to jump all over the place – which may be due to trying to incorporate reviewer responses*
*from the previous round. I actually would humbly suggest to the authors breaking up the paper*
*into multiple papers. For example, one paper could focus exclusively on the technique associated*
*with the lab-in-the-field – including discussion of the approach, statistical evaluations associated*
*with before and after perturbations (appendix A), and the limitations associated with the mass*
*spectral comparison technique. This paper could also include the limitations of this approach –*
*unknown mixing, not applicable for certain VOCs, etc. This would be appropriate for submission*
*to Atmospheric Measurement Techniques, for example. A second paper could then focus on use of*
*a-pinene and b-caryophyllene experiments to support their main conclusion and the modeling*
*(assuming all other issues are addressed). I actually think that Appendix B is too short to justify*
*inclusion. Too few experiments are performed – and including this section only increases its length*
*and makes this reader feel like we are going off on a tangent. Assuming the authors are not*
*amenable to splitting this paper, they must at least somehow improve the links between the various*
*sections. Again, assuming that the authors wish to pursue publication of this manuscript in its*
*current form, the following issues should be addressed prior to resubmission:*

Response: We thank the reviewer for the detailed comments. Taking the reviewer's comments into consideration, we changed the title correspondingly to "Experimental and Model Estimates of the Contributions from Biogenic Monoterpenes and Sesquiterpenes to Secondary Organic Aerosol in the Southeastern United States". Regarding the last point in the abstract, we have deleted the last sentence in the abstract to keep the manuscript better focused. Appendix B is added as suggested by a reviewer in the previous round. We prefer to keep Appendix B as the information could potentially be useful for other future studies.

*Specific Comments*

*1. Page 5, line 124. If the fans are shut off and the intake into the chamber is determined only by the instrumental pull, how do the authors ensure that the contents of the chamber are in fact well mixed? This subject comes up again in the box modeling section in which a dead volume (never actually defined) of 1.75 m3 (which is over 85% of the total chamber volume) is required to even come close to simulation of measured data? This to me implies that the chamber is not adequately characterized (such a characterization could appear in the suggested AMT paper).*

Response: The chamber is likely not fully well mixed, as suggested by the rapid decay of formed SOA. This was discussed in the SI section S6. However, a full mixing of the chamber is not required for evaluating which OA factors the SOA is apportioned into as both ambient OA and newly formed α-pinene SOA are sampled into AMS.

*2. Page 7, line 161. Introduction of the VOCs will naturally influence the levels of the oxidants, however (through consumption of O3 and subsequent formation of OH). The authors only present the average O3 levels in experiments and assume OH levels for modeling. Were time series of O3 within experiments investigated to see any influence of the VOC perturbation on O3?*

Response: The influence of VOC perturbation on $O_3$ concentration is minor, mainly because of the small amount of α-pinene consumption in the perturbation experiments. We added some new figures to show this. Figure R1 compares the $O_3$ concentration between "Chamber_Bf" (i.e., before α-pinene injection) and "Chamber_Af" (i.e., after α-pinene injection). The $O_3$ concentrations between two periods are very similar. In some experiments (e.g., ap_0718)1), the $O_3$ concentration is even higher in "Chamber_Af" than "Chamber_Bf" due to ambient variation. The time series of $O_3$ concentration from two α-pinene experiments (ap_0718_1 and ap_0719_3) are presented in Figure R2, which shows the minor effect of VOC injection on $O_3$ concentration.

[Figure]

Figure R1. The average O₃ concentration in Chamber_Bf and Chamber_Af periods in α-pinene experiments. The error bars represent the standard deviation.

[Figure]

Figure R2. The time series of O₃ in two α-pinene perturbation experiments.

*3. Page 7, line 178. I am somewhat troubled by the lack of VOC measurements in the experiments. How do the authors know that there were not issues with injection? Thus, the variations in amount of OA formed upon perturbation between experiments might not be due to changes in the ambient air into which the VOC is mixed but due to a change in the amount of VOC in the system.*

Response: We agree with the reviewer that the lack of VOC measurements is a disadvantage of
this study. We followed the same injection procedure for all perturbation experiments, which is
also the same protocol for our other indoor chamber experiments and the amount of VOC injected
has always been reproducible (where we have VOC measurements) (Boyd et al., 2015). If the
variations in the amount of OA formed upon perturbation between experiments are caused by
variations in the VOC injection amount, it would be too odd to observe the correlation between
amount of OA formation and $O_3$ concentration. Also, as our goal is to understand which OA factors
the α-pinene SOA is apportioned into, the uncertainties in injection would not affect any
conclusions regarding this goal.

*4. Page 12, line 310. The authors specify that they perform perturbation experiments at various*
*conditions. Did the responses to these conditions make sense? T? RH? Other factors?*

Response: The various conditions are inferred from different injection times (9am to 9pm). The T
and RH were not recorded. As both T and RH strongly depend on time of day, we plot the LO-
OOA enhancement amount as a function of injection time (Figure R3), but do not observe clear
diurnal variation.

[Figure]

Figure R3. Observations of trends in LO-OOA enhancement amount with injection time in α-
pinene perturbation experiments.

*5. Page 12, paragraph on line 319; page 13, entire page, page 1, line 366. This section deals with*
*changes in MO-OOA, IEPOX-OA, COA, and HOA when the VOC is perturbed by a-pinene or b-*
*caryophyllene. I find this to be a significant weakness of the paper – and along with the broad*

*array of topics covered, is the main basis for my recommendation. Why would MO-OOA only*

*change once? And why is IEPOX-OA only impacted when IEPOX-OA is non-zero in the ambient?*

*And why an entire paragraph about MO-OOA but only one sentence about HOA and COA? Upon*

*reading the other reviewer comments, it seems I am not alone in being troubled by this. Perhaps*

*this could be the subject of another paper – the weaknesses associated with the PMF technique? I*

*find in general that folks put a lot of faith in PMF results without often considering the fact that it*

*is a statistical technique whose goal is to minimize the residuals. I do not wish to belabor this point,*

*as Reviewer #3 from the previous review outlined reasons for this concern.*

Response: We are on the same page with the reviewer - learning more about the uncertainties of

PMF analysis is one goal of this study. For example, "IEPOX-OA factor" (with prominent signal at $m/z$ $C_5H_6O^+$) has been resolved from many prior studies, and this naming could give the impression that it is exclusively interpreted as SOA from the reactive uptake of IEPOX. In all studies from our group, we have always referred to this factor as "isoprene-OA factor" instead, as while it is mostly from IEPOX uptake, it is not entirely due to IEPOX (Xu et al., 2015b; Schwantes et al., 2015). Here, our perturbation results further suggest that the isoprene-OA factor could have interferences from α-pinene SOA and that caution is needed when interpreting the contribution of this factor.

We extensively discuss about MO-OOA because its sources are highly uncertain, which are discussed in Lines 67-74 of revised manuscript. Recent studies proposed that highly oxidized molecules (HOMs) from monoterpenes oxidation may be a source of MO-OOA. Thus, we would like to test this hypothesis from perturbation experiments. In contrast to MO-OOA, the HOA and

COA factors are relatively better understood. The lack of interference from α-pinene SOA to HOA

and COA supports our current understanding that both HOA and COA are dominantly from primary sources.

It is unclear why MO-OOA increases in one experiment. From the statistical point of view,

MO-OOA does not increase in 18 out of 19 experiments. Thus, we cannot draw any solid conclusion from only one experiment with enhanced MO-OOA formation.

Regarding the reviewer's question "*why is IEPOX-OA only impacted when IEPOX-OA is*

*non-zero in the ambient*", we mean that if the IEPOX SOA concentration is zero in the ambient environment and IEPOX-OA factor is not resolved from PMF analysis, α-pinene SOA would not be apportioned into a non-exist factor.

*6. Page 13, line 337, why is 4 ug/m3 insignificant with respect to g/p partitioning? If the total OA*

*is on the order of 10 ug/m3, this is 40% - and would likely be in the "steep" part of the Y-Mo curve.*

Response: For a semi-volatile compound with $C^* = 10$ µg m$^{-3}$, its fraction in the particle phase only changes from 0.5 to 0.58 when the OA concentration increases from 10 µg m$^{-3}$ to 14 µg m$^{-3}$.

Also, 4 µg m$^{-3}$ is the maximum value of the formed OA in perturbation experiments. The formation magnitude is lower than 4 µg m$^{-3}$ for most of experiments.

*7. Page 18, line 487. The description of model results indicates "it is not necessary to invoke any*

*unexplored mechanisms." This is potentially a dangerous statement, as it implies that they are*

*getting perfect modeling results of the exact same quantity – where in fact here they are supposing*

*that MT and SQT-based SOA = LO-OOA, and they are not getting perfect results!*

Response: We agree with the reviewer and have deleted this statement in the revised manuscript.

*8. Page 18, line 516. One point beyond 0.3 ppb does not signify a 'plateau.' The authors have no*

*idea (based on Figure 7) what is occurring between 0.3 ppb and 1.6 ppb. Also, the authors should*

*specify that the NO-free subscript refers to the H2O2 and the high-NOx subscript refers to NO2*

*and HONO. These are not labeled as such on the Figure.*

Response: We have modified the subscripts as suggested.

*9. Figure 2, the time series is unnecessary and adds nothing to the paper since it includes both the*

*ambient and chamber data. This is also true in Figure B1 (if appendix B is retained).*

Response: We think that the time series of OA factors present a full picture of PMF results. Thus we prefer to keep this in the manuscript for clarity.

*10. Appendix A, page 42, line 1208. The authors state that the chamber aerosol is lower than*

*ambient due to wall loss. How do they know this? Later, they indicate that wall loss plays a very*

*minor role in the change in LO-OOA. If the wall loss causes a significant change in the ambient*

*vs. chamber, why would it not cause just as significant a loss in the LO-OOA formed. As mentioned*

*above (page 5), is this a mixing issue (dead volume)?*

Response: The particle wall loss rate differs between the four periods in one perturbation
experiment. During the "Ambient_Bf" period, chamber is flushed with two fans on. In this stage,
the particle wall loss is large because the fans enhance the turbulence and hence the eddy
diffusivity in the chamber (Crump et al., 1982). This large particle wall loss leads to lower aerosol
concentration in the chamber than the atmosphere. However, during the perturbation
("Chamber_Af" period), the fans are turned off, leading to much slower particle wall loss rate than
"Ambient_Bf" period (i.e., fans on). During "Chamber_Af" period, wall loss still causes a loss in
the OA, but the loss is negligible compared to that caused by dilution with ambient air.

*11. Appendix A, page 43, line 1229. Why select the 8th point? Why not select the point where the*
*perturbation has a peak? How sensitive are the results to this?*

Response: We select the first 8 data points because the concentrations of total OA and OA factors
typically reach the highest at the 8th point (i.e., ~16min after injection). The results do not change
when the 7th point is selected. Selecting more than 8 points is not proper, because the
concentrations of OA factors start to decrease, which lowers the slope.

*12. Appendix A, page 44, paragraph starting on line 1247. I think it important that the descriptor*
*'pseudo' be used much more frequently to emphasize what is a 'pseudo' perturbation.*

Response: We have made the change as suggested.

*13. Appendix A, page 45, line 1287. What happened in the 5 cases when LO-OOA did not form?*
*Are these below some critical oxidant threshold? Or are you certain that the VOCs were actually*
*injected (see my previous point about the lack of VOC measurements)?*

Response: Among the 5 perturbation experiments when LO-OOA is not formed, two experiments
(ap_0728_1 and ap_0719_1) have the lowest $O_3$ concentration (<32ppb) among all 19 experiments.
The reasons for the lack of LO-OOA formation in the other three experiments are unclear. We
would like to emphasize that the conclusions are drawn based on statistical results, instead of a
single perturbation experiment, because reproducibility is challenging for this type of experiments
using ambient air.

*14. Supplement, page 12, line 323. I do not think it is appropriate for the authors to scale their*
*fresh SOA by 0.84 based on the regression shown in Figure S5. This ignores several points where*

*there was no increase in IEPOXOA – which would change the slope (and its associated R value)*
*considerably.*

Response: In Figure S5, if we include the experiments without isoprene-OA formation, the slope
is 0.14. This slope corresponds to a scale factor of 0.88, which is close to 0.84 used in the
manuscript. As acknowledged in the manuscript, the interference magnitude of α-pinene SOA on
isoprene-OA likely varies with locations and seasons and future studies are warranted to better
constrain the interference magnitude.

*15. Supplement, page 14, lines 380-400. Please see my earlier comment about wall loss and about*
*what a dead zone means (and how large it is relative to the total volume). In addition, how valid*
*is assuming that there is no inflow of a-pinene or OA in the model?*

Response: In Figure S21, the measured OA represents formed OA from α-pinene perturbation,
which has been corrected by the subtracting the ambient OA. Therefore, we assume no inflow of
ambient OA in the kinetic model and only simulates the SOA formation from the injected VOC.

In light of the reviewer's suggestion, we realize that the dead volume is not considered when
calculating the α-pinene concentration. We have now taken this into account and the revised figure
with a 1.75 m³ dead volume is shown below. Assuming a 1.75 m³ dead volume, roughly 10ppb α-
pinene (10% of initial concentration) is consumed by O₃. This amount likely serves as an upper
bound because a 10ppb decrease in O₃ concentration is not observed (Figure R1). This change
does not influence the discussion that only a small fraction of injected α-pinene reacts with
oxidants.

[Figure]

*16. Supplement, page 18, line 488. This observation of lower NO2 causing more LO-OOA is based*

*only on a small number of data points (3 at higher, 3 at lower). While the authors state that this*

*warrants further study, I'm not sure tossing in something with so few data points is appropriate.*

Response: We prefer to keep the results in the Supplement because they are reproducible from three pairs of experiments. This information could potentially be useful for other future studies.

*17. Figure S12. The y-axis label should say 'chamber' not 'bag' for consistency with the text.*

Response: We have made the suggested change.

*18. Figure S14. The authors state that the consistent fractional contributions of the various OA*

*factors remains relatively constant. Without error bars, it is hard to say whether this is statistically*

*accurate. For example, it looks as if the HOA fraction is decreasing with respect to time. Is this*

*because HOA emissions have decreased while other OA has increased/stayed constant? What is*

*the overall OA level in each year?*

Response: We have added the error bars to the plot. We also include a figure to show the changes in the concentrations of OA factors over five years. The HOA concentrations are relatively constant over the 5 years. The decrease in HOA fraction is mainly due to the increasing concentration of other factors.

*19. Figure S15. Is the use of yields at a loading of 445 ug/m3 appropriate? This seems rather large.*

Response: Saha and Grieshop (2016) used the thermal-denuder heating-induced evaporation of particles to derive the yield curve, based on the volatility distribution of the equilibrated suspended

– wall aerosol system. As discussed in Saha and Grieshop (2016), using high OA loading can reduce the effect of vapor wall loss on the aerosol system and provide more accurate volatility distribution.

Reference

[revised manuscript text omitted]